# On the Statistical Efficiency of Reward-Free Exploration in Non-Linear RL

**Jinglin Chen** *
Department of Computer Science
University of Illinois Urbana-Champaign
jinglinc@illinois.edu

**Aditya Modi** *
Microsoft
admodi@umich.edu

**Akshay Krishnamurthy**
Microsoft Research
akshaykr@microsoft.com

**Nan Jiang**
Department of Computer Science
University of Illinois Urbana-Champaign
nanjiang@illinois.edu

**Alekh Agarwal**
Google Research
alekhagarwal@google.com

## Abstract

We study reward-free reinforcement learning (RL) under general non-linear function approximation, and establish sample efficiency and hardness results under various standard structural assumptions. On the positive side, we propose the RFO-LIVE (Reward-Free OLIVE) algorithm for sample-efficient reward-free exploration under minimal structural assumptions, which covers the previously studied settings of linear MDPs (Jin et al., 2020b), linear completeness (Zanette et al., 2020b) and low-rank MDPs with unknown representation (Modi et al., 2021). Our analyses indicate that the *explorability* or *reachability* assumptions, previously made for the latter two settings, are not necessary statistically for reward-free exploration. On the negative side, we provide a statistical hardness result for both reward-free and reward-aware exploration under linear completeness assumptions when the underlying features are unknown, showing an exponential separation between low-rank and linear completeness settings.

## 1 Introduction

Designing a reward function which faithfully captures the task of interest remains a central practical hurdle in reinforcement learning (RL) applications. To address this, a series of recent works (Jin et al., 2020a; Zhang et al., 2020b; Wang et al., 2020a; Zanette et al., 2020b; Qiu et al., 2021) investigate the problem of reward-free exploration, where the agent initially interacts with its environment to collect experience ("online phase"), that enables it to perform *offline learning* of near optimal policies for any reward function from a potentially pre-specified class ("offline phase"). Reward-free exploration also provides a basic form of multitask RL, enabling zero-shot generalization, across diverse rewards, and provides a useful primitive in tasks such as representation learning (Agarwal et al., 2020; Modi et al., 2021). So far, most of the study of reward-free RL has focused on tabular and linear function approximation settings, in sharp contrast with the literature on reward-aware RL,

---

*Equal contribution

36th Conference on Neural Information Processing Systems (NeurIPS 2022).

| | Setting | Reference |
|---|---|---|
| 1 | Linear MDP | Wang et al. (2020a) |
| 2 | Linear completeness + explorability | Zanette et al. (2020b) |
| 3 | Completeness + Q-type B-E dimension | **Theorem 1** |
| 4 | Completeness + V-type B-E dimension + small $|\mathcal{A}|$ | **Theorem 3** |
| 5 | Low-rank MDP ($\phi^* \in \Phi$) + small $|\mathcal{A}|$ + reachability | Modi et al. (2021) |
| 6 | Linear completeness ($\phi^* \in \Phi$) + small $|\mathcal{A}|$ + reward-aware + explorability + reachability + generative model | **Theorem 5** (intractable) |

Table 1: Summary of our results and comparisons to most closely related works in reward-free exploration. Blue arrows represent implication ($A \to B$ means $B$ is a consequence of and hence weaker condition than $A$), and the red assumptions are what prior works need that are avoided by us. For linear settings, the true feature $\phi^*$ is assumed known unless otherwise specified (e.g., in Rows 5 & 6, $\phi^*$ is unknown but belongs to a feature class $\Phi$). "B-E" stands for (low) Bellman Eluder dimension (Jin et al., 2021). Row 6 has many assumptions, which make it strong since it is a negative result. The detailed comparisons of existing sample complexity rates and our corollaries can be found in Appendix A.

where abstract structural conditions identify when general function approximation can be used in a provably sample-efficient manner (Jiang et al., 2017; Jin et al., 2021; Du et al., 2021).

In this paper, we seek to bridge this gap and undertake a systematic study of reward-free RL in a model-free setting with general function approximation. We devise an algorithm, RFOLIVE, which is non-trivially adapted from its reward-aware counterpart (Jiang et al., 2017), and provide polynomial sample complexity guarantees under general conditions that significantly relax the assumptions needed by prior reward-free RL works. Our results produce both algorithmic contributions and important insights about the tractability of reward-free RL, as we summarize below (see also Table 1).

**Algorithmic contribution: beyond linearity** A unique challenge in reward-free RL is that the agent must exhaustively explore the environment during the online phase, since it does not know which states will be rewarding in the offline phase. A natural idea to tackle this challenge is to deploy a reward-aware RL "base algorithm" with the **0** reward function, since this algorithm must explore to certify that there is indeed no reward. Prior works adopt this idea with optimistic value-iteration (VI) approaches, which use proxy reward functions to drive the agent to new states. However these optimistic methods rely heavily on linearity assumptions to construct the proxy reward, and it is difficult to extend them to general function approximation. Instead of using optimistic VI, our basic building block is the OLIVE[2] algorithm of Jiang et al. (2017), a constraint-gathering and elimination algorithm that is a central workhorse for reward-aware RL with general function approximation. In the online phase of RFOLIVE, we run this algorithm with the **0** reward function, and we save the set of constraints gathered (in the form of *separate* datasets) for use in the offline phase.

**Algorithmic contribution: novel offline module** Prior works for reward-free RL typically use regression approaches (Ernst et al., 2005; Chen and Jiang, 2019; Jin et al., 2020b) in the offline phase, e.g., FQI (Modi et al., 2021; Zanette et al., 2020b), or its optimistic variants (Zhang et al., 2020b; Wang et al., 2020a). In the offline phase of RFOLIVE, rather than relying on regression, we enforce the constraints gathered in the online phase, which amounts to eliminating functions that have large average Bellman errors on state-distributions visited in the online phase. This generic elimination scheme does not rely on tabular or linear structures and allows us to move beyond these assumptions to obtain reward-free guarantees in much more general settings.

**Implications: positive results** The major assumptions that enable our sample complexity guarantees are Bellman-completeness (Assumption 2) and low Bellman Eluder dimension (Definition 5 and Definition 7); see Rows 3 and 4 in Table 1. These conditions significantly relax prior assumptions in the more restricted settings. Furthermore, prior works in the linear completeness and low-rank MDP settings require *explorability/reachability* assumptions (Zanette et al., 2020b; Modi et al., 2021),

---

[2]We use the Q-type and V-type versions of OLIVE from Jin et al. (2021) as their structural assumption of low Bellman Eluder dimension subsumes the low Bellman rank assumption in Jiang et al. (2017) (see Proposition 3).

which, roughly speaking, assert that every direction in the state-action feature space can be visited with sufficient probability. These assumptions are often not needed in reward-aware RL but suspected to be necessary for model-free reward-free settings. Our results do not depend on such assumptions, showing that they are not necessary for sample-efficient reward-free exploration either.

**Implications: negative results** We develop lower bounds, showing that some of the structural assumptions made here are not easily relaxed further. While the settings of linear completeness with known features (Row 3), and low-rank MDPs with unknown features (Row 4) are both independently tractable, we show a hardness result against learning under linear completeness when the features are unknown, even under a few additional assumptions (Row 6).

Taken together, our results take a significant step in bridging the sizeable gap in our understanding of reward-aware and reward-free settings and bring the two closer to an equal footing.

**Related work** In recent years, we have seen a wide range of results for reward-aware RL under general function approximation (Jiang et al., 2017; Dann et al., 2018; Sun et al., 2019; Wang et al., 2020c; Jin et al., 2021; Du et al., 2021). These works develop statistically efficient algorithms using structural assumptions on the function class. Despite their generality, a trivial extension to the reward-free setting incurs an undesirable linear dependence on the size of the reward class.

There also exists a line of research on reward-free RL in various settings: tabular MDPs (Jin et al., 2020a; Zhang et al., 2020b; Kaufmann et al., 2021; Ménard et al., 2021; Yin and Wang, 2021; Wu et al., 2022), MDPs with the linear structure (Wang et al., 2020a; Zhang et al., 2021; Zanette et al., 2020b; Huang et al., 2021; Wagenmaker et al., 2022), kernel MDPs (Qiu et al., 2021), block/low-rank MDPs (Misra et al., 2020; Agarwal et al., 2020; Modi et al., 2021), and multi-agent settings (Bai and Jin, 2020; Liu et al., 2021). Many of these settings can be subsumed by our more general setup.

Our offline module uses average Bellman error constraints, which is related to a line of work in offline RL (Xie and Jiang, 2020; Jiang and Huang, 2020; Chen and Jiang, 2022; Zanette and Wainwright, 2022). However, there is only one dataset in the standard offline RL setting, and these works form multiple average Bellman error constraints using an additional helper class for reweighting, and need to impose additional realizability- or even completeness-type assumptions on such a class. In contrast, we naturally collect *multiple* datasets in the online phase, so we do not require a parametric class for reweighting during offline learning.

## 2 Preliminaries

**Markov Decision Processes (MDPs)** We consider a finite-horizon episodic Markov decision process (MDP) defined as $M = (\mathcal{X}, \mathcal{A}, P, H)$, where $\mathcal{X}$ is the state space, $\mathcal{A}$ is the action space, $P = (P_0, \ldots, P_{H-1})$ with $P_h : \mathcal{X} \times \mathcal{A} \to \Delta(\mathcal{X})$ is the transition dynamics, and $H$ is the number of timesteps in each episode. If the number of actions is finite, we denote the cardinality $|\mathcal{A}|$ by $K$. In each episode, an agent generates a trajectory $\tau = (x_0, a_0, x_1, \ldots, x_{H-1}, a_{H-1}, x_H)$ by taking a sequence of actions $a_0, \ldots, a_{H-1}$, where $x_0$ is a fixed starting state and $x_{h+1} \sim P_h(\cdot \mid x_h, a_h)$. For simplicity, we will use $a_{i:j}$ to denote $a_i, \ldots, a_j$ and use the notation $[H]$ to refer to $\{0, 1, \ldots, H-1\}$. We use the notation $\pi$ to denote a collection of $H$ (deterministic) policy functions $\pi = (\pi_0, \ldots, \pi_{H-1})$, where $\pi_h : \mathcal{X} \to \mathcal{A}$. For any $h \in [H]$ with $h' > h$, we use the notation $\pi_{h:h'}$ to denote the policies $(\pi_h, \pi_{h+1} \ldots, \pi_{h'})$. For any policy $\pi$ and reward function[3] $R = (R_0, \ldots, R_{H-1})$ with $R_h : \mathcal{X} \times \mathcal{A} \to [0, 1]$, we define the policy-specific action-value (or Q-) function as $Q_{R,h}^\pi(x, a) = \mathbb{E}_\pi[\sum_{h'=h}^{H-1} R(x_{h'}, a_{h'}) \mid x_h = x, a_h = a]$ and state-value function as $V_{R,h}^\pi(x) = \mathbb{E}_\pi[Q_{R,h}^\pi(x, a_h) \mid x_h = x, a_h \sim \pi]$. We also use $v_R^\pi = V_{R,0}^\pi(x_0)$ to denote the expected return of policy $\pi$. For any fixed reward function $R$, there exists a policy $\pi_R^*$ such that $v_R^* = V_{R,h}^{\pi_R^*}(x) = \sup_\pi V_{R,h}^\pi(x)$ for all $x \in \mathcal{X}$ and $h \in [H]$, where $v_R^*$ denotes the optimal expected return under $R$. We use $\mathcal{T}_h^R$ to denote the reward-dependent Bellman operator: $\forall f_{h+1} \in \mathbb{R}^{\mathcal{X} \times \mathcal{A}}$, $(\mathcal{T}_h^R f_{h+1})(x, a) := R_h(x, a) + \mathbb{E}[\max_{a' \in \mathcal{A}} f_{h+1}(x', a') \mid x' \sim P_h(\cdot \mid x, a)]$ and similarly define $\mathcal{T}_h^{\mathbf{0}}$ for the operator with zero reward. The optimal action-value function (under reward $R$) $Q_R^*$ satisfies the Bellman optimality equation $Q_{R,h}^* = \mathcal{T}_h^R Q_{R,h+1}^*, \forall h \in [H]$.

---

[3] We consider deterministic reward and initial state for simplicity. Our results easily extend to stochastic versions.

**Reward-free RL with function approximation**  We study reward-free RL with value function approximation, wherein, the agent is given a function class $\mathcal{F} = \mathcal{F}_0 \times \ldots \times \mathcal{F}_{H-1}$ where $\mathcal{F}_h : \mathcal{X} \times \mathcal{A} \to [-(H-h-1), H-h-1], \forall h \in [H]$.[4] Without loss of generality, we assume $\mathbf{0} \in \mathcal{F}_h, \forall h \in [H]$ and $f_H \equiv \mathbf{0}, \forall f \in \mathcal{F}$. For any $f \in \mathcal{F}$, we use $V_{f,h}$ to denote its induced state-value function, i.e., $V_{f,h}(x) = \max_a f_h(x, a)$ and $\pi_f(x)$ as its greedy policy, i.e., $\pi_{f,h}(x) = \mathrm{argmax}_a f_h(x, a)$. When these functions take $x_h$ as input and there is no confusion, we may drop the subscript $h$ and use $V_f(x_h)$ and $\pi_f(x_h)$.

In reward-free RL, the agent is given access to a reward class $\mathcal{R}$, but the specific reward function is only selected after the agent finishes interacting with the environment. Specifically, the agent operates in two phases: an *online* phase where it explores the given MDP $M$ to collect a dataset of trajectories $\mathcal{D}$ without the reward information, and an *offline* phase, where it uses the collected dataset $\mathcal{D}$ to optimize for any revealed reward function $R \in \mathcal{R}$.

Our goal is to investigate the statistical efficiency of reward-free RL with general non-linear function approximation: how many trajectories does the agent need to collect in the online phase such that in the offline phase, with probability at least $1 - \delta$, for any $R \in \mathcal{R}$, it can compute a near-optimal policy $\pi_R$ satisfying $v_R^{\pi_R} \geq v_R^* - \varepsilon$? We measure the statistical efficiency in terms of the structural complexity of function class $\mathcal{F}$, reward class $\mathcal{R}$, horizon $H$, accuracy $\varepsilon$ and failure probability $\delta$.

As for expressivity assumptions, we assume the function class $\mathcal{F}$ is realizable and complete. Realizability requires that the optimal function $Q_R^*$ belongs to the reward-appended class $\mathcal{F} + R$, which is natural in the reward-free setting where the agent uses $\mathcal{F}$ to capture reward-independent information. Completeness requires that the Bellman backups of and $\mathcal{F}_{h+1} + R_{h+1}$ belong to $\mathcal{F}_h$, and additionally that the Bellman backup of $\mathcal{F}_{h+1} - \mathcal{F}_{h+1}$ belongs to $\mathcal{F}_h - \mathcal{F}_h$.

**Assumption 1** (Realizability of the function class). *We assume $\forall R \in \mathcal{R}, h \in [H], Q_{R,h}^* \in \mathcal{F}_h + R_h$, where $\mathcal{F}_h + R_h = \{f_h + R_h : f_h \in \mathcal{F}_h\}$.*

**Assumption 2** (Completeness). *We assume $\forall h \in [H], \mathcal{T}_h^0 \mathcal{F}_{h+1}, \mathcal{T}_h^0 (\mathcal{F}_{h+1} + \mathcal{R}_{h+1}) \subseteq \mathcal{F}_h$ and $\mathcal{T}_h^0 (\mathcal{F}_{h+1} - \mathcal{F}_{h+1}) \subseteq \mathcal{F}_h - \mathcal{F}_h$, where $\mathcal{F}_h - \mathcal{F}_h = \{f_h - f_h' : f_h, f_h' \in \mathcal{F}_h\}$.*

Next we define the covering number, which measures the statistical capacities of function classes.

**Definition 1** ($\varepsilon$-covering number, e.g., Wainwright (2019)). *We use $\mathcal{N}_\mathcal{F}(\varepsilon)$ to denote the $\varepsilon$-covering number of a set $\mathcal{F} = \mathcal{F}_0 \times \ldots \times \mathcal{F}_{H-1}$ under metric $\sigma(f, f') = \max_{h \in [H]} \|f_h - f_h'\|_\infty$ for $f, f' \in \mathcal{F}$. We define it as $\mathcal{N}_\mathcal{F}(\varepsilon) = \min |\mathcal{F}_{cover}|$ such that $\mathcal{F}_{cover} \subseteq \mathcal{F}$ and for any $f \in \mathcal{F}$, there exists $f' \in \mathcal{F}_{cover}$ that satisfies $\sigma(f, f') \leq \varepsilon$. For the reward class $\mathcal{R}$, $\mathcal{N}_\mathcal{R}(\varepsilon)$ is defined in the same way.*

Finally, as our guarantees depend on Bellman Eluder (BE) dimensions—which are structural properties of the MDP that enable sample-efficient exploration—we will need the following definitions (see Russo and Van Roy, 2013; Jin et al., 2021) which the later definitions of BE dimensions will build on.

**Definition 2** ($\varepsilon$-independence between distributions). *Let $\mathcal{F}'$ be a function class defined on some space $\mathcal{X}'$, and $\nu, \mu_1, \ldots, \mu_n$ be probability measures over $\mathcal{X}'$. We say $\nu$ is $\varepsilon$-independent of $\{\mu_i\}_{i=1}^n$ w.r.t. $\mathcal{F}'$ if $\exists f' \in \mathcal{F}'$ such that $\sqrt{\sum_{i=1}^n (\mathbb{E}_{\mu_i}[f'])^2} \leq \varepsilon$, but $|\mathbb{E}_\nu[f']| > \varepsilon$.*

**Definition 3** (Distributional Eluder (DE) dimension). *Let $\mathcal{F}'$ be a function class defined on some space $\mathcal{X}'$, and $\Gamma'$ be a family of probability measures over $\mathcal{X}'$. The DE dimension $d_{\mathrm{de}}(\mathcal{F}', \Gamma', \varepsilon)$ is the length of the longest sequence $\{\rho_i\}_{i=1}^n \subseteq \Gamma'$ s.t. $\exists \varepsilon' \geq \varepsilon$ where $\rho_i$ is $\varepsilon'$-independent of $\{\rho_j\}_{j=1}^{i-1}, \forall i = 1, \ldots, n$.*

We also introduce the notation $\mathcal{D}_\mathcal{F} := \{\mathcal{D}_{\mathcal{F},h}\}_{h \in [H]}$, where $\mathcal{D}_{\mathcal{F},h}$ denotes the collection of all possible roll-in distributions at the $h$-th step generated by $\pi_f$ for some $f \in \mathcal{F}$. Formally, $\mathcal{D}_{\mathcal{F},h} := \{d_h^{\pi_f}\}_{f \in \mathcal{F}}$ where $d_h^{\pi_f}(x, a) = \mathbb{P}_{\pi_f}[x_h = x, a_h = a]$ is the state-action occupancy measure.

## 3  RFOLIVE algorithm and results

In this section, we describe our main algorithm RFOLIVE, a reward-free variant of OLIVE (Jiang et al., 2017; Jin et al., 2021). The algorithmic template for RFOLIVE is shown in the pseudocode

---

[4]Since it is natural to use $\mathcal{F}$ to capture the reward-independent component (Assumption 1) in our reward-free setting, we assume $\mathcal{F}_h$ is upper bounded by $H - h - 1$. We include the negative range to simplify the discussions for various instantiations. Our main results also hold if we assume $\mathcal{F}_h : \mathcal{X} \times \mathcal{A} \to [0, H - h - 1]$.

([Algorithm 1](#)) and it can be instantiated with both Q-type and V-type versions of OLIVE from [Jin et al. (2021)](#).[5] In the pseudocode, we use □ as a placeholder for the respective Q/V-type definitions. For clarity, we will describe the Q-type RFOLIVE algorithm and its results in [Section 3.1](#) and then state the differences for the V-type version and corresponding results in [Section 3.2](#).

Before introducing our algorithm, we define the following average Bellman error:

**Definition 4** (Average Bellman error). *We denote $\mathcal{E}^R$ as the average Bellman error under reward $R$:*

$$\mathcal{E}^R(f, \pi, \pi', h) = \mathbb{E}\left[f_h(x_h, a_h) - R_h(x_h, a_h) - V_f(x_{h+1}) \mid a_{0:h-1} \sim \pi, a_h \sim \pi'\right].$$

*As shorthand, we use $\mathcal{E}_Q^R(f, \pi, h) = \mathcal{E}^R(f, \pi, \pi, h)$ to represent the Q-type average Bellman error and $\mathcal{E}_V^R(f, \pi, h) = \mathcal{E}^R(f, \pi, \pi_f, h)$ to represent the V-type average Bellman error ([Jin et al., 2021](#)). We use $\mathcal{E}^0$ to represent the average Bellman errors under $0$ reward.*

---

**Algorithm 1** RFOLIVE $(\mathcal{F}, \varepsilon, \delta)$: Reward-Free OLIVE

    **Online phase**, no reward information.
1: Set $\varepsilon_{\text{actv}}, \varepsilon_{\text{elim}}, n_{\text{actv}}, n_{\text{elim}}$ according to Q-type/V-type and construct $\mathcal{F}_{\text{on}} = \mathcal{F} - \mathcal{F}$.
2: Initialize $\mathcal{F}^0 \leftarrow \mathcal{F}_{\text{on}}$ (Q-type) or $\mathcal{F}^0 \leftarrow \mathcal{Z}_{\text{on}}$, where $\mathcal{Z}_{\text{on}}$ is an $(\varepsilon_{\text{elim}}/64)$-cover of $\mathcal{F}_{\text{on}}$ (V-type).
3: **for** $t = 0, 1, \ldots$ **do**
4:     Choose policy $\pi^t = \pi_{f^t}$, where $f^t = \arg\max_{f \in \mathcal{F}^t} V_f(x_0)$.
5:     Collect $n_{\text{actv}}$ trajectories $\{(x_0^{(i)}, a_0^{(i)}, \ldots, x_{H-1}^{(i)}, a_{H-1}^{(i)})\}_{i=1}^{n_{\text{actv}}}$ by following $\pi^t$ for all $h \in [H]$
       and form estimates $\tilde{\mathcal{E}}^0(f^t, \pi^t, \pi^t, h)$ for each $h \in [H]$ via [Eq. (1)](#).
6:     **if** $\sum_{h=0}^{H-1} \tilde{\mathcal{E}}^0(f^t, \pi^t, \pi^t, h) \leq H\varepsilon_{\text{actv}}$ **then**
7:        Set $T = t$ and exit the loop.
8:     **end if**
9:     Pick any $h^t \in [H]$ for which $\tilde{\mathcal{E}}^0(f^t, \pi^t, \pi^t, h^t) > \varepsilon_{\text{actv}}$.
10:    Set $\pi_{\text{est}} = \pi^t$ (Q-type) or $\pi_{\text{est}} = \text{Unif}(\mathcal{A})$, i.e., draw actions uniformly at random (V-type).
11:    Collect $n_{\text{elim}}$ samples $\mathcal{D}^t = \{(x_{h^t}^{(i)}, a_{h^t}^{(i)}, x_{h^t+1}^{(i)})\}_{i=1}^{n_{\text{elim}}}$ where $a_{0:h^t-1} \sim \pi^t$ and $a_{h^t} \sim \pi_{\text{est}}$.
12:    For all $f \in \mathcal{F}^t$, compute estimate $\hat{\mathcal{E}}_\square^0(f, \pi^t, h^t)$ via [Eq. (2)](#) (Q-type) or [Eq. (4)](#) (V-type).
13:    Update $\mathcal{F}^{t+1} = \{f \in \mathcal{F}^t : |\hat{\mathcal{E}}_\square^0(f, \pi^t, h^t)| \leq \varepsilon_{\text{elim}}\}$.
14: **end for**
15: Save the collected tuples $\{(h^t, \pi^t, \mathcal{D}^t)\}_{t=0}^{T-1}$ for the offline phase.
    **Offline phase**, the reward function $R = (R_0, \ldots, R_{H-1})$ is revealed.
16: Construct $\mathcal{F}_{\text{off}}(R) = \mathcal{F} + R$, set $\Pi_{\text{est}}^t = \{\pi^t\}$ (Q-type) or $\Pi_{\text{est}}^t = \Pi_{\text{on}} := \{\pi_f : f \in \mathcal{Z}_{\text{on}}\}$ (i.e., the greedy policies induced by $\mathcal{Z}_{\text{on}}$) (V-type).
17: For each $t \in [T]$, $g \in \mathcal{F}_{\text{off}}(R)$, and $\pi \in \Pi_{\text{est}}^t$, compute estimate $\hat{\mathcal{E}}^R(g, \pi^t, \pi, h^t)$ via [Eq. (3)](#) (Q-type) or [Eq. (5)](#) (V-type).
18: Set $\mathcal{F}_{\text{sur}}(R) = \{g \in \mathcal{F}_{\text{off}}(R) : \forall t \in [T], \forall \pi \in \Pi_{\text{est}}^t, |\hat{\mathcal{E}}^R(g, \pi^t, \pi, h^t)| \leq \varepsilon_{\text{elim}}/2\}$.
19: Return policy $\hat{\pi} = \pi_{\hat{g}}$, where $\hat{g} = \arg\max_{g \in \mathcal{F}_{\text{sur}}(R)} V_g(x_0)$.

---

### 3.1 Q-type RFOLIVE

Our algorithm, reward-free OLIVE (RFOLIVE) described in [Algorithm 1](#), takes the function class $\mathcal{F}$, the accuracy parameter $\varepsilon$, and the failure probability $\delta$ as input. As we are in the reward-free setting, it operates in two phases: an online exploration phase where it collects a dataset without an explicit reward signal, and an offline phase where it computes a near-optimal policy after the reward function $R$ is revealed. Below, we describe the two phases and the intuition behind the algorithm design.

**Online exploration phase**    During the online phase, we first set elimination thresholds $\varepsilon_{\text{actv}}, \varepsilon_{\text{elim}}$ and sample sizes $n_{\text{actv}}, n_{\text{elim}}$ and construct the following function class $\mathcal{F}_{\text{on}}$ used in the online phase:

$$\mathcal{F}_{\text{on}} = \mathcal{F} - \mathcal{F} := \left\{(f_0 - f_0', \ldots, f_{H-1} - f_{H-1}') : f_h, f_h' \in \mathcal{F}_h, \forall h \in [H]\right\}.$$

---

[5]The Q/V-type algorithms differ in whether to use uniform actions during exploration, and the distinction is needed to handle different settings of interest (see [Appendix B](#) as well as [Table 1](#)).

The detailed specification of these parameters are deferred to Theorem 1 and Theorem 3. Subsequently, we simulate Q-type OLIVE with the function class $\mathcal{F}_{\mathrm{on}}$ using the zero reward function $R = \mathbf{0}$ and the specified parameters. Similar to OLIVE, we initialize $\mathcal{F}^0 = \mathcal{F}_{\mathrm{on}}$ and maintain a version space $\mathcal{F}^t \subseteq \mathcal{F}^{t-1} \subseteq \mathcal{F}_{\mathrm{on}}$ of surviving functions after each iteration. In each iteration, we first find the optimistic function $f^t \in \mathcal{F}^t$ (line 4) and set $\pi^t = \pi_{f^t}$. In line 5, we collect $n_{\mathrm{actv}}$ trajectories to estimate the Q-type average Bellman error $\tilde{\mathcal{E}}_{\mathrm{Q}}^{\mathbf{0}}(f^t, \pi^t, h) = \tilde{\mathcal{E}}^{\mathbf{0}}(f^t, \pi^t, \pi^t, h)$ under zero reward:

$$\tilde{\mathcal{E}}^{\mathbf{0}}(f^t, \pi^t, \pi^t, h) = \frac{1}{n_{\mathrm{actv}}} \sum_{i=1}^{n_{\mathrm{actv}}} \left[ f_h^t \left( x_h^{(i)}, a_h^{(i)} \right) - V_{f^t} \left( x_{h+1}^{(i)} \right) \right]. \tag{1}$$

If the low average Bellman error condition in line 6 is satisfied, then we terminate the online phase and otherwise, we pick a step $h^t$ where the estimate $\tilde{\mathcal{E}}^{\mathbf{0}}(f^t, \pi^t, \pi^t, h^t) > \varepsilon_{\mathrm{actv}}$ (line 9). Then we collect $n_{\mathrm{elim}}$ trajectories using $a_{0:h^t} \sim \pi^t$ and set $\mathcal{D}^t$ as the transition tuples at step $h^t$. Using $\mathcal{D}^t$, we construct the Q-type average Bellman error estimates $\hat{\mathcal{E}}_{\mathrm{Q}}^{\mathbf{0}}(f, \pi^t, h^t)$ for all $f \in \mathcal{F}^t$ in line 12:

$$\hat{\mathcal{E}}_{\mathrm{Q}}^{\mathbf{0}}(f, \pi^t, h^t) = \frac{1}{n_{\mathrm{elim}}} \sum_{i=1}^{n_{\mathrm{elim}}} \left[ f_{h^t} \left( x_{h^t}^{(i)}, a_{h^t}^{(i)} \right) - V_f \left( x_{h^t+1}^{(i)} \right) \right]. \tag{2}$$

Finally, in line 13, we eliminate all the $f \in \mathcal{F}^t$ whose average Bellman error estimate $\hat{\mathcal{E}}_{\mathrm{Q}}^{\mathbf{0}}(f, \pi^t, h^t) > \varepsilon_{\mathrm{elim}}$.

The online phase returns tuples $\{(h^t, \pi^t, \mathcal{D}^t)\}_{t=0}^{T-1}$ where $T$ is the total number of iterations and each dataset $\mathcal{D}^t$ consists of $n_{\mathrm{elim}}$ transition tuples.

**Offline elimination phase** In the offline phase, the reward function $R$ is revealed, and we first construct the reward-appended function class $\mathcal{F}_{\mathrm{off}}(R) = \mathcal{F} + R := \{(f_0 + R_0, \ldots, f_{H-1} + R_{H-1}) : f_h \in \mathcal{F}_h, \forall h \in [H]\}$. Using the class $\Pi_{\mathrm{est}}^t = \{\pi^t\}$ from line 16 and the collected tuples $\{(h^t, \pi^t, \mathcal{D}^t)\}_{t=0}^{T-1}$, we estimate the reward-dependent average Bellman error (Definition 4) for all iterations $t \in [T]$ of the online phase:

$$\hat{\mathcal{E}}^R(g, \pi^t, \pi^t, h^t) = \frac{1}{n_{\mathrm{elim}}} \sum_{i=1}^{n_{\mathrm{elim}}} \left[ g_{h^t} \left( x_{h^t}^{(i)}, a_{h^t}^{(i)} \right) - R_{h^t} \left( x_{h^t}^{(i)}, a_{h^t}^{(i)} \right) - V_g \left( x_{h^t+1}^{(i)} \right) \right]. \tag{3}$$

RFOLIVE eliminates all $g \in \mathcal{F}_{\mathrm{off}}(R)$ whose average Bellman error estimates are large (line 18) and returns the optimistic function $\hat{g}$ from the surviving set (line 19).

**Remark** Similar to its counterparts in reward-aware general function approximation setting (Jiang et al., 2017; Dann et al., 2018; Jin et al., 2021; Du et al., 2021), RFOLIVE is in general not computationally efficient. We leave addressing computational tractability as a future direction.

### 3.1.1 Main results for Q-type RFOLIVE

In this part, we present the theoretical guarantee of Q-type RFOLIVE. We start with introducing the Q-type Bellman Eluder (BE) dimension (Jin et al., 2021).

**Definition 5** (Q-type BE dimension). *Let $(I - \mathcal{T}_h^R)\mathcal{F} := \{f_h - \mathcal{T}_h^R f_{h+1} : f \in \mathcal{F}\}$ be the set of Bellman differences of $\mathcal{F}$ at step $h$, and $\Gamma = \{\Gamma_h\}_{h=0}^{H-1}$ where $\Gamma_h$ is a set of distributions over $\mathcal{X} \times \mathcal{A}$. The $\varepsilon$-BE dimension of $\mathcal{F}$ w.r.t. $\Gamma$ is defined as $\dim_{\mathrm{qbe}}^R(\mathcal{F}, \Gamma, \varepsilon) := \max_{h \in [H]} d_{\mathrm{de}}\left((I - \mathcal{T}_h^R)\mathcal{F}, \Gamma_h, \varepsilon\right)$.*

We can now state our sample complexity result for Q-type RFOLIVE. To simplify presentation, we state the result here assuming parametric growth of the covering numbers, that is $\log(\mathcal{N}_{\mathcal{F}}(\varepsilon)) \leq d_{\mathcal{F}} \log(1/\varepsilon)$ and $\log(\mathcal{N}_{\mathcal{R}}(\varepsilon)) \leq d_{\mathcal{R}} \log(1/\varepsilon)$.

**Theorem 1** (Q-type RFOLIVE, parametric case). *Fix $\delta \in (0, 1)$. Given a reward class $\mathcal{R}$ and a function class $\mathcal{F}$ that satisfies Assumption 1 and Assumption 2, with probability at least $1 - \delta$, for any $R \in \mathcal{R}$, Q-type RFOLIVE (Algorithm 1) outputs a policy $\hat{\pi}$ that satisfies $v_R^{\hat{\pi}} \geq v_R^* - \varepsilon$. The required number of episodes is[6]*

$$\tilde{O}\left(\left(H^7 d_{\mathcal{F}} + H^5 d_{\mathcal{R}}\right) d_{\mathrm{qbe}}^2 \log(1/\delta)/\varepsilon^2\right),$$

*where $d_{\mathrm{qbe}} = \dim_{\mathrm{qbe}}^{\mathbf{0}}(\mathcal{F} - \mathcal{F}, \mathcal{D}_{\mathcal{F}-\mathcal{F}}, \varepsilon/(4H))$.*

---

[6]The $\tilde{O}(\cdot)$ notation suppresses poly-logarithmic factors in its argument.

The more general statement along with the specific values of $\varepsilon_{\mathrm{actv}}, \varepsilon_{\mathrm{elim}}, n_{\mathrm{actv}}, n_{\mathrm{elim}}$ are deferred to Appendix C.2, where we also present the proof. We remark that we only need the covering number of $\mathcal{R}$ to set these parameters and do not use any other information about the reward class.

We pause to compare Theorem 1 to the reward-aware case. First, our BE dimension involves the "difference" function class $\mathcal{F} - \mathcal{F}$ under zero reward as opposed to the original class with the given reward, and our completeness assumption is also related to such a "difference" function class. As we will see, these differences are inconsequential for our examples of interest. Second, our sample complexity has an additional $H^4$ dependence because (a) we consider a different reward normalization from Jiang et al. (2017); Jin et al. (2021) and (b) we use a smaller threshold in the online phase to ensure sufficient exploration. Similar gaps in $H$ factors between reward-free and reward-aware learning also appear in Wagenmaker et al. (2022). We also pay for the complexity of $\mathcal{R}$ in a lower order term, which is standard in reward-free RL (Zhang et al., 2020a; Modi et al., 2021). We believe that a similar adaptation of GOLF (Jin et al., 2021) for the reward-free setting may provide a sharper result with improved dependence on $H$ and $d_{\mathrm{qbe}}$, analogously to the reward-aware setting.

### 3.1.2 Q-type RFOLIVE for known representation linear completeness setting

Here, we instantiate the general guarantee of Q-type RFOLIVE to the linear completeness setting.[7]

**Definition 6** (Linear completeness setting (Zanette et al., 2020b)). *We call feature* $\phi^{\mathrm{lc}} = (\phi_0^{\mathrm{lc}}, \ldots, \phi_{H-1}^{\mathrm{lc}})$ *with* $\phi_h^{\mathrm{lc}} : \mathcal{X} \times \mathcal{A} \to \mathbb{R}^{d_{\mathrm{lc}}}, \|\phi_h^{\mathrm{lc}}(\cdot)\|_2 \leq 1, \forall h \in [H]$ *a linearly complete feature, if for any* $B > 0, h \in [H-1]$ *and* $\forall f_{h+1} \in \mathcal{Q}_{h+1}(\{\phi^{\mathrm{lc}}\}, B)$ *we have:* $\min_{f_h \in \mathcal{Q}_h(\{\phi^{\mathrm{lc}}\}, B)} \|f_h - \mathcal{T}_h^{\mathbf{0}} f_{h+1}\|_\infty = 0$, *where* $\mathcal{Q}_h(\{\phi^{\mathrm{lc}}\}, B) = \{\langle \phi_h^{\mathrm{lc}}, \theta_h \rangle : \|\theta_h\|_2 \leq B\sqrt{d_{\mathrm{lc}}}\}$.

When the linearly complete features (Definition 6) $\phi^{\mathrm{lc}}$ are known, we can construct the function class $\mathcal{F}(\{\phi^{\mathrm{lc}}\}) = \mathcal{F}_0(\{\phi^{\mathrm{lc}}\}, H-1) \times \ldots \times \mathcal{F}_{H-1}(\{\phi^{\mathrm{lc}}\}, 0)$, where $\mathcal{F}_h(\{\phi^{\mathrm{lc}}\}, B_h) = \{f_h(x_h, a_h) = \langle \phi_h^{\mathrm{lc}}(x_h, a_h), \theta_h \rangle : \|\theta_h\|_2 \leq B_h\sqrt{d_{\mathrm{lc}}}, \langle \phi_h^{\mathrm{lc}}(\cdot), \theta_h \rangle \in [-B_h, B_h]\}$ consists of appropriately bounded linear functions of $\phi^{\mathrm{lc}}$. Here superscript and subscript lc imply that the notations are related to the linear completeness setting. It is easy to verify that $\mathcal{F}(\{\phi^{\mathrm{lc}}\})$ satisfies the assumptions in Theorem 1. This gives us the following corollary (see the full statement and the proof in Appendix C.4):

**Corollary 2** (Informal). *Fix* $\delta \in (0,1)$. *Consider an MDP* $M$ *that satisfies linear completeness (Definition 6) with known feature* $\phi^{\mathrm{lc}}$, *and the linear reward class* $\mathcal{R} = \mathcal{R}_1 \times \ldots \times \mathcal{R}_h$, *where* $\mathcal{R}_h = \{\langle \phi_h^{\mathrm{lc}}, \eta_h \rangle : \|\eta_h\|_2 \leq \sqrt{d_{\mathrm{lc}}}, \langle \phi_h^{\mathrm{lc}}(\cdot), \eta_h \rangle \in [0,1]\}$. *With probability at least* $1-\delta$, *for any* $R \in \mathcal{R}$, *Q-type* RFOLIVE *(Algorithm 1) with* $\mathcal{F} = \mathcal{F}(\{\phi^{\mathrm{lc}}\})$ *outputs a policy* $\hat{\pi}$ *that satisfies* $v_R^{\hat{\pi}} \geq v_R^* - \varepsilon$. *The required number of samples is* $\tilde{O}\left(H^8 d_{\mathrm{lc}}^3 \log(1/\delta)/\varepsilon^2\right)$.

The reward normalization above, called *explicit regularity* in Zanette et al. (2020b), is standard. Compared to that work, our result implies that *explorability* is not necessary, which significantly relaxes the existing assumptions for this setting. Our result can also be easily extended to handle approximately linearly complete features (i.e., low inherent Bellman error). On the other hand, our algorithm is not computationally efficient owing to our general function approximation setting. Although our sample complexity bound *appears* to be worse in $H$ factors compared with their upper bound of $\tilde{O}\left(d_{\mathrm{lr}}^3 H^5 \log(1/\delta)/\varepsilon^2\right)$, it is indeed incomparable because their bound only holds when $\varepsilon \leq \tilde{O}(\nu_{\min}/\sqrt{d_{\mathrm{lc}}})$ ($\nu_{\min}$ is their explorability factor). Thus, there is an implicit dependence on $1/\nu_{\min}$ in their result, which could make the bound arbitrarily worse than ours. More discussions are deferred to Appendix A and Appendix C.4.

### 3.2 V-type RFOLIVE

In this section, we describe the instantiation of RFOLIVE with V-type definitions. For V-type RFOLIVE, we also assume that the action space is finite with size $K$.

**Online exploration phase** Instead of using $\mathcal{F}_{\mathrm{on}}$, we use its $(\varepsilon_{\mathrm{elim}}/64)$-cover $\mathcal{Z}_{\mathrm{on}}$ and maintain a version space $\mathcal{F}^t$ across iterations.[8] Since the on-policy version of Q-type and V-type Bellman

---

[7]Zanette et al. (2020b) only define linear completeness for $B = 1$. It can be easily verified that it is equivalent for any choice of $B$. More discussion can be found in Appendix C.4.

[8]Following Jin et al. (2021), we run V-type OLIVE with the discretized class $\mathcal{Z}_{\mathrm{on}}$ for the ease of presentation.

errors are the same, the termination check in line 5 and line 6 are unchanged. If the algorithm does not terminate in line 6, we again identify a deviation step $h^t$ such that $\tilde{\mathcal{E}}_V^0(f^t, \pi^t, h^t) = \tilde{\mathcal{E}}^0(f^t, \pi^t, \pi^t, h^t) > \varepsilon_{\text{actv}}$. Instead of using $\pi^t$ to collect trajectories, we use $a_{0:h^t-1} \sim \pi^t$ and choose $a_{h^t}$ uniformly at random to collect the dataset of $n_{\text{elim}}$ transition tuples at step $h^t$. Compared to Q-type RFOLIVE, we estimate $\hat{\mathcal{E}}_V^0$ for all $f \in \mathcal{F}^t$ in line 12 using importance sampling (IS):

$$\hat{\mathcal{E}}_V^0(f, \pi^t, h^t) = \frac{1}{n_{\text{elim}}} \sum_{i=1}^{n_{\text{elim}}} \frac{\mathbf{1}[a_{h^t}^{(i)} = \pi_f(x_{h^t}^{(i)})]}{1/K} \left[ f_{h^t}\left(x_{h^t}^{(i)}, a_{h^t}^{(i)}\right) - V_f\left(x_{h^t+1}^{(i)}\right) \right]. \tag{4}$$

Finally, in line 13, we eliminate all $f \in \mathcal{F}^t$ whose V-type average Bellman error estimates are large.

**Offline elimination phase**  In the offline phase, we consider the same reward-appended function class $\mathcal{F}_{\text{off}}(R)$ when reward $R \in \mathcal{R}$ is revealed. For V-type RFOLIVE, in line 16, we define the policy class $\Pi_{\text{est}}^t = \Pi_{\text{on}}$ which consists of greedy policies with respect to all $f \in \mathcal{Z}_{\text{on}}$. Using dataset $\mathcal{D}^t$, we estimate $\mathcal{E}^R(g, \pi^t, \pi', h^t)$ for all $g \in \mathcal{F}_{\text{off}}(R), \pi' \in \Pi_{\text{on}}, t \in [T]$ from its empirical version:

$$\hat{\mathcal{E}}^R(g, \pi^t, \pi', h^t) = \frac{1}{n_{\text{elim}}} \sum_{i=1}^{n_{\text{elim}}} \frac{\mathbf{1}[a_{h^t}^{(i)} = \pi'(x_{h^t}^{(i)})]}{1/K} \left[ g_h(x_h^{(i)}, a_h^{(i)}) - R_h(x_h^{(i)}, a_h^{(i)}) - V_g(x_{h+1}^{(i)}) \right] \tag{5}$$

and eliminate invalid functions in line 18. Finally, we return the optimistic policy $\hat{\pi}$ from the surviving set. Apart from estimating different average Bellman errors, the noticeable difference between Q-type and V-type RFOLIVE is that the latter uses IS to correct the uniformly drawn action to some policy $\pi' \in \Pi_{\text{on}}$ to witness the average Bellman error (Jiang et al., 2017).

### 3.2.1  Main results for V-type RFOLIVE

Here we present the theoretical guarantee of V-type RFOLIVE. Firstly, we introduce the V-type Bellman Eluder (BE) dimension (Jin et al., 2021).

**Definition 7** (V-type BE dimension). *Let $(I - \mathcal{T}_h^R)V_{\mathcal{F}} \subseteq (\mathcal{X} \to \mathbb{R})$ be the state-wise Bellman difference class of $\mathcal{F}$ at step $h$ defined as $(I - \mathcal{T}_h^R)V_{\mathcal{F}} := \{x \mapsto (f_h - \mathcal{T}_h^R f_{h+1})(x, \pi_{f_h}(x)) : f \in \mathcal{F}\}$. Let $\Gamma = \{\Gamma_h\}_{h=0}^{H-1}$ where $\Gamma_h$ is a set of distributions over $\mathcal{X}$. The V-type $\varepsilon$-BE dimension of $\mathcal{F}$ with respect to $\Gamma$ is defined as $\dim_{\text{vbe}}^R(\mathcal{F}, \Gamma, \varepsilon) := \max_{h \in [H]} d_{\text{de}}\left((I - \mathcal{T}_h^R)V_{\mathcal{F}}, \Gamma_h, \varepsilon\right)$.*

We now state the guarantee for V-type RFOLIVE, assuming polynomial covering number growth.

**Theorem 3** (V-type RFOLIVE, parametric case). *Fix $\delta \in (0, 1)$. Given a reward class $\mathcal{R}$, a function class $\mathcal{F}$ that satisfies Assumption 1, Assumption 2, with probability at least $1 - \delta$, for any $R \in \mathcal{R}$, V-type RFOLIVE outputs a policy $\hat{\pi}$ that satisfies $v_R^{\hat{\pi}} \geq v_R^* - \varepsilon$. The required number of episodes is*

$$\tilde{O}\left(\left(H^7 d_{\mathcal{F}} + H^5 d_{\mathcal{R}}\right) d_{\text{vbe}}^2 K \log(1/\delta)/\varepsilon^2\right),$$

*where $d_{\text{vbe}} = \dim_{\text{vbe}}^0(\mathcal{F} - \mathcal{F}, \mathcal{D}_{\mathcal{F}-\mathcal{F}}, \varepsilon/(8H))$.*

The detailed proof and the specific values of $\varepsilon_{\text{actv}}, \varepsilon_{\text{elim}}, n_{\text{actv}}, n_{\text{elim}}$ are deferred to Appendix D.2. Our rate is again loose in $H$ factors when compared with the reward-aware version. Compared with the Q-type version, here we also incur a dependence on $K = |\mathcal{A}|$, analogous to the reward-aware case.

### 3.2.2  V-type RFOLIVE for unknown representation low-rank MDPs

As a special case, we instantiate our V-type RFOLIVE result to low-rank MDPs (Modi et al., 2021):

**Definition 8** (Low-rank factorization). *A transition operator $P_h : \mathcal{X} \times \mathcal{A} \to \Delta(\mathcal{X})$ admits a low-rank decomposition of dimension $d_{\text{lr}}$ if there exists $\phi_h^{\text{lr}} : \mathcal{X} \times \mathcal{A} \to \mathbb{R}^{d_{\text{lr}}}$ and $\mu_h^{\text{lr}} : \mathcal{X} \to \mathbb{R}^{d_{\text{lr}}}$ s.t. $\forall x, x' \in \mathcal{X}, a \in \mathcal{A} : P_h(x' \mid x, a) = \langle \phi_h^{\text{lr}}(x, a), \mu_h^{\text{lr}}(x') \rangle$, and additionally $\|\phi_h^{\text{lr}}(\cdot)\|_2 \leq 1$ and $\forall f' : \mathcal{X} \to [-1, 1]$, we have $\left\| \int f'(x) \mu_h^{\text{lr}}(x) dx \right\|_2 \leq \sqrt{d_{\text{lr}}}$. We say $M$ is low-rank with embedding dimension $d_{\text{lr}}$, if for each $h \in [H]$, the transition operator $P_h$ admits a rank-$d_{\text{lr}}$ decomposition.*

Here superscript and subscript lr imply that the notations are related to low-rank MDPs. As in Modi et al. (2021), we consider low-rank MDPs in a representation learning setting, where we are given realizable feature class $\Phi^{\text{lr}}$ rather than the feature $\phi^{\text{lr}} = (\phi_0^{\text{lr}}, \ldots, \phi_{H-1}^{\text{lr}})$ directly:

**Assumption 3** (Realizability of low-rank feature class). *We assume that a finite feature class* $\Phi^{\mathrm{lr}} = \Phi_0^{\mathrm{lr}} \times \ldots \times \Phi_{H-1}^{\mathrm{lr}}$ *satisfies* $\phi_h^{\mathrm{lr}} \in \Phi_h^{\mathrm{lr}}, \forall h \in [H]$. *In addition,* $\forall h \in [H], \phi_h \in \Phi_h^{\mathrm{lr}}$, $\|\phi_h(\cdot)\|_2 \leq 1$.

Similar to the linear completeness setting (Section 3.1.2), we construct $\mathcal{F}(\Phi^{\mathrm{lr}}) = \mathcal{F}_0(\Phi^{\mathrm{lr}}, H - 1) \times \ldots \times \mathcal{F}_{H-1}(\Phi^{\mathrm{lr}}, 0)$, where $\mathcal{F}_h(\Phi^{\mathrm{lr}}, B_h) = \{f_h(x_h, a_h) = \langle \phi_h(x_h, a_h), \theta_h \rangle : \phi_h \in \Phi_h^{\mathrm{lr}}, \|\theta_h\|_2 \leq B_h\sqrt{d_{\mathrm{lr}}}, \langle \phi_h(\cdot), \theta_h \rangle \in [-B_h, B_h]\}$. In Proposition 4, we show that the V-type Bellman Eluder dimension of $\mathcal{F}(\Phi^{\mathrm{lr}}) - \mathcal{F}(\Phi^{\mathrm{lr}})$ in this case is $\tilde{O}(d_{\mathrm{lr}})$ which leads to the following corollary:

**Corollary 4** (Informal, parametric case). *Fix* $\delta \in (0, 1)$. *Consider a low-rank MDP $M$ of embedding dimension $d_{\mathrm{lr}}$ with a realizable feature class $\Phi^{\mathrm{lr}}$ (Assumption 3) and a reward class $\mathcal{R}$. With probability at least $1 - \delta$, for any $R \in \mathcal{R}$, V-type* RFOLIVE *(Algorithm 1) with $\mathcal{F}(\Phi^{\mathrm{lr}})$ outputs a policy $\hat{\pi}$ that satisfies $v_R^{\hat{\pi}} \geq v_R^* - \varepsilon$. The required number of episodes is*

$$\tilde{O}\left(\left(H^8 d_{\mathrm{lr}}^3 \log(|\Phi^{\mathrm{lr}}|) + H^5 d_{\mathrm{lr}}^2 d_{\mathcal{R}}\right) K \log(1/\delta)/\varepsilon^2\right).$$

We defer the full statement and detailed proof of the corollary to Appendix D.3. In the low-rank MDP setting, Modi et al. (2021) propose a more computationally viable algorithm, but additionally require a reachability assumption. Our result shows that reachability is not necessary for statistically efficiency, which opens an interesting avenue for designing an algorithm that is both computationally and statistically efficient without reachability. Moreover, our result significantly improves upon their sample complexity bound. The detailed comparisons are deferred to Appendix A and Appendix D.3. Notice that here $K$ shows up in our bound. As another corollary, in the linear MDP (Definition 8 plus $\phi^{\mathrm{lr}}$ is known), Q-type RFOLIVE yields a $K$ independent bound. The details can be found in Appendix C.5.

### 3.3 Intuition and proof sketch for RFOLIVE

We first provide the intuition. Since the online phase of RFOLIVE is equivalent to running OLIVE with $\mathbf{0}$ reward function, any policy $\pi_f$ attains zero value (i.e., $V_{\mathbf{0},0}^{\pi_f}(x_0) = 0, \forall f \in \mathcal{F}_{\mathrm{on}}$). By the policy loss decomposition lemma (Jiang et al., 2017), the value error for the greedy policy, $V_f(x_0) - V_{\mathbf{0},0}^{\pi_f}(x_0)$, is small when the algorithm stops (line 6). Therefore, all $f \in \mathcal{F}_{\mathrm{on}}$ which predict large values $V_f(x_0)$ must have been eliminated before OLIVE terminates. This implies that, in the online phase, we gather a diverse set of constraints (roll-in distributions $\pi^t$) that can witness the average Bellman error of functions in $\mathcal{F}_{\mathrm{on}}$. In this sense, our algorithm focuses on function space elimination and does not try to reach all latent states or directions (Modi et al., 2021; Zanette et al., 2020b), which is the key conceptual difference that enables us to avoid reachability and explorability assumptions.

On the technical side, note that the way we use OLIVE in the offline setting is novel to our knowledge and is crucial to getting a good sample complexity under our assumptions, as opposed to more standard FQI style approaches. Because we have to coordinate between the online and offline phases, the analysis bears significant novelty beyond the original analysis of OLIVE (and its reward-aware follow-up works), and this is one of our key contributions. The most crucial part is to show that any bad $g \in \mathcal{F}_{\mathrm{off}}(R)$ whose average Bellman error is large under the true reward $R$ will be eliminated in the offline phase. To prove this, we construct $\tilde{f} \in \mathcal{F}_{\mathrm{on}}$ that has the same average Bellman error as $g$ and predicts a large positive value $V_{\tilde{f}}(x_0)$, which implies that it will be eliminated during the online phase. Finally, by our construction, the constraint used to eliminate $\tilde{f}$ directly witnesses the average Bellman error of $g$, thus ruling out $g$ in the offline phase. We discuss it in more detail in Appendix C.3.

## 4 Hardness result for unknown representation linear completeness setting

In Section 3.1.2, we showed that Q-type RFOLIVE requires polynomial sample for reward-free RL in the known feature linear completeness setting. For low-rank MDPs, when given a realizable feature class, we showed V-type RFOLIVE is statistically efficient in Section 3.2.2. A natural next step is to relax the low-rank assumption on the MDP and show a sample efficiency result for the more general linear completeness and unknown feature case. However, below we state a hardness result which shows that a polynomial dependence on the feature class ($|\Phi^{\mathrm{lc}}|$) or an exponential dependence on $H$ is *unavoidable*. We first introduce the realizability of a linearly complete feature class.

**Assumption 4** (Realizability of the linearly complete feature class). *We assume that there exists a finite candidate feature class* $\Phi^{\mathrm{lc}} = \Phi_0^{\mathrm{lc}} \times \cdots \times \Phi_{H-1}^{\mathrm{lc}}$, *such that* $\forall h \in [H]$, *we have* $\phi_h^{\mathrm{lc}} \in \Phi_h^{\mathrm{lc}}$. *In addition,* $\forall h \in [H], \phi_h \in \Phi_h^{\mathrm{lc}}, \forall (x, a) \in \mathcal{X} \times \mathcal{A}$, *we have* $\|\phi_h(\cdot)\|_2 \leq 1$.

Now we state of hardness result for learning in the linear completeness setting with a realizable feature class (Assumption 4). A complete proof and more discussions are provided in Appendix E.

**Theorem 5.** *There exists a family of MDPs* $\mathcal{M}$, *a reward class* $\mathcal{R}$ *and a feature set* $\Phi^{\mathrm{lc}}$, *such that* $\forall M \in \mathcal{M}$, *the* $(M, \Phi^{\mathrm{lc}})$ *pair satisfies Assumption 4, yet it is information-theoretically impossible for an algorithm to obtain a* $\mathrm{poly}\left(d_{\mathrm{lc}}, H, \log(|\Phi^{\mathrm{lc}}|), \log(|\mathcal{R}|), 1/\varepsilon, \log(1/\delta)\right)$ *sample complexity for reward-free exploration with the given reward class* $\mathcal{R}$.

The hardness result in Theorem 5 is also applicable to easier settings: (i) learning with a generative model (or using a local access protocol, Hao et al. (2022)), (ii) reward-free learning with explorability (Zanette et al., 2020b) and reachability (Modi et al., 2021) assumptions and (iii) reward-aware learning as $\mathcal{R}$ is a known singleton class. Thus, the result highlights an exponential separation between the low-rank MDP and linear completeness assumptions by showing that linearly complete true feature $\phi^{\mathrm{lc}} \in \Phi^{\mathrm{lc}}$ is not sufficient for polynomial sample efficiency and additional assumptions are required to account for the unknown representation.

# 5  Conclusion and discussion

In this paper, we investigated the statistical efficiency of reward-free RL under general function approximation. The proposed algorithm, RFOLIVE, is the first algorithm to address reward-free exploration under general function approximation. Contrary to prior works which either try to reach all states or all directions in the feature space, RFOLIVE follows a value function elimination template and ensures that the collected exploration data can be used to identify and eliminate non-optimal value functions for downstream planning. This significantly sets us apart from the existing reward-free exploration works. Our positive results significantly relax the existing assumptions in the reward-free exploration framework. Our negative result shows the first sharp separation between low-rank MDP and the linear completeness settings with unknown representations. In addition, we provide an algorithm-specific counterexample in Appendix F that shows RFOLIVE can fail when the completeness assumption is violated. As realizability alone is sufficient for reward-aware RL (Jiang et al., 2017; Jin et al., 2021; Du et al., 2021), our results also elicit the further question:

**Are realizability-type assumptions sufficient for statistically efficient reward-free RL?**

We conjecture that the answer is no, and we believe that the hardness between reward-aware and reward-free RL has a deep connection to the sharp separation between realizability and completeness (Chen and Jiang, 2019; Wang et al., 2020b, 2021; Xie and Jiang, 2021; Weisz et al., 2021a,b, 2022; Foster et al., 2021).

# Acknowledgements

JC would like to thank Tengyang Xie for helpful discussions. Part of this work was done while AM was at University of Michigan and was supported in part by a grant from the Open Philanthropy Project to the Center for Human-Compatible AI, and in part by NSF grant CAREER IIS-1452099. NJ acknowledges funding support from ARL Cooperative Agreement W911NF-17-2-0196, NSF IIS-2112471, NSF CAREER IIS-2141781, and Adobe Data Science Research Award.

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
