}}} \frac{\mathbf{1}[a^{(i)}_{h^t} = \pi_f(x^{(i)}_{h^t})]}{1/K} \left[ f_{h^t}\left(x^{(i)}_{h^t}, a^{(i)}_{h^t}\right) - V_f\left(x^{(i)}_{h^t+1}\right) \right]. \tag{4}$$

Finally, in line 13, we eliminate all $f \in \mathcal{F}^t$ whose V-type average Bellman error estimates are large.

**Offline elimination phase**   In the offline phase, we consider the same reward-appended function class $\mathcal{F}_{\mathrm{off}}(R)$ when reward $R \in \mathcal{R}$ is revealed. For V-type RFOLIVE, in line 16, we define the policy class $\Pi^t_{\mathrm{est}} = \Pi_{\mathrm{on}}$ which consists of greedy policies with respect to all $f \in \mathcal{Z}_{\mathrm{on}}$. Using dataset $\mathcal{D}^t$, we estimate $\mathcal{E}^R(g, \pi^t, \pi', h^t)$ for all $g \in \mathcal{F}_{\mathrm{off}}(R), \pi' \in \Pi_{\mathrm{on}}, t \in [T]$ from its empirical version:

$$\hat{\mathcal{E}}^R(g, \pi^t, \pi', h^t) = \frac{1}{n_{\mathrm{elim}}} \sum_{i=1}^{n_{\mathrm{elim}}} \frac{\mathbf{1}[a^{(i)}_{h^t} = \pi'(x^{(i)}_{h^t})]}{1/K} \left[ g_h(x^{(i)}_h, a^{(i)}_h) - R_h(x^{(i)}_h, a^{(i)}_h) - V_g(x^{(i)}_{h+1}) \right] \tag{5}$$

and eliminate invalid functions in line 18. Finally, we return the optimistic policy $\hat{\pi}$ from the surviving set. Apart from estimating different average Bellman errors, the noticeable difference between Q-type and V-type RFOLIVE is that the latter uses IS to correct the uniformly drawn action to some policy $\pi' \in \Pi_{\mathrm{on}}$ to witness the average Bellman error (Jiang et al., 2017).

### 3.2.1   Main results for V-type RFOLIVE

Here we present the theoretical guarantee of V-type RFOLIVE. Firstly, we introduce the V-type Bellman Eluder (BE) dimension (Jin et al., 2021).

**Definition 7** (V-type BE dimension). *Let* $(I - \mathcal{T}^R_h)V_{\mathcal{F}} \subseteq (\mathcal{X} \to \mathbb{R})$ *be the state-wise Bellman difference class of $\mathcal{F}$ at step $h$ defined as* $(I - \mathcal{T}^R_h)V_{\mathcal{F}} := \left\{ x \mapsto (f_h - \mathcal{T}^R_h f_{h+1})(x, \pi_{f_h}(x)) : f \in \mathcal{F} \right\}$. *Let* $\Gamma = \{\Gamma_h\}^{H-1}_{h=0}$ *where $\Gamma_h$ is a set of distributions over $\mathcal{X}$. The V-type $\varepsilon$-BE dimension of $\mathcal{F}$ with respect to $\Gamma$ is defined as* $\dim^R_{\mathrm{vbe}}(\mathcal{F}, \Gamma, \varepsilon) := \max_{h \in [H]} d_{\mathrm{de}}\left((I - \mathcal{T}^R_h)V_{\mathcal{F}}, \Gamma_h, \varepsilon\right)$.

We now state the guarantee for V-type RFOLIVE, assuming polynomial covering number growth.

**Theorem 3** (V-type RFOLIVE, parametric case). *Fix $\delta \in (0, 1)$. Given a reward class $\mathcal{R}$, a function class $\mathcal{F}$ that satisfies Assumption 1, Assumption 2, with probability at least $1 - \delta$, for any $R \in \mathcal{R}$, V-type RFOLIVE outputs a policy $\hat{\pi}$ that satisfies $v^{\hat{\pi}}_R \geq v^*_R - \varepsilon$. The required number of episodes is*

$$\tilde{O}\left(\left(H^7 d_{\mathcal{F}} + H^5 d_{\mathcal{R}}\right) d^2_{\mathrm{vbe}} K \log(1/\delta)/\varepsilon^2\right),$$

*where* $d_{\mathrm{vbe}} = \dim^{\mathbf{0}}_{\mathrm{vbe}}(\mathcal{F} - \mathcal{F}, \mathcal{D}_{\mathcal{F}-\mathcal{F}}, \varepsilon/(8H))$.

The detailed proof and the specific values of $\varepsilon_{\mathrm{actv}}, \varepsilon_{\mathrm{elim}}, n_{\mathrm{actv}}, n_{\mathrm{elim}}$ are deferred to Appendix D.2. Our rate is again loose in $H$ factors when compared with the reward-aware version. Compared with the Q-type version, here we also incur a dependence on $K = |\mathcal{A}|$, analogous to the reward-aware case.

### 3.2.2   V-type RFOLIVE for unknown representation low-rank MDPs

As a special case, we instantiate our V-type RFOLIVE result to low-rank MDPs (Modi et al., 2021):

**Definition 8** (Low-rank factorization). *A transition operator $P_h : \mathcal{X} \times \mathcal{A} \to \Delta(\mathcal{X})$ admits a low-rank decomposition of dimension $d_{\mathrm{lr}}$ if there exists $\phi^{\mathrm{lr}}_h : \mathcal{X} \times \mathcal{A} \to \mathbb{R}^{d_{\mathrm{lr}}}$ and $\mu^{\mathrm{lr}}_h : \mathcal{X} \to \mathbb{R}^{d_{\mathrm{lr}}}$ s.t. $\forall x, x' \in \mathcal{X}, a \in \mathcal{A} : P_h(x' \mid x, a) = \langle \phi^{\mathrm{lr}}_h(x, a), \mu^{\mathrm{lr}}_h(x') \rangle$, and additionally $\|\phi^{\mathrm{lr}}_h(\cdot)\|_2 \leq 1$ and $\forall f' : \mathcal{X} \to [-1, 1]$, we have $\left\|\int f'(x)\mu^{\mathrm{lr}}_h(x)dx\right\|_2 \leq \sqrt{d_{\mathrm{lr}}}$. We say $M$ is low-rank with embedding dimension $d_{\mathrm{lr}}$, if for each $h \in [H]$, the transition operator $P_h$ admits a rank-$d_{\mathrm{lr}}$ decomposition.*

Here superscript and subscript lr imply that the notations are related to low-rank MDPs. As in Modi et al. (2021), we consider low-rank MDPs in a representation learning setting, where we are given realizable feature class $\Phi^{\mathrm{lr}}$ rather than the feature $\phi^{\mathrm{lr}} = (\phi^{\mathrm{lr}}_0, \ldots, \phi^{\mathrm{lr}}_{H-1})$

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

# A    Comparisons of sample complexity rates

In this section, we provide comparisons of sample complexity rates. Some more specific and detailed discussions can be found in Appendix C.4, Appendix C.5, and Appendix D.3 respectively. In Table 2, we transfer all bounds in related works into our notations and compare them with ours.

| Setting | Sample complexity |
|---|---|
| Linear MDP (Wagenmaker et al., 2022) | $\frac{d_{\mathrm{lr}}^2 H^5 \log(1/\delta)}{\varepsilon^2}$ |
| Linear MDP (**Corollary 6**) | $\frac{d_{\mathrm{lr}}^3 H^8 \log(1/\delta)}{\varepsilon^2}$ |
| Linear completeness + explorability (Zanette et al., 2020b) | $\frac{d_{\mathrm{lc}}^3 H^5 \log(1/\delta)}{\varepsilon^2}$ |
| Linear completeness (**Corollary 2**) | $\frac{d_{\mathrm{lc}}^3 H^8 \log(1/\delta)}{\varepsilon^2}$ |
| Low-rank MDP + small $|\mathcal{A}|$ + reachability (Modi et al., 2021) | $\frac{d_{\mathrm{lr}}^{11} H^7 K^{14} \log(|\Phi^{\mathrm{lr}}||\mathcal{R}|/\delta)}{\min\{\varepsilon^2 \eta_{\min}, \eta_{\min}^5\}}$ |
| Low-rank MDP + small $|\mathcal{A}|$ (**Corollary 4**) | $\frac{d_{\mathrm{lr}}^3 H^8 K \log(|\Phi^{\mathrm{lr}}||\mathcal{R}|/\delta)}{\varepsilon^2}$ |
| Completeness + Q-type BE dimension (**Theorem 1**) | $\frac{d_{\mathrm{qbe}}^2 (H^7 d_{\mathcal{F}} + H^5 d_{\mathcal{R}}) \log(1/\delta)}{\varepsilon^2}$ |
| Completeness + V-type BE dimension + small $|\mathcal{A}|$ (**Theorem 3**) | $\frac{d_{\mathrm{vbe}}^2 K (H^7 d_{\mathcal{F}} + H^5 d_{\mathcal{R}}) \log(1/\delta)}{\varepsilon^2}$ |

Table 2: Comparisons between our results and most closely related works in reward-free exploration. Red assumptions are what prior works need that are avoided by us. For simplicity, we only show the orders and hide polylog terms (i.e., using $\tilde{O}(\cdot)$ notation). $\eta_{\min}$ is the reachability factor in Modi et al. (2021).

In linear MDPs, our bound (Corollary 6) is $d_{\mathrm{lr}} H^3$ worse compared with the most recent work (Wagenmaker et al., 2022), but our result is also independent of $K$. It should be noted that both these bounds are sub-optimal in $H$ dependence when compared to the lower bound of $\Omega\left(d^2 H^2/\varepsilon^2\right)$ shown in Wagenmaker et al. (2022). In the reward-aware setting, GOLF has a sharper rate than the subroutine OLIVE under the completeness type assumption (Jin et al., 2021). Since in RFOLIVE we only collect data when running a single (zero) reward OLIVE during the online phase and completeness (Assumption 2) is satisfied in our paper, we believe that there also exists a reward-free version of GOLF (by running GOLF with zero reward function in the online phase and performing function elimination in the offline phase) that can potentially improve an $H d_{\mathrm{lr}}$ factor.

As for the linear completeness setting, our rate (Corollary 2) *appears* to be $H^3$ worse than Zanette et al. (2020b). However, we want to remark that they need to assume $\varepsilon$ to be "asymptotically small" (more specifically, $\varepsilon \leq \tilde{O}(\nu_{\min}/\sqrt{d_{\mathrm{lc}}})$, where $\nu_{\min}$ is their explorability factor). Thus there is an implicit dependence on $1/\nu_{\min}$ in their sample complexity bound. Since such a factor can be arbitrarily large while $H$ is always bounded in a finite horizon problem, our bound could be much better than theirs. Again, there could be an $H d_{\mathrm{lc}}$ tighter bound for the reward-free version of GOLF, which implies that the optimal $d_{\mathrm{lc}}$ dependence in the linear completeness setting could also be improved.

In low-rank MDPs, it is easy to see that our result (Corollary 4) significantly improves upon the rate of Modi et al. (2021) in $d_{\mathrm{lr}}$ and $K$ factors, while slightly worse in the $H$ factor. In addition, they require the reachability assumption ($\eta_{\min}$ is their reachability factor), which means that their bound can be arbitrary worse than ours. Similar dependence on reachability factor $1/\eta_{\min}$ also exists in the sample complexity bounds of the more restricted block MDPs (Du et al., 2019; Misra et al., 2020) as they assume the reachability assumption.

Finally, regarding lower bounds, we do not necessarily need a direct one in our general function approximation setting (or even the more restricted linear completeness setting/low-rank MDPs) to compare with. The lower bound for reward-free exploration in linear MDPs (Wagenmaker et al., 2022) is applicable to each of these and shows the necessary dependence on the respective complexity

measures. Coming up with a method which incorporates general function approximation while incurring better sample complexity rates on these special instances is a challenging and interesting avenue for future work.

## B  Discussions on Q-type and V-type

In this paper we study both Q-type and V-type, and they are not specific to the reward-free exploration. Different versions (Q-type and V-type) already exist in the reward-aware general function approximation RL (e.g., Jiang et al. (2017); Jin et al. (2021); Du et al. (2021)). They capture different scenarios of interest and so far, it seems difficult to unify them even in the reward-aware setting. Therefore, to give a comprehensive treatment of general function approximation, we consider both together. The algorithms and analyses for the two types are not very different, with only moderate differences.

Since we consider the BE dimension and it subsumes Bellman rank (Jin et al., 2021), we first provide a detailed comparison between Q-type and V-type Bellman rank. As discussed in Agarwal and Zhang (2022), V-type permits representation learning and other non-linear scenarios that are not easily captured in Q-type. For instance, any contextual bandit problem is admissible under the V-type assumption (the V-type Bellman rank is 1), while Q-type does not capture all finite action, non-linear contextual bandit problems with a realizable reward. We refer the reader to the detailed lower bound on the Q-type Bellman rank in the contextual bandit setting in Appendix B of Agarwal and Zhang (2022). In contrast, Q-type has a more linear like structure, but it also includes problems whose V-type Bellman rank is large (e.g., linear completeness setting in Zanette et al. (2020a)). Further, V-type RFOLIVE (or V-type OLIVE) requires one uniform action in exploration and therefore has an additional $K$ factor (the cardinality of action space) in the sample complexity bound.

Then we discuss the BE dimension. It can be shown that the Q-type BE dimension could also be exponentially larger than the V-type BE dimension. In Agarwal and Zhang (2022), the authors show that the Q-type Bellman rank for a contextual bandit instance can be made arbitrarily higher whereas the V-type Bellman rank is always 1 for a CB setting. Here, we show that the same instance can be shown to have high Bellman Eluder dimension as well. The construction considers a context distribution which is uniform on $1, \ldots, N$, where we have $N$ unique contexts. We have two actions $\{a_1, a_2\}$. We also have $|\mathcal{F}| = N + 1$ with the following structure:

$$f^*(x, a_1) = f_{N+1}(x, a_1) = 0$$
$$f^*(x, a_2) = f_{N+1}(x, a_2) = 0.5.$$

For $i < N + 1$, we have $f_i(x, a) = f^*(x, a)$ when $x \neq i$, and $f_i(x, a_1) = 1$, $f_i(x, a_2) = 0.5$ implying that the function $f_i$ makes incorrect prediction on context $i$ for action $a_1$. Now, the bound on V-type BE dimension can be obtained by using the Bellman rank to BE dimension conversion result in Proposition 21 from Jin et al. (2021) or Proposition 3 in our paper. Since, the V-type Bellman rank is 1, the BE dimension is bounded as $\dim_{\text{vbe}}^R(\mathcal{F}, \mathcal{D}_{\mathcal{F}}, 1/(2N)) \leq \tilde{O}(\log(N))$. We now show that the Q-type BE dimension is $\Omega(N)$. Consider the sequence of policies $\pi_1, \ldots, \pi_N$. For any $i \in \{1, \ldots, N\}$ and sequence $\pi_1, \ldots, \pi_{i-1}$, (Q-type) Bellman residuals incurred by the function $f_i$ is: $\sqrt{\sum_{k=1}^{i-1} \left( \frac{1}{N} \sum_{j=1}^{N} \mathbb{E}_{a \sim \pi_k} [f_i(j, a) - f^*(j, a)] \right)^2} = 0$. The same residual on the distribution induced by $\pi_i$ can be written as $\left| \sum_{j=1}^{N} \frac{1}{N} \mathbb{E}_{a \sim \pi_i} [f_i(j, a) - f^*(j, a)] \right| = 1/N$. Hence, $\pi_i$ is $1/(2N)$-independent of $\{\pi_1, \ldots, \pi_{i-1}\}$ (recall Definition 2). Thus, for $\varepsilon = 1/(2N)$, the sequence $\pi_1, \ldots, \pi_N$ can be used to show that the DE dimension (Definition 3) and the Q-type BE dimension for this instance is $\tilde{O}(N)$. Hence, the Q-type BE dimension is exponentially larger than the V-type BE dimension for this instance.

## C  Q-type RFOLIVE results

In this section, we present the results related to Q-type RFOLIVE. In Appendix C.1, we introduce the theoretical guarantee of Q-type OLIVE (Jiang et al., 2017; Jin et al., 2021) for completeness. In Appendix C.2, we show the detailed proof of the sample complexity bound of Q-type RFOLIVE (Theorem 1). In Appendix C.4, we discuss the instantiation of Q-type RFOLIVE to the known

representation linear completeness setting. In Appendix C.5, we provide another instantiation of Q-type RFOLIVE to the linear MDP with known feature.

## C.1   Q-type OLIVE

We first introduce the following assumption that will be useful for the OLIVE results (Proposition 1 and Proposition 2). Notice that this realizability assumption is for the single reward-aware OLIVE, where the function class captures reward-appended optimal value functions. Thus it is different from our reward-free realizability assumption (Assumption 1).

**Assumption 5** ($\varepsilon$-approximate realizability of the single-reward function class). *For the reward function $R$, optimal $Q$-function $Q_R^*$, and the value function class $\mathcal{F}$, there exists $Q_R^c \in \mathcal{F}$ so that $\max_{h \in [H]} \|Q_{R,h}^* - Q_{R,h}^c\|_\infty \leq \varepsilon$.*

Then we state the sample complexity result of Q-type OLIVE (Algorithm 2 in Jin et al. (2021)). In this paper, we consider the uniformly bounded reward setting ($0 \leq r_h \leq 1, \forall h \in [H]$) instead of bounded total reward setting ($\forall h \in [H], r_h \geq 0$ and $\sum_{h=0}^{H-1} r_h \leq 1$) in Jiang and Agarwal (2018); Jiang et al. (2017); Jin et al. (2021). Therefore we need to pay an additional $H^2$ dependency in $n_{\text{actv}}$ and $n_{\text{elim}}$ because the range of value function is $H$ times larger than the original ones, which induces an additional $H^2$ factor in the concentration inequalities.

**Proposition 1** (Sample complexity of Q-type OLIVE, modification of Theorem 18 in Jin et al. (2021)). *Under Assumption 5 with exact realizability (zero approximation error), if we set*

$$\varepsilon_{actv} = \frac{\varepsilon}{2H}, \; \varepsilon_{elim} = \frac{\varepsilon}{8H\sqrt{d_{\text{qbe}}}}, \; n_{actv} = \frac{H^4\iota}{\varepsilon^2}, \; and \; n_{elim} = \frac{H^4 d_{\text{qbe}} \log(\mathcal{N}_\mathcal{F}(\varepsilon_{elim}/64))\iota}{\varepsilon^2}$$

*where $d_{\text{qbe}} = \dim_{\text{qbe}}^R(\mathcal{F}, \mathcal{D}_\mathcal{F}, \varepsilon/(4H))$ and $\iota = c_1 \log(H d_{\text{qbe}}/\delta\varepsilon)$, then with probability at least $1 - \delta$, Q-type OLIVE (Algorithm 2 in Jin et al. (2021)) with $\mathcal{F}$ will output an $\varepsilon$-optimal policy (under a single reward function $R$) using at most $O(H d_{\text{qbe}}(n_{actv} + n_{elim}))$ episodes. Here $c_1$ is a large enough constant.*

This sample complexity result directly follows from Jin et al. (2021) with minor adaptation of the parameters. We refer the reader to Jin et al. (2021) for the detailed proof.

## C.2   Proof of Q-type RFOLIVE under general function approximation

In this part, we first provide the general statement of Theorem 1 and then show the detailed proof. We also provide a detailed discussion on the different and novel part in our proof compared with Jiang et al. (2017); Jin et al. (2021).

**Theorem** (Full version of Theorem 1). *Fix $\delta \in (0, 1)$. Given a reward class $\mathcal{R}$ and a function class $\mathcal{F}$ that satisfies Assumption 1 and Assumption 2, with probability at least $1 - \delta$, for any $R \in \mathcal{R}$, Q-type RFOLIVE (Algorithm 1) with $\mathcal{F}$ outputs a policy $\hat{\pi}$ that satisfies $v_R^{\hat{\pi}} \geq v_R^* - \varepsilon$. The required number of episodes is*

$$O\left(\frac{\left(H^7 \log\left(\mathcal{N}_\mathcal{F}\left(\varepsilon/512H^2\sqrt{d_{\text{qbe}}}\right)\right) + H^5 \log\left(\mathcal{N}_\mathcal{R}\left(\varepsilon/512H^2\sqrt{d_{\text{qbe}}}\right)\right)\right) d_{\text{qbe}}^2 \iota}{\varepsilon^2}\right).$$

*In RFOLIVE, we set*

$$\varepsilon_{actv} = \frac{\varepsilon}{2H^2}, \varepsilon_{elim} = \frac{\varepsilon}{8H^2\sqrt{d_{\text{qbe}}}}, n_{actv} = \frac{H^6\iota}{\varepsilon^2},$$

*and*

$$n_{elim} = \frac{\left(H^6 \log(\mathcal{N}_\mathcal{F}(\varepsilon_{elim}/64)) + H^4 \log(\mathcal{N}_\mathcal{R}(\varepsilon_{elim}/64))\right) d_{\text{qbe}} \iota}{\varepsilon^2}$$
$$= \frac{\left(H^6 \log\left(\mathcal{N}_\mathcal{F}\left(\varepsilon/512H^2\sqrt{d_{\text{qbe}}}\right)\right) + H^4 \log\left(\mathcal{N}_\mathcal{R}\left(\varepsilon/512H^2\sqrt{d_{\text{qbe}}}\right)\right)\right) d_{\text{qbe}} \iota}{\varepsilon^2},$$

*where $d_{\text{qbe}} = \dim_{\text{qbe}}^0(\mathcal{F} - \mathcal{F}, \mathcal{D}_{\mathcal{F}-\mathcal{F}}, \varepsilon/(4H))$, $\iota = c_2 \log(H d_{\text{qbe}}/\delta\varepsilon)$, and $c_2$ is a large enough constant.*

*Proof.* From the online phase of Q-type RFOLIVE (Algorithm 1), we can see that this phase is equivalent to running Q-type OLIVE (Algorithm 2 in Jin et al. (2021)) with the input function class $\mathcal{F} - \mathcal{F}$, the specified parameters $\varepsilon_{\mathrm{actv}}, \varepsilon_{\mathrm{elim}}, n_{\mathrm{elim}}, n_{\mathrm{actv}}$ and under the reward function $R = \mathbf{0}$. In Proposition 1, we know that realizability (Assumption 5) holds because $\mathbf{0} \in \mathcal{F} - \mathcal{F} = \mathcal{F}_{\mathrm{on}}$. Then the sample complexity is immediately from our specified values of $\varepsilon_{\mathrm{actv}}, \varepsilon_{\mathrm{elim}}, n_{\mathrm{actv}}, n_{\mathrm{elim}}$ and Proposition 1 as we only collect samples in the online phase. Notice that the log-covering number $\log\left(\mathcal{N}_{\mathcal{F}_{\mathrm{on}}}(\cdot)\right) = \log\left(\mathcal{N}_{\mathcal{F}-\mathcal{F}}(\cdot)\right) \leq 2\log\left(\mathcal{N}_{\mathcal{F}}(\cdot)\right)$ and such a constant 2 is absorbed by large enough $c_2$. Therefore, it remains to show that the algorithm can indeed output an $\varepsilon$-optimal policy with probability $1 - \delta$ in the offline phase. We will show the following three claims hold with probability at least $1 - \delta$.

**Claim 1** For any $g \in \mathcal{F}_{\mathrm{off}}(R)$, if $\exists h \in [H]$, s.t. $|\mathcal{E}_Q^R(g, \pi_g, h)| \geq \varepsilon/H$, then it will be eliminated in the offline phase.

**Claim 2** $Q_R^* \in \mathcal{F}_{\mathrm{off}}(R)$ and $Q_R^*$ will not be eliminated in the offline phase.

**Claim 3** At the end of the offline phase, picking the optimistic function from the survived value functions gives us $\varepsilon$-optimal policy.

Before showing these three claims, we first state properties from the online phase of Q-type RFOLIVE and the concentration results in the offline phase.

**Properties from the online phase of Q-type RFOLIVE** From the equivalence between the online phase of Q-type RFOLIVE (Algorithm 1) and Q-type OLIVE (Algorithm 2 in Jin et al. (2021)) with reward $\mathbf{0}$, we know that with probability at least $1 - \delta/4$, the online phase terminates within $d_{\mathrm{qbe}}H + 1$ iterations. In addition, with probability at least $1 - \delta/4$, the following properties (Eq. (6) and Eq. (7)) hold for the first $d_{\mathrm{qbe}}H + 1$ iterations:

(i) When the online phase exits at iteration $T$ in line 7 (i.e., the elimination procedure is not activated in RFOLIVE), for any $f \in \mathcal{F}^T$, it predicts no more than $\varepsilon/H$ value:

$$V_f(x_0) \leq V_{f^T}(x_0) = V_{f^T}(x_0) - V_{\mathbf{0},0}^{\pi_{f^T}}(x_0) = \sum_{h=0}^{H-1} \mathcal{E}_Q^R(f^T, \pi^T, h) < 2H\varepsilon_{\mathrm{actv}} = \varepsilon/H. \quad (6)$$

The first equality is due to any policy evaluation has value 0 under the reward function $\mathbf{0}$. The second equality is due to the policy loss decomposition in Jiang et al. (2017). The second inequality is adapted from the "concentration in the activation procedure" part of the proof for Theorem 18 in Jin et al. (2021).

(ii) For $T \leq d_{\mathrm{qbe}}H + 1$, the concentration argument holds for any $f \in \mathcal{F}_{\mathrm{on}}$ and $t \in [T]$:

$$\left|\hat{\mathcal{E}}_Q^{\mathbf{0}}(f, \pi^t, h^t) - \mathcal{E}_Q^{\mathbf{0}}(f, \pi^t, h^t)\right| < \varepsilon_{\mathrm{elim}}/8. \quad (7)$$

This is from the "concentration in the elimination procedure" step of the proof for Theorem 18 in Jin et al. (2021) and we adapt it with our parameters.

**Concentration results in the offline phase** In RFOLIVE we use $\hat{\mathcal{E}}^R(g, \pi^t, \pi, h^t)$ and $\pi \in \Pi_{\mathrm{est}}$ in line 18. Since we are in the Q-type version, we have $\Pi_{\mathrm{est}} = \{\pi^t\}$. In addition, from Definition 4, we know that $\mathcal{E}^R(g, \pi^t, \pi^t, h^t) = \mathcal{E}_Q^R(g, \pi^t, h^t)$. Therefore, in line 18, it is equivalent to eliminating according to $\hat{\mathcal{E}}_Q^R(g, \pi^t, h^t)$. Throughout this proof, we will use $\mathcal{E}_Q^R(g, \pi^t, h^t)$ and $\hat{\mathcal{E}}_Q^R(g, \pi^t, h^t)$ notations for simplicity. Now we show the concentration results in the offline phase.

Let $\overline{\mathcal{R}}$ be an $(\varepsilon_{\mathrm{elim}}/64)$-cover of $\mathcal{R}$. For every $R \in \mathcal{R}$, let $R^{\mathrm{c}} = \mathrm{argmin}_{R' \in \overline{\mathcal{R}}} \max_{h \in [H]} \|R_h - R_h'\|_\infty$. Firstly, consider any fixed $R \in \overline{\mathcal{R}}$ and let $\mathcal{Z}(R)$ be an $(\varepsilon_{\mathrm{elim}}/64)$-cover of $\mathcal{F}_{\mathrm{off}}(R)$ with cardinality $\mathcal{N}_{\mathcal{F}_{\mathrm{off}}(R)}(\varepsilon_{\mathrm{elim}}/64) = \mathcal{N}_{\mathcal{F}}(\varepsilon_{\mathrm{elim}}/64)$. For every $g \in \mathcal{F}_{\mathrm{off}}(R)$, let $g^{\mathrm{c}} = \mathrm{argmin}_{g' \in \mathcal{Z}(R)} \max_{h \in [H]} \|g_h - g_h'\|_\infty$.

Applying Hoeffding's inequality to all $(t, g') \in [T] \times \mathcal{Z}(R)$ and taking a union bound, we have that with probability at least $1 - \delta/(2\mathcal{N}_{\mathcal{R}}(\varepsilon_{\mathrm{elim}}/64))$, the following holds for all $(t, g') \in [T] \times \mathcal{Z}(R)$

$$\left| \hat{\mathcal{E}}_{\mathrm{Q}}^{R}(g', \pi^t, h^t) - \mathcal{E}_{\mathrm{Q}}^{R}(g', \pi^t, h^t) \right| \leq 4H \sqrt{\frac{\log(4T\mathcal{N}_{\mathcal{R}}(\varepsilon_{\mathrm{elim}}/64)\mathcal{N}_{\mathcal{F}}(\varepsilon_{\mathrm{elim}}/64)/\delta)}{2n_{\mathrm{elim}}}} < \varepsilon_{\mathrm{elim}}/8.$$

The second inequality is due to $\varepsilon_{\mathrm{elim}} = \varepsilon/\left(8H^2\sqrt{d_{\mathrm{qbe}}}\right)$, $\iota = c_2 \log(Hd_{\mathrm{qbe}}/\delta\varepsilon)$, and

$$n_{\mathrm{elim}} = \frac{(H^6 \log(\mathcal{N}_{\mathcal{F}}(\varepsilon_{\mathrm{elim}}/64)) + H^4 \log(\mathcal{N}_{\mathcal{R}}(\varepsilon_{\mathrm{elim}}/64)))d_{\mathrm{qbe}}\iota}{\varepsilon^2},$$

with $c_2$ in $\iota$ being chosen large enough.

Therefore for any $g \in \mathcal{F}_{\mathrm{off}}(R)$, we get

$$\left| \hat{\mathcal{E}}_{\mathrm{Q}}^{R}(g, \pi^t, h^t) - \mathcal{E}_{\mathrm{Q}}^{R}(g, \pi^t, h^t) \right|$$
$$\leq \left| \hat{\mathcal{E}}_{\mathrm{Q}}^{R}(g, \pi^t, h^t) - \hat{\mathcal{E}}_{\mathrm{Q}}^{R}(g^{\mathrm{c}}, \pi^t, h^t) \right| + \left| \hat{\mathcal{E}}_{\mathrm{Q}}^{R}(g^{\mathrm{c}}, \pi^t, h^t) - \mathcal{E}_{\mathrm{Q}}^{R}(g^{\mathrm{c}}, \pi^t, h^t) \right|$$
$$\quad + \left| \mathcal{E}_{\mathrm{Q}}^{R}(g^{\mathrm{c}}, \pi^t, h^t) - \mathcal{E}_{\mathrm{Q}}^{R}(g, \pi^t, h^t) \right|$$
$$\leq 2\varepsilon_{\mathrm{elim}}/64 + \varepsilon_{\mathrm{elim}}/8 + 2\varepsilon_{\mathrm{elim}}/64$$
$$= 3\varepsilon_{\mathrm{elim}}/16.$$

Union bounding over $R \in \overline{\mathcal{R}}$, with probability at least $1 - \delta/2$, for all $t \in [T]$, $R \in \overline{\mathcal{R}}$, $g \in \mathcal{F}_{\mathrm{off}}(R)$, we have

$$\left| \hat{\mathcal{E}}_{\mathrm{Q}}^{R}(g, \pi^t, h^t) - \mathcal{E}_{\mathrm{Q}}^{R}(g, \pi^t, h^t) \right| \leq 3\varepsilon_{\mathrm{elim}}/16.$$

Therefore, with probability at least $1 - \delta/2$, for all $t \in [T]$, $R \in \mathcal{R}$, $g \in \mathcal{F}_{\mathrm{off}}(R)$, we have

$$\left| \hat{\mathcal{E}}_{\mathrm{Q}}^{R}(g, \pi^t, h^t) - \mathcal{E}_{\mathrm{Q}}^{R}(g, \pi^t, h^t) \right|$$
$$\leq \left| \hat{\mathcal{E}}_{\mathrm{Q}}^{R}(g, \pi^t, h^t) - \hat{\mathcal{E}}_{\mathrm{Q}}^{R^{\mathrm{c}}}(g, \pi^t, h^t) \right| + \left| \hat{\mathcal{E}}_{\mathrm{Q}}^{R^{\mathrm{c}}}(g, \pi^t, h^t) - \mathcal{E}_{\mathrm{Q}}^{R^{\mathrm{c}}}(g, \pi^t, h^t) \right|$$
$$\quad + \left| \mathcal{E}_{\mathrm{Q}}^{R^{\mathrm{c}}}(g, \pi^t, h^t) - \mathcal{E}_{\mathrm{Q}}^{R}(g, \pi^t, h^t) \right|$$
$$\leq \varepsilon_{\mathrm{elim}}/64 + 3\varepsilon_{\mathrm{elim}}/16 + \varepsilon_{\mathrm{elim}}/64$$
$$< \varepsilon_{\mathrm{elim}}/4. \tag{8}$$

All statements in our subsequent proof are under the event that all the different high-probability events (the online phase terminates within $d_{\mathrm{qbe}}H + 1$ iterations, and Eq. (6), Eq. (7), Eq. (8) hold for the first $d_{\mathrm{qbe}}H + 1$ iterations) discussed above hold with a total failure probability of $\delta$.

**Proof of Claim 1**  Consider any $g \in \mathcal{F}_{\mathrm{off}}(R)$ such that $\exists h \in [H], |\mathcal{E}_{\mathrm{Q}}^{R}(g, \pi_g, h)| \geq \varepsilon/H$. Recall the definition of $\mathcal{F}_{\mathrm{off}}(R)$, we know that $g$ can be written as $g = (g_0, \ldots, g_{H-1}) = (f_0 + R_0, \ldots, f_{H-1} + R_{H-1})$, $f_h \in \mathcal{F}_h$. We will discuss the positive average Bellman error and the negative average Bellman error cases separately.

**Case (i) of Claim 1**  $\mathcal{E}_{\mathrm{Q}}^{R}(g, \pi_g, h) = \mathbb{E}[g_h(x_h, a_h) - R_h(x_h, a_h) - V_g(x_{h+1}) \mid a_{0:h} \sim \pi_g] \geq \varepsilon/H$.

Since $g_h = f_h + R_h$, we know that

$$\varepsilon/H \leq \mathbb{E}\left[g_h(x_h, a_h) - R_h(x_h, a_h) - V_g(x_{h+1}) \mid a_{0:h} \sim \pi_g\right]$$
$$= \mathbb{E}[f_h(x_h, a_h) - V_g(x_{h+1}) \mid a_{0:h} \sim \pi_g]$$
$$= \mathbb{E}[f_h(x_h, a_h) - (\mathcal{T}_h^{\mathbf{0}} g_{h+1})(x_h, a_h) \mid a_{0:h} \sim \pi_g] \qquad \text{(Definition of } \mathcal{T}_h^{\mathbf{0}})$$
$$= \mathbb{E}[\tilde{f}_h(x_h, a_h) \mid a_{0:h} \sim \pi_g]. \qquad (\tilde{f}_h := f_h - \mathcal{T}_h^{\mathbf{0}} g_{h+1})$$

Here we construct a function $\tilde{f}$ that has the same value as $f_h - \mathcal{T}_h^{\mathbf{0}} g_{h+1}$ at level $h$, uses zero reward Bellman backup for any level before $h$, and assigns zero value after level $h$. More formally, it is

defined as

$$\tilde{f}_{h'}(x_{h'}, a_{h'}) = \begin{cases} (\mathcal{T}_{h'}^{\mathbf{0}}\tilde{f}_{h'+1})(x_{h'}, a_{h'}) = \mathbb{E}[\max_a \tilde{f}_{h'+1}(x_{h'+1}, a) \mid x_{h'}, a_{h'}] & 0 \le h' \le h-1 \\ f_h(x_h, a_h) - (\mathcal{T}_h^{\mathbf{0}}g_{h+1})(x_h, a_h) & h' = h \\ 0 & h+1 \le h' \le H-1. \end{cases}$$

From the definition of Q-type average Bellman error and the construction, we know that for any policy $\pi$ we can translate the Q-type reward-dependent average Bellman error for a function $g \in \mathcal{F}_{\text{off}}(R)$ to the zero reward Q-type average Bellman error of a function $\tilde{f} \in \mathcal{F}_{\text{on}}$ as the following

$$\begin{aligned} \mathcal{E}_Q^R(g, \pi, h) &= \mathbb{E}[g_h(x_h, a_h) - R_h(x_h, a_h) - V_g(x_{h+1}) \mid a_{0:h-1} \sim \pi, a_h \sim \pi] \\ &= \mathbb{E}[f_h(x_h, a_h) - (\mathcal{T}_h^{\mathbf{0}}g_{h+1})(x_h, a_h) \mid a_{0:h} \sim \pi] \\ &= \mathbb{E}[\tilde{f}_h(x_h, a_h) - \mathbf{0} - \tilde{f}_{h+1}(x_{h+1}, a_{h+1}) \mid a_{0:h} \sim \pi, a_{h+1} \sim \pi_{\tilde{f}}] \\ &= \mathcal{E}_Q^{\mathbf{0}}(\tilde{f}, \pi, h), \end{aligned} \tag{9}$$

where in the third equality we notice that $\tilde{f}_{h+1} = \mathbf{0}$.

We can verify that $\tilde{f} = (\tilde{f}_0, \ldots, \tilde{f}_{H-1}) \in \mathcal{F}_{\text{on}}$. First let us consider level $h' = h$. From completeness (Assumption 2), we know that $\mathcal{T}_h^{\mathbf{0}}g_{h+1} = \mathcal{T}_h^{\mathbf{0}}(f_{h+1} + R_{h+1}) \in \mathcal{F}_h$. Therefore, we have $\tilde{f}_h = f_h - \mathcal{T}_h^{\mathbf{0}}g_{h+1} \in \mathcal{F}_h - \mathcal{F}_h$. Then we consider level $0 \le h' \le h-1$. By the definition, we use zero reward Bellman backup. From completeness and $\tilde{f}_h \in \mathcal{F}_h - \mathcal{F}_h$, we have $\tilde{f}_{h-1} = \mathcal{T}_h^{\mathbf{0}}\tilde{f}_h \in \mathcal{F}_{h-1} - \mathcal{F}_{h-1}$. By performing this inductive process backward, we have $\tilde{f}_{h'} \in \mathcal{F}_{h'} - \mathcal{F}_{h'}$ for any $0 \le h' \le h-1$. For level $h+1 \le h' \le H-1$, we immediately get $\tilde{f}_{h'} = \mathbf{0} \in \mathcal{F}_{h'} - \mathcal{F}_{h'}$. Therefore, we can see $\tilde{f} = (\tilde{f}_0, \ldots, \tilde{f}_{H-1}) \in \mathcal{F}_{\text{on}}$ from the definition of $\mathcal{F}_{\text{on}}$.

From the construction of $\tilde{f}$ (zero reward Bellman backup for level $0 \le h' \le h-1$), we have

$$\begin{aligned} V_{\tilde{f}}(x_0) &= \mathbb{E}[\tilde{f}_h(x_h, a_h) \mid a_{0:h} \sim \pi_{\tilde{f}}] \\ &\ge \mathbb{E}[\tilde{f}_h(x_h, a_h) \mid a_{0:h} \sim \pi_g] \\ &= \mathbb{E}[g_h(x_h, a_h) - R_h(x_h, a_h) - V_g(x_{h+1}) \mid a_{0:h} \sim \pi_g] \\ &\ge \varepsilon/H, \end{aligned}$$

where $\pi_{\tilde{f}}$ is the greedy policy of $\tilde{f}$ and in fact it is the optimal policy when treating $\tilde{f}_h$ as the reward at level $h$ and there are no intermediate rewards. From the first property of the online phase (Eq. (6)), we know that all the survived value functions at the end of the online phase predict no more than $\varepsilon/H$. Therefore $\tilde{f}$ will be eliminated. We assume it is eliminated at iteration $t$ by policy $\pi^t$ in level $h^t$.

From the Bellman backup construction of $\tilde{f}$, we know that $\tilde{f}$ can only be eliminated at level $h$. This can be seen from the following argument: By the construction of $\tilde{f}$, we have $\mathcal{E}_Q^{\mathbf{0}}(\tilde{f}, \pi, h') = 0$ for any $\pi$ and $h' \in [H], h' \ne h$. Applying the second property of the online phase (Eq. (7)), we have $\left|\hat{\mathcal{E}}_Q^{\mathbf{0}}(\tilde{f}, \pi^t, h^t) - \mathcal{E}_Q^{\mathbf{0}}(\tilde{f}, \pi^t, h^t)\right| \le 3\varepsilon_{\text{elim}}/4$, which gives us $\left|\hat{\mathcal{E}}_Q^{\mathbf{0}}(\tilde{f}, \pi^t, h^t)\right| \le 3\varepsilon_{\text{elim}}/4$ if $h^t \ne h$. Since the elimination threshold is set to $\varepsilon_{\text{elim}}$, $\tilde{f}$ will not be eliminated at level $h^t \ne h$.

This implies that at some iteration $t$ in the online phase, we will collect some $\pi^t$ that eliminates $\tilde{f}$ at level $h$, i.e., it satisfies $\left|\hat{\mathcal{E}}_Q^{\mathbf{0}}(\tilde{f}, \pi^t, h^t)\right| > \varepsilon_{\text{elim}}$ and $h^t = h$. Applying the second property of the online phase (Eq. (7)), we have $\left|\hat{\mathcal{E}}_Q^{\mathbf{0}}(\tilde{f}, \pi^t, h^t) - \mathcal{E}_Q^{\mathbf{0}}(\tilde{f}, \pi^t, h^t)\right| \le \varepsilon_{\text{elim}}/8$. This tells us $\left|\mathcal{E}_Q^{\mathbf{0}}(\tilde{f}, \pi^t, h^t)\right| > 7\varepsilon_{\text{elim}}/8$. Then from Eq. (9) we have

$$\left|\mathcal{E}_Q^R(g, \pi^t, h^t)\right| = \left|\mathcal{E}_Q^{\mathbf{0}}(\tilde{f}, \pi^t, h^t)\right| > 7\varepsilon_{\text{elim}}/8.$$

Finally, the concentration argument of the offline phase (Eq. (8)) implies that $\left|\hat{\mathcal{E}}_Q^R(g, \pi^t, h^t) - \mathcal{E}_Q^R(g, \pi^t, h^t)\right| < \varepsilon_{\text{elim}}/4$. Hence, we get $\left|\hat{\mathcal{E}}_Q^R(g, \pi^t, h^t)\right| > \varepsilon_{\text{elim}}/2$. This means that we will eliminate such $g$ by $\pi^t$ in the offline phase.

**Case (ii) of Claim 1** $\mathcal{E}_{\mathrm{Q}}^{R}(g, \pi_g, h) = \mathbb{E}[g_h(x_h, a_h) - R_h(x_h, a_h) - V_g(x_{h+1}) \mid a_{0:h} \sim \pi_g] \leq -\varepsilon/H$.

Same as before, we have $\mathcal{E}_{\mathrm{Q}}^{R}(g, \pi_g, h) = \mathbb{E}[f_h(x_h, a_h) - (\mathcal{T}_h^{\mathbf{0}} g_{h+1})(x_h, a_h) \mid a_{0:h} \sim \pi_g] \leq -\varepsilon/H$.
Now we let $\tilde{f}_h$ be the negated version of the one in case (i), and define $\tilde{f}$ as

$$\tilde{f}_{h'}(x_{h'}, a_{h'}) = \begin{cases} (\mathcal{T}_{h'}^{\mathbf{0}} \tilde{g}_{h'+1})(x_{h'}, a_{h'}) = \mathbb{E}[\max_a \tilde{g}_{h'+1}(x_{h'+1}, a) \mid x_{h'}, a_{h'}] & 0 \leq h' \leq h-1 \\ (\mathcal{T}_h^{\mathbf{0}} g_{h+1})(x_h, a_h) - f_h(x_h, a_h) & h' = h \\ 0 & h+1 \leq h' \leq H-1. \end{cases}$$

Following the same steps as in case (i) we can verify that $\tilde{f} \in \mathcal{F}_{\mathrm{on}}$, and that $V_{\tilde{f}}(x_0) \geq \varepsilon/H$. From here the argument is identical to case (i).

**Proof of Claim 2** (i) From the assumption, we know that realizability condition $Q_R^* = (Q_{R,0}^*, \ldots, Q_{R,H-1}^*) \in \mathcal{F}_{\mathrm{off}}(R)$ holds. (ii) For the second argument, we note that $\mathcal{E}_{\mathrm{Q}}^{R}(Q_R^*, \pi, h) = 0$ for any $\pi$ and $h \in [H]$ by the definition of the average Bellman error. From the concentration argument in the offline phase (Eq. (8)), we have $\left| \hat{\mathcal{E}}_{\mathrm{Q}}^{R}(Q_R^*, \pi^t, h^t) \right| \leq \left| \mathcal{E}_{\mathrm{Q}}^{R}(Q_R^*, \pi^t, h^t) \right| + \varepsilon_{\mathrm{elim}}/4 = \varepsilon_{\mathrm{elim}}/4$. As a result, $Q_R^*$ will not be eliminated.

**Proof of Claim 3** From Claim 1, we know that in the offline phase for any $g \in \mathcal{F}_{\mathrm{off}}(R)$, if $\exists h \in [H]$, s.t. $|\mathcal{E}_{\mathrm{Q}}^{R}(g, \pi_g, h)| \geq \varepsilon/H$, then it will be eliminated. Therefore from the policy loss decomposition in Jiang et al. (2017), for all survived $g \in \mathcal{F}_{\mathrm{sur}}(R)$ in the offline phase, we have

$$V_g(x_0) - V_{R,0}^{\pi_g}(x_0) = \sum_{h=0}^{H-1} \mathcal{E}_{\mathrm{Q}}^{R}(g, \pi_g, h) < \varepsilon.$$

Since $Q_R^*$ is not eliminated, similar as Jiang et al. (2017); Jin et al. (2021), we have

$$V_{R,0}^{\pi_{\hat{g}}}(x_0) > V_{\hat{g}}(x_0) - \varepsilon \geq V_{R,0}^*(x_0) - \varepsilon.$$

Notice that Claim 3 directly implies that RFOLIVE returns an $\varepsilon$-near optimal policy. This completes the proof. $\qquad\square$

## C.3 Technical novelty over reward-aware OLIVE

The key step of the analyses of reward-aware OLIVE (Jiang et al., 2017; Jin et al., 2021) is to show that any bad function whose average Bellman error is large under the given reward function is eliminated (recall that they only have the online phase and the reward is always revealed). This is ensured by the online exploration process. However, the difficulty in our reward-free RL setting is that such a reward function is only revealed in the offline phase, where we no longer actively explore. To overcome this difficulty, we use completely new and novel proof techniques here: For each bad function $g \in \mathcal{F}_{\mathrm{off}}(R)$ with a large average Bellman error under the true reward $R$, we construct a surrogate function $\tilde{f}$ in the online phase. Our construction guarantees that $\tilde{f}$ has the same large average Bellman error as $g$, but the error is instead under the zero reward which we use during exploration (Eq. (9)). Then we show that all these constructed $\tilde{f}$ belong to the "difference" function class $\mathcal{F}_{\mathrm{on}}$ and $\tilde{f}$ will be eliminated in the online phase since we use $\mathcal{F}_{\mathrm{on}}$ and zero reward there. The collected data tuples (gathered constraints) that eliminate $\tilde{f}$ will be used in the offline phase and they guarantee eliminating its corresponding bad function $g$. Notice that in the design/definition of $\tilde{f}$, we need to guarantee that it has a large average Bellman error at the same timestep as $g$ does so that it can correctly witness the average Bellman error of $g$, which we ensure via a Bellman backup construction.

In summary, both the construction of the surrogate function $\tilde{f}$ and the translation of average Bellman error from bad function $g \in \mathcal{F}_{\mathrm{off}}(R)$ to $\tilde{f} \in \mathcal{F}_{\mathrm{on}}$ are novel to the best of our knowledge. They reflect crucial difference between reward-aware and reward-free RL. And at the same time, no reward-aware RL works have used such mechanisms before.

We also provide a counterexample in Appendix G that shows other variant of OLIVE could fail even under realizability, completeness, and low Bellman Eluder dimension, where we know RFOLIVE has polynomial sample complexities.

## C.4 Q-type RFOLIVE for known representation linear completeness setting

We first discuss why stating a specific $B$ is equivalent to stating any $B > 0$ in Definition 6. Assuming the statement hold for $B$, we will show that it holds for any $B' > 0$. The reason is the following. Consider any $Q'_{h+1} = \langle \phi^{\mathrm{lc}}_{h+1}, \theta'_{h+1} \rangle \in \mathcal{Q}_{h+1}(\{\phi^{\mathrm{lc}}\}, B')$, where $\|\theta'_{h+1}\|_2 \leq B'\sqrt{d_{\mathrm{lc}}}$. For $Q_{h+1} = \left\langle \phi^{\mathrm{lc}}_{h+1}, \theta'_{h+1} \frac{B\sqrt{d_{\mathrm{lc}}}}{\|\theta'_{h+1}\|_2} \right\rangle \in \mathcal{Q}_{h+1}(\{\phi^{\mathrm{lc}}\}, B)$, there exists $\theta_h$ that satisfies $\langle \phi^{\mathrm{lc}}_h, \theta_h \rangle = \mathcal{T}^0_h Q_{h+1}$ and $\|\theta_h\|_2 \leq B\sqrt{d_{\mathrm{lc}}}$. Then we know that $\frac{\|\theta'_{h+1}\|_2}{B\sqrt{d_{\mathrm{lc}}}} \langle \phi^{\mathrm{lc}}_h, \theta_h \rangle = \mathcal{T}^0_h Q'_{h+1}$. Now we can choose $\theta'_h = \frac{\|\theta'_{h+1}\|_2}{B\sqrt{d_{\mathrm{lc}}}} \theta_h$, and therefore we have $\mathcal{T}^0_h Q'_{h+1} = \langle \phi^{\mathrm{lc}}_h, \theta'_h \rangle$ and $\|\theta'_h\|_2 \leq \|\theta'_{h+1}\|_2 \leq B'\sqrt{d_{\mathrm{lc}}}$ satisfies the norm constraint, i.e., $\langle \phi^{\mathrm{lc}}_h, \theta'_h \rangle \in \mathcal{Q}_h(\{\phi^{\mathrm{lc}}\}, B')$.

Next, we show the formal corollary statement and the detailed proof of the theoretical result of Q-type RFOLIVE when instantiated to linear completeness setting.

**Corollary** (Full version of Corollary 2). *Fix $\delta \in (0, 1)$. Consider an MDP $M$ that satisfies linear completeness (Definition 6) with the known feature $\phi^{\mathrm{lc}}$, and the linear reward class $\mathcal{R} = \mathcal{R}_1 \times \ldots \times \mathcal{R}_h$, where $\mathcal{R}_h = \{\langle \phi^{\mathrm{lc}}_h, \eta_h \rangle : \|\eta_h\|_2 \leq \sqrt{d_{\mathrm{lc}}}, \langle \phi^{\mathrm{lc}}_h(\cdot), \eta_h \rangle \in [0, 1]\}$. With probability at least $1 - \delta$, for any $R \in \mathcal{R}$, Q-type RFOLIVE (Algorithm 1) with $\mathcal{F} = \mathcal{F}(\{\phi^{\mathrm{lc}}\})$ outputs a policy $\hat{\pi}$ that satisfies $v^{\hat{\pi}}_R \geq v^*_R - \varepsilon$. The required number of episodes is*

$$\tilde{O}\left(\frac{H^8 d^3_{\mathrm{lc}} \log(1/\delta)}{\varepsilon^2}\right).$$

*In RFOLIVE, we set*

$$\varepsilon_{actv} = \frac{\varepsilon}{2H^2}, \varepsilon_{elim} = \frac{\varepsilon}{8H^2\sqrt{d_{\mathrm{lc}}}\iota}, n_{actv} = \frac{H^6\iota}{\varepsilon^2}, n_{elim} = \frac{H^7 d^2_{\mathrm{lc}}\iota^3}{\varepsilon^2},$$

*where $\iota = c_3 \log(Hd_{\mathrm{lc}}/\delta\varepsilon)$ and $c_3$ is a large enough constant.*

We remark that although the $\tilde{O}\left(\frac{H^5 d^3_{\mathrm{lc}} \log(1/\delta)}{\varepsilon^2}\right)$ sample complexity rate of Zanette et al. (2020b) looks better than us, they need to assume $\varepsilon \leq \tilde{O}(\nu_{\min}/\sqrt{d_{\mathrm{lc}}})$, where $\nu_{\min}$ is their explorability factor. Thus there is an implicit dependence on $1/\nu_{\min}$ in their sample complexity bound and their results are incomparable to us. More related discussions can be found in Appendix A.

*Proof.* We first verify that $\mathcal{F}(\{\phi^{\mathrm{lc}}\})$ satisfies the assumptions in Theorem 1. Here we have that $\mathcal{F}(\{\phi^{\mathrm{lc}}\}) = \mathcal{F}_0(\{\phi^{\mathrm{lc}}\}, H-1) \times \ldots \times \mathcal{F}_{H-1}(\{\phi^{\mathrm{lc}}\}, 0)$, where

$$\mathcal{F}_h(\{\phi^{\mathrm{lc}}\}, B_h) = \{f_h(x_h, a_h) = \langle \phi^{\mathrm{lc}}_h(x_h, a_h), \theta_h \rangle : \|\theta_h\|_2 \leq B_h\sqrt{d_{\mathrm{lc}}}, \langle \phi^{\mathrm{lc}}_h(\cdot), \theta_h \rangle \in [-B_h, B_h]\}.$$

We first verify the realizability assumption (Assumption 1). For the last level, we have

$$Q^*_{R,H-1} = R_{H-1} + \mathbf{0} \in \mathcal{F}_{H-1}(\{\phi^{\mathrm{lc}}\}, 0) + R_{H-1} = \mathcal{F}_{H-1} + R_{H-1}.$$

In addition, $Q^*_{R,H-1} = R_{H-1} = \langle \phi^{\mathrm{lc}}_{H-1}, \eta_{H-1} \rangle$, where $\|\eta_{H-1}\|_2 \leq \sqrt{d_{\mathrm{lc}}}$ and $\langle \phi^{\mathrm{lc}}_{H-1}(\cdot), \eta_{H-1} \rangle \in [0, 1]$. Then for level $H-2$, we have

$$Q^*_{R,H-2}(x_{H-2}, a_{H-2}) = R_{H-2}(x_{H-2}, a_{H-2}) + \mathbb{E}[\max_{a_{H-1}} Q^*_{R,H-1}(x_{H-1}, a_{H-1}) \mid x_{H-2}, a_{H-2}]$$

$$= R_{H-2}(x_{H-2}, a_{H-2}) + \langle \phi^{\mathrm{lc}}_{H-2}(x_{H-2}, a_{H-2}), \theta'_{H-2} \rangle,$$

where $\|\theta'_{H-2}\|_2 \leq \sqrt{d_{\mathrm{lc}}}$ and $\langle \phi^{\mathrm{lc}}_{H-2}(\cdot), \theta'_{H-2} \rangle \in [0, 1]$. Here we apply the property of linear completeness (Definition 6). Therefore, we can set $\theta^*_{H-2} = \theta'_{H-2}$ and get $Q^*_{R,H-2} = R_{H-2} + \langle \phi^{\mathrm{lc}}_{H-2}, \theta^*_{H-2} \rangle \in \mathcal{F}_{H-2}(\{\phi^{\mathrm{lc}}\}, 1) + R_{H-2}$. Continuing this induction process backward, we get $Q^*_{R,h} \in \mathcal{F}_h + R_h, \forall h \in [H]$, thus $Q^*_R \in \mathcal{F} + R$.

For completeness assumption (Assumption 2), again from the property of linear completeness, for any $h \in [H], f_{h+1} \in \mathcal{F}_{h+1}, R_{h+1} \in \mathcal{R}_{h+1}$, we have that

$$(\mathcal{T}^0_h(f_{h+1} + R_{h+1}))(x_h, a_h) = \mathbb{E}\left[\max_{a_{h+1}} \langle \phi^{\mathrm{lc}}_{h+1}(x_{h+1}, a_{h+1}), \theta_{h+1} + \eta_{h+1} \rangle \mid x_h, a_h\right]$$

$$= \langle \phi^{\mathrm{lc}}_h(x_h, a_h), \theta_{f+R,h} \rangle,$$

where $\|\theta_{f+R,h}\|_2 \leq (H-h-1)\sqrt{d_{\mathrm{lc}}}$ and $\langle \phi_h^{\mathrm{lc}}(\cdot), \theta_{f+R,h}\rangle \in [-(H-h-1), H-h-1]$. Thus $\langle \phi_h^{\mathrm{lc}}, \theta_{f+R,h}\rangle \in \mathcal{F}_h$, which implies that for any $f_{h+1} \in \mathcal{F}_{h+1}, R_{h+1} \in \mathcal{R}_{h+1}$ we have $\mathcal{T}_h^{\mathbf{0}}(f_{h+1} + R_{h+1}) \in \mathcal{F}_h$. Similarly, we can show $\mathcal{T}_h^{\mathbf{0}} f_{h+1} \in \mathcal{F}_h$.

Moreover, for any $f_{h+1}, f_{h+1}' \in \mathcal{F}_{h+1}$, we can assume that $f_{h+1} = \langle \phi_{h+1}^{\mathrm{lc}}, \theta_{h+1}\rangle$ and $f_{h+1}' = \langle \phi_{h+1}^{\mathrm{lc}}, \theta_{h+1}'\rangle$, where $\|\theta_{h+1}\|_2, \|\theta_{h+1}'\|_2 \leq (H-h-2)\sqrt{d_{\mathrm{lc}}}$ and $\langle \phi_{h+1}^{\mathrm{lc}}(\cdot), \theta_{h+1}\rangle, \langle \phi_{h+1}^{\mathrm{lc}}(\cdot), \theta_{h+1}'\rangle \in [-(H-h-2), H-h-2]$. Therefore, we have $f_{h+1}(\cdot) - f_{h+1}'(\cdot) \in [-2(H-h-2), 2(H-h-2)]$. From linear completeness (Definition 6), we know that there exists $\theta_h''$ that satisfy $\langle \phi_h^{\mathrm{lc}}, \theta_h''\rangle = \mathcal{T}_h^{\mathbf{0}}(\langle \phi_{h+1}^{\mathrm{lc}}, \theta_{h+1} - \theta_{h+1}'\rangle)$ and $\|\theta_h''\|_2 \leq 2(H-h-2)\sqrt{d_{\mathrm{lc}}}$ with $\langle \phi_h^{\mathrm{lc}}(\cdot), \theta_h''\rangle \in [-2(H-h-2), 2(H-h-2)]$. Now, choosing $\theta_h = \theta_h''/2$ and $\theta_h' = -\theta_h''/2$, we know that $\langle \phi_h^{\mathrm{lc}}, \theta_h\rangle - \langle \phi_h^{\mathrm{lc}}, \theta_h'\rangle = \mathcal{T}_h^{\mathbf{0}}(f_{h+1} - f_{h+1}')$ and $\langle \phi_h^{\mathrm{lc}}, \theta_h\rangle, \langle \phi_h^{\mathrm{lc}}, \theta_h'\rangle \in \mathcal{F}_h$. Hence we have $\mathcal{T}_h^{\mathbf{0}}(f_{h+1} - f_{h+1}') \in \mathcal{F}_h - \mathcal{F}_h$.

Therefore, from the above discussions, we get that completeness holds.

Invoking Theorem 1, the covering number argument (Lemma 8), and the bound on Bellman Eluder dimension (Proposition 6), we know that the output policy is $\varepsilon$-optimal and the sample complexity is

$$\tilde{O}\left(\frac{\left(H^7 \log\left(\mathcal{N}_{\mathcal{F}}\left(\varepsilon/512H^2\sqrt{d_{\mathrm{lc}}}\iota\right)\right) + H^5 \log\left(\mathcal{N}_{\mathcal{R}}\left(\varepsilon/512H^2\sqrt{d_{\mathrm{lc}}}\iota\right)\right)\right) d_{\mathrm{lc}}^2 \iota^3}{\varepsilon^2}\right)$$

$$= \tilde{O}\left(\frac{H^8 d_{\mathrm{lc}}^3 \log(1/\delta)}{\varepsilon^2}\right). \qquad \square$$

As a final remark, Zanette et al. (2020b) assume $R$ is unknown but linear in $\phi^{\mathrm{lc}}$. In this case, we can instead construct $\mathcal{F}_{\mathrm{off}}(R) = \mathcal{F}_0(\{\phi^{\mathrm{lc}}\}, H) \times \ldots \times \mathcal{F}_{H-1}(\{\phi^{\mathrm{lc}}\}, 1)$, where the norm bound and the value range bound in $\mathcal{F}_h(\{\phi^{\mathrm{lc}}\}, H-h)$ are larger than that in $\mathcal{F}_h(\{\phi^{\mathrm{lc}}\}, H-h-1)$, thus capturing the reward-appended functions. One can easily follow the proof of Theorem 1 and get the sample complexity result when using this new $\mathcal{F}_{\mathrm{off}}(R)$ in the offline phase of RFOLIVE. This variant has the same sample complexity as Corollary 2.

### C.5 Q-type RFOLIVE for known representation linear MDPs

In this part, we instantiate the general theoretical guarantee of Q-type RFOLIVE (Theorem 1) to the linear MDP setting, where the transition dynamics satisfy the low-rank decomposition (Definition 8) and $\phi^{\mathrm{lr}}$ is known. We construct the function class $\mathcal{F}(\{\phi^{\mathrm{lr}}\})$ as $\mathcal{F}(\{\phi^{\mathrm{lr}}\}) = \mathcal{F}_0(\{\phi^{\mathrm{lr}}\}, H-1) \times \ldots \times \mathcal{F}_{H-1}(\{\phi^{\mathrm{lr}}\}, 0)$, where

$$\mathcal{F}_h(\{\phi^{\mathrm{lr}}\}, B_h) = \left\{f_h(x_h, a_h) = \langle \phi_h^{\mathrm{lr}}(x_h, a_h), \theta_h\rangle : \|\theta_h\|_2 \leq B_h\sqrt{d_{\mathrm{lr}}}, \langle \phi_h^{\mathrm{lr}}(\cdot), \theta_h\rangle \in [-B_h, B_h]\right\}.$$

In the following, we state the sample complexity result.

**Corollary 6** (Q-type RFOLIVE for linear MDPs). *Fix $\delta \in (0, 1)$. Consider an MDP $M$ that admits a low-rank factorization in Definition 8 and the feature $\phi^{\mathrm{lr}}$ is known, and we are given a reward function class $\mathcal{R}$. With probability at least $1 - \delta$, for any reward function $R \in \mathcal{R}$, running Q-type version of RFOLIVE (Algorithm 1) with $\mathcal{F} = \mathcal{F}(\{\phi^{\mathrm{lr}}\})$ outputs a policy $\hat{\pi}$ that satisfies $v_R^{\hat{\pi}} \geq v_R^* - \varepsilon$. The required number of episodes is*

$$\tilde{O}\left(\frac{\left(H^8 d_{\mathrm{lr}}^3 + H^5 d_{\mathrm{lr}}^2 \log(\mathcal{N}_{\mathcal{R}}(\varepsilon/512H^2\sqrt{d_{\mathrm{lr}}}\iota))\right)\log(1/\delta)}{\varepsilon^2}\right).$$

*In RFOLIVE, we set*

$$\varepsilon_{actv} = \frac{\varepsilon}{2H^2}, \varepsilon_{elim} = \frac{\varepsilon}{8H^2\sqrt{d_{\mathrm{lr}}}\iota}, n_{actv} = \frac{H^6\iota}{\varepsilon^2},$$

*and*

$$n_{elim} = \frac{(H^7 d_{\mathrm{lr}}^2 + H^4 d_{\mathrm{lr}} \log(\mathcal{N}_{\mathcal{R}}(\varepsilon_{elim}/64)))\iota^3}{\varepsilon^2} = \frac{(H^7 d_{\mathrm{lr}}^2 + H^4 d_{\mathrm{lr}} \log(\mathcal{N}_{\mathcal{R}}(\varepsilon/512H^2\sqrt{d_{\mathrm{lr}}}\iota)))\iota^3}{\varepsilon^2},$$

*where $\iota = c_4 \log(Hd_{\mathrm{lr}}/\delta\varepsilon)$ and $c_4$ is a large enough constant.*

**Remark** If we consider the entire linear reward class $\mathcal{R} = \mathcal{R}_0 \times \ldots \times \mathcal{R}_{H-1}$, where $\mathcal{R}_h = \{\langle \phi^{\mathrm{lr}}, \eta_h \rangle : \|\eta_h\|_2 \le \sqrt{d_{\mathrm{lr}}}, \langle \phi_h^{\mathrm{lr}}, \eta_h \rangle \in (\mathcal{X} \times \mathcal{A} \to [0,1])\}$, then invoking Corollary 6 and applying the similar covering argument of Lemma 8 on the entire linear reward function class $\mathcal{R}$ yields the sample complexity

$$\tilde{O}\left(\frac{H^8 d_{\mathrm{lr}}^3 \log(1/\delta)}{\varepsilon^2}\right).$$

Recently, Wagenmaker et al. (2022) improved the sharpest rate in the reward-free linear MDPs to $\tilde{O}\left(\frac{d_{\mathrm{lr}} H^5 (d_{\mathrm{lr}} + \log(1/\delta))}{\varepsilon^2} + \frac{d_{\mathrm{lr}}^{9/2} H^6 \log^{7/2}(1/\delta)}{\varepsilon}\right)$. Although the focus of our work is not to obtain the optimal rate, the sample complexity bound of RFOLIVE is also independent of $K$ and not much worse than the current state of the art. In the reward-aware setting, GOLF has a sharper rate than the subroutine OLIVE under the completeness type assumption (Jin et al., 2021). Since in RFOLIVE we only collect data when running a single (zero) reward OLIVE during the online phase and completeness (Assumption 2) is satisfied in our paper, we believe that there also exists a reward-free version of GOLF (by running GOLF with zero reward function in the online phase and performing function elimination in the offline phase) that can potentially improve an $H d_{\mathrm{qbe}}$ factor compared with RFOLIVE, thus matching the optimal $d_{\mathrm{lr}}$ dependence in linear MDPs.

*Proof.* Similar as the proof of Corollary 4, we can verify that Assumption 1 and Assumption 2 hold. Invoking Theorem 1 and noticing that the covering number argument (Lemma 8) and the bound on Q-type Bellman Eluder dimension (Proposition 5) completes the proof. $\square$

## D  V-type RFOLIVE results

In this section, we present the results related to V-type RFOLIVE. In Appendix D.1, we provide the theoretical guarantee of V-type OLIVE (Jiang et al., 2017; Jin et al., 2021) for completeness. In Appendix D.2, we show the detailed proof of the sample complexity bound of V-type RFOLIVE (Theorem 3). In Appendix D.3, we discuss the instantiation of V-type RFOLIVE to low-rank MDPs.

### D.1  V-type OLIVE

First, we state the sample complexity of V-type OLIVE. Similar as Q-type OLIVE, since we consider the uniformly bounded reward setting ($0 \le r_h \le 1$) instead of bounded total reward setting ($\forall h \in [H], r_h \ge 0$ and $\sum_{h=0}^{H-1} r_h \le 1$), we need to pay an additional $H^2$ dependency in $n_{\mathrm{actv}}$ and $n_{\mathrm{elim}}$.

**Proposition 2** (Sample complexity of V-type OLIVE, modification of Theorem 23 in Jin et al. (2021)). *Assume ($\frac{\varepsilon}{128 H \sqrt{d_{\mathrm{vbe}}}} = \varepsilon_{elim}/8$) single-reward approximate realizability holds for $\mathcal{F}$ in Assumption 5 and $\mathcal{F}$ is finite. If we set*

$$\varepsilon_{actv} = \frac{\varepsilon}{4H}, \ \varepsilon_{elim} = \frac{\varepsilon}{16 H \sqrt{d_{\mathrm{vbe}}}}, n_{actv} = \frac{H^4 \iota}{\varepsilon^2}, \ and \ n_{elim} = \frac{H^4 d_{\mathrm{vbe}} K \log(|\mathcal{F}|) \iota}{\varepsilon^2}$$

*where $d_{\mathrm{vbe}} = \dim_{\mathrm{vbe}}^R(\mathcal{F}, \mathcal{D}_{\mathcal{F}}, \varepsilon/8H)$ and $\iota = c_5 \log(H d_{\mathrm{vbe}} K/\delta\varepsilon)$, then with probability at least $1 - \delta$, V-type OLIVE (Algorithm 4 in Jin et al. (2021)) with $\mathcal{F}$ will output an $\varepsilon$-optimal policy (under a single reward $R$) using at most $O(d_{\mathrm{vbe}} H(n_{actv} + n_{elim}))$ episodes. Here $c_5$ is a large enough constant.*

### D.2  Proof of V-type RFOLIVE under general function approximation

In this part, we first provide the general statement of Theorem 3 and then show the detailed proof.

**Theorem** (Full version of Theorem 3). *Fix $\delta \in (0, 1)$. Given a reward class $\mathcal{R}$ and a function class $\mathcal{F}$ that satisfies Assumption 1 and Assumption 2, with probability at least $1 - \delta$, for any $R \in \mathcal{R}$, V-type RFOLIVE (Algorithm 1) with $\mathcal{F}$ outputs a policy $\hat{\pi}$ that satisfies $v_R^{\hat{\pi}} \ge v_R^* - \varepsilon$. The required number of episodes is*

$$O\left(\frac{\left(H^7 \log\left(\mathcal{N}_{\mathcal{F}}\left(\varepsilon/2048 H^2 \sqrt{d_{\mathrm{vbe}}}\right)\right) + H^5 \log\left(\mathcal{N}_{\mathcal{R}}\left(\varepsilon/2048 H^2 \sqrt{d_{\mathrm{vbe}}}\right)\right)\right) d_{\mathrm{vbe}}^2 K \iota}{\varepsilon^2}\right).$$

*In* RFOLIVE, *we set*

$$\varepsilon_{actv} = \frac{\varepsilon}{8H^2}, \ \varepsilon_{elim} = \frac{\varepsilon}{32H^2\sqrt{d_{\mathrm{vbe}}}}, \ n_{actv} = \frac{H^6\iota}{\varepsilon^2},$$

*and*

$$\begin{aligned}
n_{elim} &= \frac{(H^6\log\left(\mathcal{N}_{\mathcal{F}}(\varepsilon_{elim}/64)\right) + H^4\log\left(\mathcal{N}_{\mathcal{R}}(\varepsilon_{elim}/64)\right))d_{\mathrm{vbe}}K\iota}{\varepsilon^2} \\
&= \frac{\left(H^6\log\left(\mathcal{N}_{\mathcal{F}}\left(\varepsilon/2048H^2\sqrt{d_{\mathrm{vbe}}}\right)\right) + H^4\log\left(\mathcal{N}_{\mathcal{R}}\left(\varepsilon/2048H^2\sqrt{d_{\mathrm{vbe}}}\right)\right)\right)d_{\mathrm{vbe}}K\iota}{\varepsilon^2},
\end{aligned}$$

*where* $d_{\mathrm{vbe}} = \dim_{\mathrm{vbe}}^{\mathbf{0}}(\mathcal{F} - \mathcal{F}, \mathcal{D}_{\mathcal{F}-\mathcal{F}}, \varepsilon/(8H))$, $\iota = c_6\log(Hd_{\mathrm{vbe}}K/\delta\varepsilon)$ *and* $c_6$ *is a large enough constant.*

*Proof.* This proof follows the similar structure as the proof of Theorem 1. The major difference is now we consider a discretized function class $\mathscr{Z}_{\mathrm{on}}$ in the online phase and consider a class of policy $\Pi_{\mathrm{on}}$ in the offline elimination.

When we construct $\mathscr{Z}_{\mathrm{on}}$ (an $(\varepsilon_{\mathrm{elim}}/64)$-cover of $\mathcal{F}_{\mathrm{on}}$), w.l.o.g, we can assume $\mathbf{0} \in \mathscr{Z}_{\mathrm{on}}$, therefore the approximate realizability (Assumption 5) holds. From the online phase of V-type RFOLIVE (Algorithm 1), we can see that this phase is equivalent to running V-type RFOLIVE (Algorithm 4 in Jin et al. (2021)) with the input function class $\mathscr{Z}_{\mathrm{on}}$, the specified parameters $\varepsilon_{\mathrm{actv}}, \varepsilon_{\mathrm{elim}}, n_{\mathrm{elim}}, n_{\mathrm{actv}}$, and under the reward function $R = \mathbf{0}$. Then the sample complexity is immediately from our specified values of $\varepsilon_{\mathrm{actv}}, \varepsilon_{\mathrm{elim}}, n_{\mathrm{actv}}, n_{\mathrm{elim}}$ and Proposition 2 as we only collect samples in the online phase. Notice that we have the bound $\log(|\mathscr{Z}_{\mathrm{on}}|) \leq 2\log\left(\mathcal{N}_{\mathcal{F}}(\varepsilon_{\mathrm{elim}}/64)\right)$ and such a constant 2 is absorbed by large enough $c_6$. Therefore, it remains to show that the algorithm can indeed output an $\varepsilon$-optimal policy with probability $1 - \delta$ in the offline phase. We will show the following three claims hold with probability at least $1 - \delta$.

**Claim 1** For any $g \in \mathcal{F}_{\mathrm{off}}(R)$, if $\exists h \in [H]$, s.t. $|\mathcal{E}_{\mathrm{V}}^R(g, \pi_g, h)| \geq \varepsilon/H$, then it will be eliminated in the offline phase.

**Claim 2** $Q_R^* \in \mathcal{F}_{\mathrm{off}}(R)$ and $Q_R^*$ will not be eliminated in the offline phase.

**Claim 3** At the end of the offline phase, picking the optimistic function from the survived value functions gives us $\varepsilon$-optimal policy.

Before showing these three claims, we first state show properties from the online phase of V-type RFOLIVE and the concentration results in the offline phase.

**Properties from the online phase of V-type RFOLIVE** From the equivalence between the online phase of V-type RFOLIVE (Algorithm 1) and V-type OLIVE (Algorithm 4 in Jin et al. (2021)) with reward $\mathbf{0}$, we know that with probability at least $1 - \delta/4$, the online phase terminates within $d_{\mathrm{vbe}}H + 1$ iterations. In addition, with probability at least $1 - \delta/4$, the following properties (Eq. (10) and Eq. (11)) hold for the first $d_{\mathrm{vbe}}H + 1$ iterations:

(i) When the online phase exists at iteration $T$ in line 7 (i.e., the elimination procedure is not activated in RFOLIVE), for any $f \in \mathcal{F}^T$, it predicts no more than $\varepsilon/(2H)$ value:

$$V_f(x_0) \leq V_{f^T}(x_0) = V_{f^T}(x_0) - V_{\mathbf{0},0}^{\pi_{f^T}}(x_0) = \sum_{h=0}^{H-1}\mathcal{E}_{\mathrm{V}}^R(f^T, \pi^T, h) < 2H\varepsilon_{\mathrm{actv}} < \varepsilon/(2H). \quad (10)$$

The first equality is due to any policy evaluation has value 0 under under the reward function $\mathbf{0}$. The second equality is due to the policy loss decomposition in Jiang et al. (2017). The second inequality is adapted from the "concentration in the activation procedure" part of the proof for Theorem 23 in Jin et al. (2021).

(ii) For $T \leq d_{\mathrm{vbe}}H + 1$, the concentration argument holds for any $f \in \mathscr{Z}_{\mathrm{on}}$ and $t \in [T]$:

$$\left|\hat{\mathcal{E}}_{\mathrm{V}}^{\mathbf{0}}(f, \pi^t, h^t) - \mathcal{E}_{\mathrm{V}}^{\mathbf{0}}(f, \pi^t, h^t)\right| < \varepsilon_{\mathrm{elim}}/8. \quad (11)$$

This is from the "concentration in the elimination procedure" step of the proof for Theorem 23 in Jin et al. (2021) and we adapt it with our parameters.

**Concentration results in the offline phase** Let $\overline{\mathcal{R}}$ be an $(\varepsilon_{\text{elim}}/64)$-cover of $\mathcal{R}$. For every $R \in \mathcal{R}$, let $R^c = \operatorname{argmin}_{R' \in \overline{\mathcal{R}}} \max_{h \in [H]} \|R_h - R'_h\|_\infty$. First consider any fixed $\pi' \in \Pi_{\text{on}}$ and $R \in \overline{\mathcal{R}}$. Let $\mathcal{Z}(R)$ be an $(\varepsilon_{\text{elim}}/64)$-cover of $\mathcal{F}_{\text{off}}(R)$ with cardinality $\mathcal{N}_{\mathcal{F}_{\text{off}}(R)}(\varepsilon_{\text{elim}}/64) = \mathcal{N}_{\mathcal{F}}(\varepsilon_{\text{elim}}/64)$. For every $g \in \mathcal{F}_{\text{off}}(R)$, let $g^c = \operatorname{argmin}_{g' \in \mathcal{Z}(R)} \max_{h \in [H]} \|g_h - g'_h\|_\infty$. Then we consider any fixed $(t, g') \in [T] \times \mathcal{Z}(R)$ and calculate the upper bound of the second moment for $g$

$$\frac{\mathbf{1}\left[a_{h^t}^{(i)} = \pi'_{h^t}(x_{h^t}^{(i)})\right]}{1/K}\left(g'_{h^t}\left(x_{h^t}^{(i)}, a_{h^t}^{(i)}\right) - r_{h^t}^{(i)} - V_{g'}\left(x_{h^t+1}^{(i)}\right)\right).$$

Let $y(x_{h^t}, a_{h^t}, r_{h^t}, x_{h^t+1}) = g'_{h^t}\left(x_{h^t}^{(i)}, a_{h^t}^{(i)}\right) - r_{h^t}^{(i)} - V_{g'}\left(x_{h^t+1}^{(i)}\right) \subseteq [-2H, 2H]$, then we have

$$\mathbb{E}\left[\left(K\mathbf{1}\left[a_{h^t}^{(i)} = \pi'_{h^t}\left(x_{h^t}^{(i)}\right)\right]y(x_{h^t}, a_{h^t}, r_{h^t}, x_{h^t+1})\right)^2 \mid x_{h^t} \sim \pi^t, a_{h^t} \sim \text{unif}(\mathcal{A})\right]$$

$$\leq 4H^2 K^2 \mathbb{E}\left[\mathbf{1}\left[a_{h^t}^{(i)} = \pi'_{h^t}\left(x_{h^t}^{(i)}\right)\right] \mid x_{h^t} \sim \pi^t, a_{h^t} \sim \text{unif}(\mathcal{A})\right] = 4H^2 K.$$

Applying Bernstein's inequality and noticing the variance of the random variable is upper bounded by the second moment, with probability at least $1 - \frac{\delta}{2T\mathcal{N}_{\mathcal{F}}(\varepsilon_{\text{elim}}/64)\mathcal{N}_{\mathcal{R}}(\varepsilon_{\text{elim}}/64)|\Pi_{\text{on}}|}$ we have

$$\left|\hat{\mathcal{E}}^R(g', \pi^t, \pi', h^t) - \mathcal{E}^R(g', \pi^t, \pi', h^t)\right|$$

$$\leq \sqrt{\frac{4H^2 K \log(4T\mathcal{N}_{\mathcal{F}}(\varepsilon_{\text{elim}}/64)\mathcal{N}_{\mathcal{R}}(\varepsilon_{\text{elim}}/64)|\Pi_{\text{on}}|/\delta)}{n_{\text{elim}}}}$$

$$+ \frac{4HK \log(4T\mathcal{N}_{\mathcal{F}}(\varepsilon_{\text{elim}}/64)\mathcal{N}_{\mathcal{R}}(\varepsilon_{\text{elim}}/64)|\Pi_{\text{on}}|/\delta)}{3n_{\text{elim}}}$$

$$< \frac{\varepsilon_{\text{elim}}}{8}.$$

The second inequality follows from $\varepsilon_{\text{elim}} = \varepsilon / \left(32H^2\sqrt{d_{\text{vbe}}}\right)$, $\iota = c_6 \log(Hd_{\text{vbe}}K/\delta\varepsilon)$, and

$$n_{\text{elim}} = \frac{(H^6 \log\left(\mathcal{N}_{\mathcal{F}}(\varepsilon_{\text{elim}}/64)\right) + H^4 \log\left(\mathcal{N}_{\mathcal{R}}(\varepsilon_{\text{elim}}/64)\right))d_{\text{vbe}}K \cdot \iota}{\varepsilon^2}$$

with $c_6$ in $\iota$ being chosen large enough. Here we also notice that $|\Pi_{\text{on}}| = |\mathcal{Z}_{\text{on}}|$ and $\log\left(|\mathcal{Z}_{\text{on}}|\right) = \log\left(\mathcal{N}_{\mathcal{F}_{\text{on}}}(\varepsilon_{\text{elim}}/64)\right) \leq 2\log\left(\mathcal{N}_{\mathcal{F}}(\varepsilon_{\text{elim}}/64)\right)$.

Union bounding over $(t, g') \in [T] \times \mathcal{Z}(R)$, with probability at least $1 - \frac{\delta}{2\mathcal{N}_{\mathcal{R}}(\varepsilon_{\text{elim}}/64)|\Pi_{\text{on}}|}$, we have that for any fixed $\pi' \in \Pi_{\text{on}}, R \in \overline{\mathcal{R}}$ and all $g' \in \mathcal{Z}(R), t \in [T]$

$$\left|\hat{\mathcal{E}}^R(g', \pi^t, \pi', h^t) - \mathcal{E}^R(g', \pi^t, \pi', h^t)\right| < \varepsilon_{\text{elim}}/8.$$

Union bounding over $\pi' \in \Pi_{\text{on}}, R \in \overline{\mathcal{R}}$, we have that with probability at least $1 - \delta/2$, for all $R \in \overline{\mathcal{R}}, \pi' \in \Pi_{\text{on}}, g \in \mathcal{F}_{\text{off}}(R), t \in [T]$,

$$\left|\hat{\mathcal{E}}^R\left(g, \pi^t, \pi, h^t\right) - \mathcal{E}^R\left(g, \pi^t, \pi', h^t\right)\right| < \varepsilon_{\text{elim}}/8.$$

Therefore, with probability at least $1 - \delta/2$, for all $R \in \mathcal{R}, \pi' \in \Pi_{\text{on}}, g \in \mathcal{F}_{\text{off}}(R), t \in [T]$, we have

$$\left|\hat{\mathcal{E}}^R\left(g, \pi^t, \pi', h^t\right) - \mathcal{E}^R\left(g, \pi^t, \pi', h^t\right)\right|$$

$$\leq \left|\hat{\mathcal{E}}^R\left(g, \pi^t, \pi', h^t\right) - \hat{\mathcal{E}}^{R^c}\left(g, \pi^t, \pi', h^t\right)\right| + \left|\hat{\mathcal{E}}^{R^c}\left(g, \pi^t, \pi', h^t\right) - \mathcal{E}^{R^c}\left(g, \pi^t, \pi', h^t\right)\right|$$

$$+ \left|\mathcal{E}^{R^c}\left(g, \pi^t, \pi', h^t\right) - \mathcal{E}^R\left(g, \pi^t, \pi', h^t\right)\right|$$

$$\leq \varepsilon_{\text{elim}}/64 + \varepsilon_{\text{elim}}/8 + \varepsilon_{\text{elim}}/64$$

$$< \varepsilon_{\text{elim}}/4. \tag{12}$$

All statements in our subsequent proof are under the event that all the different high-probability events (the online phase terminates within $d_{\text{vbe}}H + 1$ iterations, and Eq. (10), Eq. (11), Eq. (12) hold for the first $d_{\text{vbe}}H + 1$ iterations) discussed above hold with a total failure probability of $\delta$.

**Proof of Claim 1**   We consider any $g \in \mathcal{F}_{\mathrm{off}}(R)$ that satisfies $\exists h \in [H]$, such that $|\mathcal{E}_{\mathrm{V}}^R(g, \pi_g, h)| \geq \varepsilon/H$. Recall the definition of $\mathcal{F}_{\mathrm{off}}(R)$, we know that $g$ can be written as $g = (g_0, \ldots, g_{H-1}) = (f_0 + R_0, \ldots, f_{H-1} + R_{H-1}) = (f_0 + R_0, \ldots, f_{H-1} + R_{H-1})$. We will discuss the positive average Bellman error and the negative average Bellman error case separately.

**Case (i) of Claim 1**   $\mathcal{E}_{\mathrm{V}}^R(g, \pi_g, h) = \mathbb{E}[g_h(x_h, a_h) - R_h(x_h, a_h) - V_g(x_{h+1}) \mid a_{0:h} \sim \pi_g] \geq \varepsilon/H.$

Since $\mathcal{E}_{\mathrm{V}}^R(g, \pi_g, h) = \mathcal{E}_{\mathrm{Q}}^R(g, \pi_g, h)$, similar as the proof of Theorem 1, we know that

$$\varepsilon/H \leq \mathbb{E}[\tilde{f}_h(x_h, a_h) \mid a_{0:h} \sim \pi_g]. \qquad (\tilde{f}_h := f_h - \mathcal{T}_h^{\mathbf{0}} g_{h+1})$$

Same as in the proof of Theorem 1, here we construct a function $\tilde{f}$ as

$$\tilde{f}_{h'}(x_{h'}, a_{h'}) = \begin{cases} (\mathcal{T}_{h'}^{\mathbf{0}} \tilde{f}_{h'+1})(x_{h'}, a_{h'}) = \mathbb{E}[\max_a \tilde{f}_{h'+1}(x_{h'+1}, a) \mid x_{h'}, a_{h'}] & 0 \leq h' \leq h-1 \\ f_h(x_h, a_h) - (\mathcal{T}_h^{\mathbf{0}} g_{h+1})(x_h, a_h) & h' = h \\ 0 & h+1 \leq h' \leq H-1. \end{cases}$$

From the definition of V-type average Bellman error and the construction, we know that for any policy $\pi$ we can translate the zero reward V-type average Bellman error of a function $\tilde{f} \in \mathcal{F}_{\mathrm{on}}$ with roll-in policy $\pi$ to the average Bellman error under $R$ for a function $g \in \mathcal{F}_{\mathrm{off}}(R)$ with roll-in policy $\pi_{0:h-1} \circ \pi_{\tilde{f}, h}$ (Definition 4) as the following

$$\begin{aligned} &\mathcal{E}_{\mathrm{V}}^{\mathbf{0}}(\tilde{f}, \pi, h) \\ &= \mathbb{E}\left[\tilde{f}_h(x_h, a_h) - \mathbf{0} - V_{\tilde{f}}(x_{h+1}) \mid a_{0:h-1} \sim \pi, a_h \sim \pi_{\tilde{f}}\right] \\ &= \mathbb{E}\left[\tilde{f}_h(x_h, a_h) \mid a_{0:h-1} \sim \pi, a_h \sim \pi_{\tilde{f}}\right] \\ &= \mathbb{E}\left[f_h(x_h, a_h) - (\mathcal{T}_h^{\mathbf{0}} g_{h+1})(x_h, a_h) \mid a_{0:h-1} \sim \pi, a_h \sim \pi_{\tilde{f}}\right] \\ &= \mathbb{E}\left[g_h(x_h, a_h) - R_h(x_h, a_h) - V_g(x_{h+1}) \mid a_{0:h-1} \sim \pi, a_h \sim \pi_{\tilde{f}}\right] \\ &= \mathcal{E}^R(g, \pi, \pi_{\tilde{f}}, h) \end{aligned} \qquad (13)$$

where the second equality is due to $\tilde{f}_{h+1} = \mathbf{0}$.

As the construction of $\tilde{f}$ and the assumptions of $\mathcal{F}$ are the same as that in Q-type RFOLIVE and we use the same $\mathcal{F}_{\mathrm{on}} = \mathcal{F} - \mathcal{F}$ in both places, following the same proof of Theorem 1 directly gives us that $\tilde{f} = (\tilde{f}_0, \ldots, \tilde{f}_{H-1}) \in \mathcal{F}_{\mathrm{on}}$ and $V_{\tilde{f}}(x_0) \geq \varepsilon/H$.

Since in the online phase we use $\mathcal{Z}_{\mathrm{on}}$, which is an $(\varepsilon_{\mathrm{elim}}/64)$-cover of $\mathcal{F}_{\mathrm{on}}$, we know that there exists $\tilde{f}^{\mathrm{c}} \in \mathcal{Z}_{\mathrm{on}}$ such that $\max_{h' \in [H]} \|\tilde{f}_{h'} - \tilde{f}_{h'}^{\mathrm{c}}\|_\infty \leq \varepsilon_{\mathrm{elim}}/64 \leq \varepsilon/(2H)$. Notice that since $\forall h+1 \leq h' \leq H-1$ we have $\mathbf{0} \in \mathcal{Z}_{h'}$ and $f_{h'} = \mathbf{0}$, thus w.l.o.g., we can assume that $\tilde{f}_{h'}^{\mathrm{c}} = \mathbf{0}, \forall h+1 \leq h' \leq H-1$.

From the definition of $\tilde{f}^{\mathrm{c}}$ and $\tilde{f}_0(x_0, \pi_{\tilde{f}}(x_0)) = V_{\tilde{f}}(x_0) \geq \varepsilon/H$, we have

$$V_{\tilde{f}^{\mathrm{c}}}(x_0) = \tilde{f}_0^{\mathrm{c}}(x_0, \pi_{\tilde{f}^{\mathrm{c}}}(x_0)) \geq \tilde{f}_0^{\mathrm{c}}(x_0, \pi_{\tilde{f}}(x_0)) \geq \tilde{f}_0(x_0, \pi_{\tilde{f}}(x_0)) - \varepsilon/(2H) \geq \varepsilon/(2H).$$

From the first property of the online phase (Eq. (10)), we know that all the survived value functions at the end of the online phase predict no more than $\varepsilon/(2H)$. Therefore $\tilde{f}^{\mathrm{c}}$ will be eliminated. We assume it is eliminated at iteration $t$ by policy $\pi^t$ in level $h^t$.

For any policy $\pi$ and $h' \in [H], h' \neq h$, we have

$$\begin{aligned} &\mathcal{E}_{\mathrm{V}}^{\mathbf{0}}\left(\tilde{f}^{\mathrm{c}}, \pi, h'\right) \\ &= \mathbb{E}[\tilde{f}_{h'}^{\mathrm{c}}(x_{h'}, a_{h'}) - \mathbf{0} - \tilde{f}_{h'+1}^{\mathrm{c}}(x_{h'+1}, a_{h'+1}) \mid a_{0:h'-1} \sim \pi, a_{h':h'+1} \sim \pi_{\tilde{f}^{\mathrm{c}}}] \\ &\geq \mathbb{E}[\tilde{f}_{h'}(x_{h'}, a_{h'}) - \tilde{f}_{h'+1}(x_{h'+1}, a_{h'+1}) \mid a_{0:h'-1} \sim \pi, a_{h':h'+1} \sim \pi_{\tilde{f}^{\mathrm{c}}}] - 2\varepsilon_{\mathrm{elim}}/64 \\ &\geq \mathbb{E}[\tilde{f}_{h'}(x_{h'}, a_{h'}) - \tilde{f}_{h'+1}(x_{h'+1}, a_{h'+1}) \mid a_{0:h'-1} \sim \pi, a_{h'} \sim \pi_{\tilde{f}^{\mathrm{c}}}, a_{h'+1} \sim \pi_{\tilde{f}}] - 2\varepsilon_{\mathrm{elim}}/64 \\ &= -\varepsilon_{\mathrm{elim}}/32. \end{aligned}$$

The first inequality is from the definition of $\tilde{f}^c$. The second inequality is due to $\pi_{\tilde{f}}$ is the greedy policy of $\tilde{f}$. The last equality is due to the construction of $\tilde{f}$.

Similarly, on the other end, we also have

$$
\begin{aligned}
&\mathcal{E}_V^0 \left( \tilde{f}^c, \pi, h' \right) \\
&= \mathbb{E}[\tilde{f}_{h'}^c(x_{h'}, a_{h'}) - \mathbf{0} - \tilde{f}_{h'+1}^c(x_{h'+1}, a_{h'+1}) \mid a_{0:h'-1} \sim \pi, a_{h':h'+1} \sim \pi_{\tilde{f}^c}] \\
&\leq \mathbb{E}[\tilde{f}_{h'}^c(x_{h'}, a_{h'}) - \tilde{f}_{h'+1}^c(x_{h'+1}, a_{h'+1}) \mid a_{0:h'-1} \sim \pi, a_{h'} \sim \pi_{\tilde{f}^c}, a_{h'+1} \sim \pi_{\tilde{f}}] \\
&\leq \mathbb{E}[\tilde{f}_{h'}(x_{h'}, a_{h'}) - \tilde{f}_{h'+1}(x_{h'+1}, a_{h'+1}) \mid a_{0:h'-1} \sim \pi, a_{h'} \sim \pi_{\tilde{f}^c}, a_{h'+1} \sim \pi_{\tilde{f}}] + 2\varepsilon_{\mathrm{elim}}/64 \\
&= \varepsilon_{\mathrm{elim}}/32.
\end{aligned}
$$

Therefore, for any policy $\pi$ and $h' \in [H], h' \neq h$ we get

$$
\left| \mathcal{E}_V^0 \left( \tilde{f}^c, \pi, h' \right) \right| \leq \varepsilon_{\mathrm{elim}}/32. \tag{14}
$$

From the Bellman backup construction of $\tilde{f}^c$, we know that $\tilde{f}^c$ can only be eliminated at level $h$. This can be seen from the following argument: Applying the concentration result of the online phase (Eq. (11)), we have $|\hat{\mathcal{E}}_V^0(\tilde{f}^c, \pi^t, h^t) - \mathcal{E}_V^0(\tilde{f}^c, \pi^t, h^t)| \leq 3\varepsilon_{\mathrm{elim}}/4$. Further notice that Eq. (14), we have $|\hat{\mathcal{E}}_V^0(\tilde{f}^c, \pi^t, h^t)| \leq 3\varepsilon_{\mathrm{elim}}/4 + \varepsilon_{\mathrm{elim}}/32$ if $h^t \neq h$. Since the elimination threshold is set to $\varepsilon_{\mathrm{elim}}$, $\tilde{f}^c$ will not be eliminated at level $h^t \neq h$.

This implies that at some iteration $t$ in the online phase, we will collect some $\pi^t$ that eliminates $\tilde{f}^c$ at level $h$, i.e., it satisfies $|\hat{\mathcal{E}}_V^0(\tilde{f}^c, \pi^t, h^t)| > \varepsilon_{\mathrm{elim}}$ and $h^t = h$. Applying concentration argument for the online phase (Eq. (11)), we have $|\hat{\mathcal{E}}_V^0(\tilde{f}^c, \pi^t, h^t) - \mathcal{E}_V^0(\tilde{f}^c, \pi^t, h^t)| \leq 3\varepsilon_{\mathrm{elim}}/16$. Therefore,

$$
|\mathcal{E}_V^0(\tilde{f}^c, \pi^t, h^t)| > 13\varepsilon_{\mathrm{elim}}/16. \tag{15}
$$

From the definition of the average Bellman error and $\tilde{f}, \tilde{f}^c$, we have the following equations

$$
\begin{aligned}
\mathcal{E}_V^0(\tilde{f}^c, \pi^t, h^t) &= \mathbb{E}\left[ \tilde{f}_h^c(x_h, a_h) - \mathbf{0} - \tilde{f}_{h+1}^c(x_{h+1}, a_{h+1}) \mid a_{0:h-1} \sim \pi^t, a_{h:h+1} \sim \pi_{\tilde{f}^c} \right] \\
&= \mathbb{E}\left[ \tilde{f}_h^c(x_h, a_h) \mid a_{0:h-1} \sim \pi^t, a_h \sim \pi_{\tilde{f}^c} \right], \qquad (\tilde{f}_{h+1}^c = \mathbf{0})
\end{aligned}
$$

and

$$
\begin{aligned}
&\mathcal{E}^R(g, \pi^t, \pi', h^t) \\
&= \mathbb{E}\left[ g_h(x_h, a_h) - R_h(x_h, a_h) - V_g(x_{h+1}) \mid a_{0:h-1} \sim \pi^t, a_h \sim \pi' \right] \\
&= \mathbb{E}\left[ \tilde{f}_h(x_h, a_h) \mid a_{0:h-1} \sim \pi^t, a_h \sim \pi' \right] \qquad \text{(Eq. (13))} \\
&\geq \mathbb{E}\left[ \tilde{f}_h^c(x_h, a_h) \mid a_{0:h-1} \sim \pi^t, a_h \sim \pi' \right] - 2\varepsilon_{\mathrm{elim}}/64,
\end{aligned}
$$

where $\pi' \in \Pi_{\mathrm{est}}^t = \Pi_{\mathrm{on}}$. Because $\pi_{\tilde{f}^c}$ is the greedy policy of $\tilde{f}^c$ and $\tilde{f}^c \in \mathcal{Z}_{\mathrm{on}}$, we know that in the offline phase $\pi' = \pi_{\tilde{f}^c} \in \Pi_{\mathrm{on}}$ will be chosen for elimination. Then we get

$$
\begin{aligned}
&\mathcal{E}^R \left( g, \pi^t, \pi_{\tilde{f}^c}, h^t \right) \\
&\geq \mathbb{E}\left[ \tilde{f}_h^c(x_h, a_h) \mid a_{0:h-1} \sim \pi^t, a_h \sim \pi_{\tilde{f}^c} \right] - 2\varepsilon_{\mathrm{elim}}/64 \\
&= \mathbb{E}\left[ \tilde{f}_h^c(x_h, a_h) - \mathbf{0} - \tilde{f}_{h+1}^c(x_{h+1}, a_{h+1}) \mid a_{0:h-1} \sim \pi^t, a_{h:h+1} \sim \pi_{\tilde{f}^c} \right] - 2\varepsilon_{\mathrm{elim}}/64 \\
&= \mathcal{E}_V^0(\tilde{f}^c, \pi^t, h^t) - \varepsilon_{\mathrm{elim}}/32,
\end{aligned}
$$

where the first equality is due to $\tilde{f}_{h+1}^c = \mathbf{0}$.

Similarly we have

$$\mathcal{E}^R\left(g, \pi^t, \pi_{\tilde{f}^c}, h^t\right)$$

$$\leq \mathbb{E}\left[\tilde{f}_h^c(x_h, a_h) \mid a_{0:h-1} \sim \pi^t, a_h \sim \pi_{\tilde{f}^c}\right] + 2\varepsilon_{\text{elim}}/64$$

$$= \mathbb{E}\left[\tilde{f}_h^c(x_h, a_h) - \mathbf{0} - \tilde{f}_{h+1}^c(x_{h+1}, a_{h+1}) \mid a_{0:h-1} \sim \pi^t, a_{h:h+1} \sim \pi_{\tilde{f}^c}\right] + 2\varepsilon_{\text{elim}}/64$$

$$= \mathcal{E}_{\text{V}}^{\mathbf{0}}(\tilde{f}^c, \pi^t, h^t) + \varepsilon_{\text{elim}}/32.$$

Further, using the concentration argument for the offline phase (Eq. (12)), we get

$$\left|\hat{\mathcal{E}}^R\left(g, \pi^t, \pi_{\tilde{f}^c}, h^t\right) - \mathcal{E}^R\left(g, \pi^t, \pi_{\tilde{f}^c}, h^t\right)\right| < \varepsilon_{\text{elim}}/4.$$

Hence, if $\mathcal{E}_{\text{V}}^{\mathbf{0}}(\tilde{f}^c, \pi^t, h^t) \geq 0$, we get $\mathcal{E}_{\text{V}}^{\mathbf{0}}(\tilde{f}^c, \pi^t, h^t) \geq 13\varepsilon_{\text{elim}}/16$ from Eq. (15), which yields

$$\hat{\mathcal{E}}^R\left(g, \pi^t, \pi_{\tilde{f}^c}, h^t\right) > \mathcal{E}^R\left(g, \pi^t, \pi_{\tilde{f}^c}, h^t\right) - \varepsilon_{\text{elim}}/4$$

$$\geq \mathcal{E}_{\text{V}}^{\mathbf{0}}(\tilde{f}^c, \pi^t, h^t) - \varepsilon_{\text{elim}}/32 - \varepsilon_{\text{elim}}/4$$

$$\geq 13\varepsilon_{\text{elim}}/16 - \varepsilon_{\text{elim}}/32 - \varepsilon_{\text{elim}}/4 > \varepsilon_{\text{elim}}/2.$$

Otherwise, we are in the case $\mathcal{E}_{\text{V}}^{\mathbf{0}}(\tilde{f}^c, \pi^t, h^t) < 0$ and we have $\mathcal{E}_{\text{V}}^{\mathbf{0}}(\tilde{f}^c, \pi^t, h^t) < -13\varepsilon_{\text{elim}}/16$ from Eq. (15). This yields

$$\hat{\mathcal{E}}^R\left(g, \pi^t, \pi_{\tilde{f}^c}, h^t\right) < \mathcal{E}^R\left(g, \pi^t, \pi_{\tilde{f}^c}, h^t\right) + \varepsilon_{\text{elim}}/4$$

$$\leq \mathcal{E}_{\text{V}}^{\mathbf{0}}(\tilde{f}^c, \pi^t, h^t) + \varepsilon_{\text{elim}}/32 + \varepsilon_{\text{elim}}/4$$

$$\leq -13\varepsilon_{\text{elim}}/16 + \varepsilon_{\text{elim}}/32 + \varepsilon_{\text{elim}}/4 < -\varepsilon_{\text{elim}}/2.$$

Thus we always have $\left|\hat{\mathcal{E}}^R\left(g, \pi^t, \pi_{\tilde{f}^c}, h^t\right)\right| > \varepsilon_{\text{elim}}/2$, which implies that we eliminate such $g$ by $\pi_{0:h^t-1}^t \circ \pi_{\tilde{f}^c, h^t}$ in the offline phase.

**Case (ii) of Claim 1** $\mathcal{E}_{\text{V}}^R(g, \pi_g, h) = \mathbb{E}[g_h(x_h, a_h) - R_h(x_h, a_h) - V_g(x_{h+1}) \mid a_{0:h} \sim \pi_g] \leq -\varepsilon/H$.

Same as before, we have $\mathcal{E}_{\text{V}}^R(g, \pi_g, h) = \mathbb{E}[f_h(x_h, a_h) - (\mathcal{T}_h^{\mathbf{0}} g_{h+1})(x_h, a_h) \mid a_{0:h} \sim \pi_g] \leq -\varepsilon/H$. Now we let $\tilde{f}_h$ be the negated version of the one in case (i), and define $\tilde{f}$ as

$$\tilde{f}_{h'}(x_{h'}, a_{h'}) = \begin{cases} (\mathcal{T}_{h'}^{\mathbf{0}} \tilde{g}_{h'+1})(x_{h'}, a_{h'}) = \mathbb{E}[\max_a \tilde{g}_{h'+1}(x_{h'+1}, a) \mid x_{h'}, a_{h'}] & 0 \leq h' \leq h-1 \\ (\mathcal{T}_h^{\mathbf{0}} g_{h+1})(x_h, a_h) - f_h(x_h, a_h) & h' = h \\ 0 & h+1 \leq h' \leq H-1. \end{cases}$$

Following the same steps as in case (i) we can verify $\tilde{f} \in \mathcal{F}_{\text{on}}$, construct $\tilde{f}^c \in \mathcal{Z}_{\text{on}}$ with $V_{\tilde{f}^c}(x_0) \geq \varepsilon/(2H)$, and show that $g$ is eliminated by $\pi_{0:h^t-1}^t \circ \pi_{\tilde{f}^c, h^t}$ in the offline phase for some $t, h^t$.

**Proof of Claim 2** (i) From the assumption, we know that realizability condition $Q_R^* = (Q_{R,0}^*, \ldots, Q_{R,H-1}^*) \in \mathcal{F}_{\text{off}}(R)$ holds. (ii) For the second argument, we note that $\mathcal{E}(Q_R^*, \pi, \pi', h) = 0$ for any $\pi$, $h \in [H]$, $\pi' \in \Pi_{\text{est}}^t$ by the definition of the average Bellman error. From the concentration argument in the offline phase (Eq. (12)), we have $|\hat{\mathcal{E}}^R(Q_R^*, \pi^t, \pi', h^t)| \leq |\mathcal{E}^R(Q_R^*, \pi^t, \pi', h^t)| + \varepsilon_{\text{elim}}/4 = \varepsilon_{\text{elim}}/4$. As a result, $Q_R^*$ will not be eliminated.

**Proof of Claim 3** From Claim 1, we know that in the offline phase, for any $g \in \mathcal{F}_{\text{off}}(R)$, if $\exists h \in [H]$, s.t. $|\mathcal{E}_{\text{V}}^R(g, \pi_g, h)| \geq \varepsilon/H$, then it will be eliminated. Therefore from the policy loss decomposition in Jiang et al. (2017), for all survived $g \in \mathcal{F}_{\text{sur}}(R)$ in the offline phase, we have

$$V_g(x_0) - V_{R,0}^{\pi_g}(x_0) = \sum_{h=0}^{H-1} \mathcal{E}_{\text{V}}^R(g, \pi_g, h) < \varepsilon.$$

Since $Q_R^*$ is not eliminated, similar as Jiang et al. (2017), we have

$$V_{R,0}^{\pi_{\hat{g}}}(x_0) > V_{\hat{g}}(x_0) - \varepsilon \geq V_{R,0}^*(x_0) - \varepsilon.$$

In sum, we can see the three claims hold with probability at least $1 - \delta$. Since Claim 3 directly implies that RFOLIVE returns an $\varepsilon$-near optimal policy, we complete the proof. $\qquad\square$

### D.3 V-type RFOLIVE for unknown representation low-rank MDPs

Here we provide the details of instantiating V-type RFOLIVE to low-rank MDPs (Agarwal et al., 2020; Modi et al., 2021; Uehara et al., 2021). Firstly we remark that they assume the normalization in Definition 8 holds for $f' : \mathcal{X} \to [0, 1]$ instead of $f' : \mathcal{X} \to [-1, 1]$. We use the different version for ease of presentation and our results also hold under their normalization. In addition, both versions are implied by the definition in Jin et al. (2020b).

Now we show the complete corollary statement.

**Corollary** (Full version of Corollary 4). *Fix $\delta \in (0, 1)$. Consider a low-rank MDP $M$ of embedding dimension $d_{\mathrm{lr}}$ with a realizable feature class $\Phi^{\mathrm{lr}}$ (Assumption 3) and a reward function class $\mathcal{R}$. With probability at least $1 - \delta$, for any $R \in \mathcal{R}$, V-type RFOLIVE (Algorithm 1) with $\mathcal{F}(\Phi^{\mathrm{lr}})$ outputs a policy $\hat{\pi}$ that satisfies $v_R^{\hat{\pi}} \geq v_R^* - \varepsilon$. The required number of episodes is*

$$\tilde{O}\left( \frac{\left(H^8 d_{\mathrm{lr}}^3 \log(|\Phi^{\mathrm{lr}}|) + H^5 d_{\mathrm{lr}}^2 \log(\mathcal{N}_{\mathcal{R}}(\varepsilon/2048H^2\sqrt{d_{\mathrm{lr}}\iota}))\right) K \log(1/\delta)}{\varepsilon^2} \right).$$

*In RFOLIVE, we*

$$\varepsilon_{actv} = \frac{\varepsilon}{8H^2}, \varepsilon_{elim} = \frac{\varepsilon}{32H^2\sqrt{d_{\mathrm{lr}}\iota}}, \ n_{actv} = \frac{H^6\iota}{\varepsilon^2}$$

*and*

$$n_{elim} = \frac{\left(H^7 d_{\mathrm{lr}}^2 \log(|\Phi^{\mathrm{lr}}|) + H^4 d_{\mathrm{lr}} \log(\mathcal{N}_{\mathcal{R}}(\varepsilon_{elim}/64))\right) K\iota^3}{\varepsilon^2}$$
$$= \frac{\left(H^7 d_{\mathrm{lr}}^2 \log(|\Phi^{\mathrm{lr}}|) + H^4 d_{\mathrm{lr}} \log(\mathcal{N}_{\mathcal{R}}(\varepsilon/2048H^2\sqrt{d_{\mathrm{lr}}\iota}))\right) K\iota^3}{\varepsilon^2},$$

*where $\iota = c_7 \log(HdK/\delta\varepsilon)$ and $c_7$ is large enough constant.*

Before the formal proof, we provide some discussions and comparisons. Firstly, when $\mathcal{R}$ is finite, the bound becomes $\tilde{O}\left( \frac{\left(H^8 d_{\mathrm{lr}}^3 \log(|\Phi^{\mathrm{lr}}|) + H^5 d_{\mathrm{lr}}^2 \log(|\mathcal{R}|)\right) K \log(1/\delta)}{\varepsilon^2} \right)$. Compared with Modi et al. (2021), our result significantly improves upon their $\tilde{O}\left( \frac{H^6 d_{\mathrm{lr}}^{11} K^{14} \log(|\Phi^{\mathrm{lr}}|/\delta)}{\eta_{\min}^5} + \frac{H^7 d_{\mathrm{lr}}^3 K^5 \log(|\Phi^{\mathrm{lr}}||\mathcal{R}|/\delta)}{\varepsilon^2 \eta_{\min}} \right)$ rate and does not require the reachability assumption ($\eta_{\min}$ is their reachability factor). On the other hand, their algorithm is more computationally viable and achieves the optimal deployment complexity (Huang et al., 2021). With the additional access to and the realizability assumption of the right feature candidate class $\Upsilon^{\mathrm{lr}}$ in low-rank MDPs, another related work Agarwal et al. (2020) provide a computationally efficient reward-free exploration guarantee but their rate $\frac{H^{22} d_{\mathrm{lr}}^7 K^9 \log(|\Phi^{\mathrm{lr}}||\Upsilon^{\mathrm{lr}}|/\delta)}{\varepsilon^{10}}$ is also much worse than ours.

In the sequel, we present the detailed proof for the corollary.

*Proof.* We first verify that $\mathcal{F}$ satisfies the assumptions in Theorem 3. Here we have that $\mathcal{F} = \mathcal{F}(\Phi^{\mathrm{lr}}) = \mathcal{F}_0(\Phi^{\mathrm{lr}}, H - 1) \times \ldots \times \mathcal{F}_{H-1}(\Phi^{\mathrm{lr}}, 0)$, where

$$\mathcal{F}_h(\Phi^{\mathrm{lr}}, B_h) = \left\{ f_h(x_h, a_h) = \langle \phi_h^{\mathrm{lr}}(x_h, a_h), \theta_h \rangle \colon \|\theta_h\|_2 \leq B_h \sqrt{d_{\mathrm{lr}}}, \langle \phi_h^{\mathrm{lr}}(\cdot), \theta_h \rangle \in [-B_h, B_h] \right\}.$$

Applying the property of linear MDPs (Lemma 9) gives us that

$$Q_{R,h}^*(x_h, a_h) = R_h(x_h, a_h) + \mathbb{E}\left[ \max_{a_{h+1}} Q_{R,h+1}^*(x_{h+1}, a_{h+1}) \mid x_h, a_h \right]$$
$$= R_h(x_h, a_h) + \langle \phi_h^{\mathrm{lr}}(x_h, a_h), \theta_h^* \rangle,$$

where $\|\theta_h^*\|_2 \leq (H - (h+1))\sqrt{d_{\mathrm{lr}}}$ and $\langle \phi_h^{\mathrm{lr}}(\cdot), \theta_h^* \rangle \in [0, H - (h+1)]$. Therefore, for any $h \in [H]$, we have $Q_{R,h}^* \in \mathcal{F}_h(\Phi^{\mathrm{lr}}, H - h - 1) + R_h = \mathcal{F}_h + R_h$. This implies that for any $R \in \mathcal{R}$, we get $Q_R^* \in \mathcal{F} + R$, thus, realizability (Assumption 1) holds.

Again, applying Lemma 9, for any $h \in [H], f_{h+1} \in \mathcal{F}_{h+1}, R_{h+1} \in \mathcal{R}_{h+1}$, we have that

$$(\mathcal{T}_h^0(f_{h+1} + R_{h+1}))(x_h, a_h) = \mathbb{E}\left[\max_{a_{h+1}}(f_{h+1}(x_{h+1}, a_{h+1}) + R_{h+1}(x_{h+1}, a_{h+1})) \mid x_h, a_h\right]$$
$$= \left\langle \phi_h^{\mathrm{lr}}(x_h, a_h), \theta_{f+R,h}^* \right\rangle,$$

where $\|\theta_{f+R,h}^*\|_2 \leq (H - h - 1)\sqrt{d_{\mathrm{lr}}}$ and $\langle \phi_h^{\mathrm{lr}}(\cdot), \theta_{f+R,h}^* \rangle \in [-(H - h - 1), H - h - 1]$. Thus $\langle \phi_h^{\mathrm{lr}}, \theta_{f+R,h}^* \rangle \in \mathcal{F}_h$. This implies that for any $f_{h+1} \in \mathcal{F}_{h+1}, R_{h+1} \in \mathcal{R}_{h+1}$, we have $\mathcal{T}_h^0(f_{h+1} + R_{h+1}) \in \mathcal{F}_h$. Similarly, we can show $\mathcal{T}_h^0 f_{h+1} \in \mathcal{F}_h$.

Moreover, for any $f_{h+1}, f_{h+1}' \in \mathcal{F}_{h+1}$, we have that $\|f_{h+1} - f_{h+1}'\|_\infty \leq 2(H - h - 2)$. Therefore, there exists $\theta_{f-f',h}^*$ such that $\mathcal{T}_h^0(f_{h+1} - f_{h+1}') = \langle \phi_h^{\mathrm{lr}}, \theta_{f-f',h}^* \rangle \subseteq (\mathcal{X} \times \mathcal{A} \to [-2(H - h - 2), 2(H - h - 2)])$ and $\|\theta_{f-f',h}^*\|_2 \leq 2(H - h - 2)\sqrt{d_{\mathrm{lr}}}$. Then choosing $\theta_h = \theta_{f-f',h}^*/2, \theta_h' = -\theta_{f-f',h}^*/2$ and $f_h = \langle \phi_h^{\mathrm{lr}}, \theta_h \rangle, f_h' = \langle \phi_h^{\mathrm{lr}}, \theta_h' \rangle$ gives us both $f_h(\cdot), f_h'(\cdot) \in [-(H - h - 2), (H - h - 2)] \subseteq [-(H - h - 1), H - h - 1]$ and $\|\theta_h\|_2, \|\theta_h'\|_2 \leq (H - h - 1)\sqrt{d_{\mathrm{lr}}}$. Therefore, we have that $f_h - f_h' \in \mathcal{F}_h - \mathcal{F}_h$.

The above discussions imply that completeness (Assumption 2) holds.

Invoking Theorem 3 and further noticing the covering number argument (Lemma 8) and the bound on V-type Bellman Eluder dimension (Proposition 4), we know that the output policy is $\varepsilon$-optimal and the sample complexity is

$$\tilde{O}\left(\frac{\left(H^7 \log(\mathcal{N}_\mathcal{F}(\varepsilon/2048H^2\sqrt{d_{\mathrm{lr}}\iota}))\iota + H^5 \log(\mathcal{N}_\mathcal{R}(\varepsilon/2048H^2\sqrt{d_{\mathrm{lr}}\iota}))\right) d_{\mathrm{lr}}^2 K\iota^3}{\varepsilon^2}\right)$$

$$= \tilde{O}\left(\frac{\left(H^8 d_{\mathrm{lr}}^3 \log(|\Phi^{\mathrm{lr}}|) + H^5 d_{\mathrm{lr}}^2 \log(\mathcal{N}_\mathcal{R}(\varepsilon/2048H^2\sqrt{d_{\mathrm{lr}}\iota}))\right) K \log(1/\delta)}{\varepsilon^2}\right). \qquad \square$$

## E Hardness result for unknown representation linear completeness setting

In this section, we provide the detailed construction and proof for the hardness result and more discussions.

**Theorem** (Restatement of Theorem 5). *There exists a family of MDPs $\mathcal{M}$, a reward class $\mathcal{R}$ and a feature set $\Phi^{\mathrm{lc}}$, such that $\forall M \in \mathcal{M}$, the $(M, \Phi^{\mathrm{lc}})$ pair satisfies Assumption 4 (i.e., $\Phi^{\mathrm{lc}}$ is realizable linear complete feature class for any $M \in \mathcal{M}$), yet it is information-theoretically impossible for an algorithm to obtain a $\mathrm{poly}\left(d_{\mathrm{lc}}, H, \log(|\Phi^{\mathrm{lc}}|), \log(|\mathcal{R}|), 1/\varepsilon, \log(1/\delta)\right)$ sample complexity for reward-free exploration with the given reward class $\mathcal{R}$.*

*Proof.* We present an exponential tree MDP as a hard instance, similar to the lower bound instances in Modi et al. (2020), and design "one-hot" realizable feature inspired by the construction in Zanette et al. (2020a) which they used to show that a low-IBE (Inherent Bellman Error) setting does not imply a low-rank/linear MDP.

**Family of hard instances** We consider a class of deterministic finite state MDPs $\mathcal{M}$ with a singleton reward class. In our construction, for simplicity, the MDPs have a layered structure where the set of states an agent can encounter at any two timesteps $h$ and $h'$ ($h \neq h'$) are disjoint. Hence, we denote the respective state spaces for each timestep $h$ as $\mathcal{X}_h$, and we always have $x_h \in \mathcal{X}_h$. In this layered MDP, for each timestep $h \in [H]$, we only define the corresponding feature $\phi$, rewards at each $\mathcal{X}_h$, and transition functions from $\mathcal{X}_h$ to $\mathcal{X}_{h+1}$. To convert it to the non-layered MDPs, at each timestep $h \in [H]$, we only need to let the features $\phi$ and reward functions be $\mathbf{0}$ at the states outside $\mathcal{X}_h$ and let transitions have 0 probability when transiting from states in $\mathcal{X}_h$ to states outside $\mathcal{X}_{h+1}$ and define the transition functions from some states outside $\mathcal{X}_h$ arbitrarily.

Consider a complete binary tree of depth $H - 2$ (we count the first layer $x_0$ as depth 0). The vertices at each level $h$ from the state space $\mathcal{X}_h$ and the two outgoing edges represent the available actions at each state. The reward class is a singleton class $\{R\}$, where all states get zero reward other than $R_{H-1}(x^+, \mathrm{NULL}) = 1$. The starting state of the MDP is the root node $x_0$ and the

dynamics are deterministic at all levels: each action $\{\text{left}, \text{right}\}$ transits to the corresponding child node. Of all the $2^{H-2}$ nodes at level $H-2$, on one node $x_{H-2}^*$, one action $a_{H-2}^*$ transits to $x^+$ with probability 1 whereas the other action and all actions for other nodes transit to $x^-$ (i.e., only $P_{H-1}(x^+ \mid x_{H-2}^*, a_{H-2}^*) = 1$). As we have $2^{H-1}$ many choices for $(x_{H-2}^*, a_{H-2}^*)$, we have $|\mathcal{M}| = 2^{H-1}$. We provide an illustration for $H = 4$ in Figure 1.

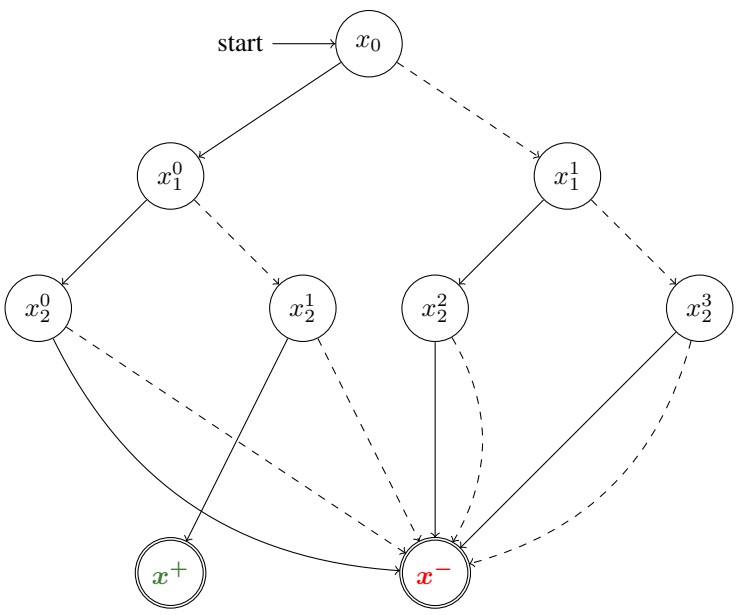

Figure 1: A hard instance for $H = 4$ with two actions: left (solid arrow) and right (dashed arrow). The complete binary tree portion ranges from timestep 0 to 2, and $x^+, x^-$ belong to timestep 3. On timestep $h = 2$, only $(x_2^*, a_2^*) = (x_2^1, \text{left})$ transits to the good state $x^+$, and all other state-action pairs transit to bad state $x^-$. Rewards for all state-action pairs are 0 other than $R_3(x^+, \text{NULL}) = 1$.

**Constructing the feature class** We now construct a feature class $\Phi^{\text{lc}}$ such that for any MDP $M \in \mathcal{M}$, $\Phi^{\text{lc}}$ satisfies Assumption 4 (i.e., the linearly complete feature under $M$ belongs to $\Phi^{\text{lc}}$). We define the feature class in the following way: for each timestep $h \in [H-1]$, we define $\Phi_h^{\text{lc}} = \{\phi_h^i : i \in [2^{h+1}], \phi_h^i[j, a] = \mathbf{1}[i = 2 \cdot j + a], \forall j \in [2^h], a \in \{0, 1\}\}$, where $\phi_h^i[j, a]$ denotes the value of feature $\phi_h^i$ on the $j$-th state $(x_h^j)$ and action $a$ at level $h$. Finally, the two nodes at timestep $H-1$ have a feature value of $\phi_{H-1}(x^+, \text{NULL}) = 1$ for the rewarding node and $\phi_{H-1}(x^-, \text{NULL}) = 0$ for the non-rewarding node. Since we define a feature class of size $|\Phi_h^{\text{lc}}| = 2^{h+1}$ for each level $h \in [H-1]$, the total size of the product class is $|\Phi^{\text{lc}}| = \Pi_{h=0}^{H-2} |\Phi_h^{\text{lc}}| = 2^{(H-1)H/2}$.

**Verifying Assumption 4** Notice that from our construction of $\mathcal{M}$, there is a one-on-one correspondence between one of $2^{H-1}$ state-action pair $(x_{H-2}^*, a_{H-2}^*)$ and one of $2^{H-1}$ MDP $M \in \mathcal{M}$. Therefore, for $i$-th such state-action pair ($i \in [2^{H-1}]$) at level $H-2$, we use $M^i$ to denote its corresponding MDP. Now consider any MDP $M^i \in \mathcal{M}$. For any level $h \in [H-1]$, let $i_h$ denote the state-action pair (among $2^{h+1}$-many state-action pairs at depth $h$) which lies along the path from the root to the rewarding node $x^+$. To verify the realizability condition, we show that the feature $\phi^{\text{lc},i} = \left(\phi_0^{i_0}, \ldots, \phi_h^{i_h}, \ldots, \phi_{H-2}^{i_{H-2}}, \phi_{H-1}\right) \in \Phi^{\text{lc}}$ satisfies the linear completeness structure (Definition 6). Firstly, note that by definition $\left\|\phi_h^{i_h}(x_h, a_h)\right\|_2 \le 1$ for all $h \in [H], x_h, a_h$. Now, we verify that the requirements in Definition 6 are satisfied.

For any $h \in [H-1]$ and pair $(x_h, a_h)$ with $\phi_h^{\text{lc},i}(x_h, a_h) = 0$, all subsequent states $x_{h+1}$ reachable from $x_h$ and any action $a$ also have $\phi_{h+1}^{\text{lc},i}(x_{h+1}, a) = 0$: zero-feature intermediate state-action pairs only transit to zero feature value states at the next timestep. Therefore, the backup condition is

satisfied by default:

$$\left\langle \phi_h^{\mathrm{lc},i}(x_h, a_h), \theta_h \right\rangle - \left( \mathcal{T}_h^{\mathbf{0}} \left\langle \phi_{h+1}^{\mathrm{lc},i}, \theta_{h+1} \right\rangle \right)(x_h, a_h) = 0 - 0 = 0.$$

On the other hand, for any $h \in [H-1]$ and pair $(x_h, a_h)$ with $\phi_h^{\mathrm{lc},i}(x_h, a_h) = 1$, we have $\phi_{h+1}^{\mathrm{lc},i}(x_{h+1}, a) = 1$ for one action along the path to $x^+$ and $\phi_{h+1}^{\mathrm{lc},i}(x_{h+1}, a') = 0$ for the other. For any $\theta_{h+1} \in \mathbb{R}$ (notice that $d_{\mathrm{lc}} = 1$ in our construction), we have

$$\left( \mathcal{T}_h^{\mathbf{0}} \left\langle \phi_{h+1}^{\mathrm{lc},i}, \theta_{h+1} \right\rangle \right)(x_h, a_h) = \begin{cases} \theta_{h+1} & \theta_{h+1} \geq 0 \\ 0 & \theta_{h+1} < 0. \end{cases}$$

Thus, for both cases, we can set $\theta_h = \theta_{h+1}$ or $0$ to satisfy the linear completeness condition $\langle \phi_h^{\mathrm{lc},i}(x_h, a_h), \theta_h \rangle - \left( \mathcal{T}_h^{\mathbf{0}} \langle \phi_{h+1}^{\mathrm{lc},i}, \theta_{h+1} \rangle \right)(x_h, a_h) = 0$. Hence, the chosen feature mapping $\phi^{\mathrm{lc},i}$ satisfies the linear completeness structure in Definition 6.

**Lower bound for exploration**   Learning in this family of MDPs $\mathcal{M}$ is provably hard as the feature and reward classes do not reveal any information about the pair $(x_{H-2}^*, a_{H-2}^*)$ and the agent has to try each of the $2^{H-1}$ paths (Krishnamurthy et al., 2016). Hence, any learning agent has to sample $\Omega(2^H)$ trajectories to find the optimal policy in any given instance $M \in \mathcal{M}$. The stated lower bound statement follows from the fact that $d_{\mathrm{lc}} = 1$, $A = 2$, $1/\varepsilon$ is constant and the sample complexity is $\Omega(2^H)$ which scales with $\left| \Phi^{\mathrm{lc}} \right| = 2^{(H-1)H/2}$ and $|\mathcal{X}| = 2^{H-1} + 1$. □

**Discussions**   The family $\mathcal{M}$ of hard instances highlights a fundamental distinction between the low-rank and linear completeness settings when underlying true representations are unknown. Our result further highlights that assuming reachability (Modi et al., 2021) and/or explorability (Zanette et al., 2020b) does not alleviate the fundamental hardness. Reachability is satisfied as for each MDP in $\mathcal{M}$, each node at every level can be reached with probability 1 by taking the correct actions which lie along the path from the root node. Similarly, for explorability, we need to verify that for any $\theta \in \mathbb{R}$, the constant $\max_\pi \min_{|\theta|=1} |\mathbb{E}_\pi[\langle \phi_h(x_h, a_h), \theta \rangle]|$ is large for all $h \in [H]$ (notice that $\theta$ is one dimension so $\|\theta\|_2 = |\theta|$). Again, it is easy to see that for both values of $\theta \in \{-1, 1\}$, the policy corresponding to the path from root node $x_0$ to $x^+$ maximizes this constant for all steps with a value of 1.

Moreover, our constructed family of hard instances is quite general as it is applicable to the settings of online reward-specific exploration and learning with a generative model. In order to verify this for the former setting, note that our reward class is a singleton reward $\{R\}$ and exposing this reward (reward class) to the agent still does not disclose any information about the pair $(x_{H-2}^*, a_{H-2}^*)$ to the agent. Hence, the required number of trajectories to identify this pair is again $\Omega(2^H)$. Similarly, for a generative model, the problem of identifying the pair $(x_{H-2}^*, a_{H-2}^*)$ is inherently a best-arm identification problem among the $2^{H-1}$ possibilities. Thus, the existing lower bounds for best-arm identification (Krishnamurthy et al., 2016) directly lead to a sample complexity bound of $\Omega(2^H)$. In addition, we can see from the construction that our hardness result also shows that a polynomial in $|\mathcal{X}|$ dependence is unavoidable in this case. We also remark that the stated hardness result can be easily tweaked to show a $1/\varepsilon^2$ dependence for identifying an $\varepsilon$-optimal policy by moving from deterministic transitions at timestep $H-1$ to stochastic transition probabilities: $P_{H-1}(x^+ \mid x_{H-2}^*, a_{H-2}^*) = \frac{1}{2} + \varepsilon$, $P_{H-1}(x^- \mid x_{H-2}^*, a_{H-2}^*) = \frac{1}{2} - \varepsilon$, and $P_{H-1}(x^+ \mid x_{H-2}, a_{H-2}) = P_{H-1}(x^- \mid x_{H-2}, a_{H-2}) = \frac{1}{2}$ if $x_{H-2} \neq x_{H-2}^*$ or $a_{H-2} \neq a_{H-2}^*$. The realizable feature $\phi^{\mathrm{lc},i}$ will be a two-dimensional representation after this modification, where we change previous one dimension values 0 and 1 to two dimension $(0,0)$ and $(1,-1)$ respectively.

The hardness result highlights the insufficiency of realizability of a linearly complete feature (Assumption 4) in the representation learning setting and indicates that realizability of stronger completeness style features may be necessary for provably efficient reward-free RL.

## F   Algorithm-specific counterexample of RFOLIVE

In this section, we show an algorithm-specific counterexample of RFOLIVE (Algorithm 1) that satisfies realizability (Assumption 1) and has a low Bellman Eluder dimension, while only violates

completeness (Assumption 2). Together with the positive results (Theorem 1 and Theorem 3), we conjecture that realizability-type assumptions are not sufficient for statistically efficient reward-free RL. As we know that OLIVE (Jiang et al., 2017; Jin et al., 2021) only requires realizability and low Bellman Eluder dimension for reward-aware RL and RFOLIVE is its natural extension to the reward-free setting, we believe that the hardness between reward-aware and reward-free RL has a deep connection to the sharp separation between realizability and completeness (Chen and Jiang, 2019; Wang et al., 2020b, 2021; Xie and Jiang, 2021; Weisz et al., 2021a,b, 2022; Foster et al., 2021).

**Theorem 7.** *There exists an MDP $M$, a function class $\mathcal{F}$, a reward class $\mathcal{R}$, where Assumption 1 holds and the function class $\mathcal{F} - \mathcal{F}$ has a low Bellman Eluder dimension ($d_{\mathrm{qbe}}$ defined in Theorem 1). However, with probability 0.25, (Q-type) RFOLIVE with infinite amount of data cannot output a 0.1-optimal policy for some $R \in \mathcal{R}$.*

*Proof.* We first discuss the counterexample shown in Figure 2 and Table 3 at a high level. In our construction, with probability 0.25, the agent will only explore and collect data at some specific place because it is sufficient to eliminate all candidate functions predicting large positive values at $x_0$. Then in the offline phase, the agent cannot eliminate some bad function because of lack of support in the collected data. Then performing function elimination in the offline phase fails. We provide more details in the sequel.

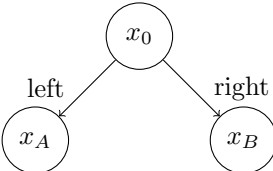

Figure 2: Algorithm-specific counterexample of RFOLIVE without completeness assumption (Assumption 2).

|  | $(x_0, \mathrm{left})$ | $(x_0, \mathrm{right})$ | $(x_A, \mathrm{NULL})$ | $(x_B, \mathrm{NULL})$ |
|---|---|---|---|---|
| $R_1$ | 0 | 0 | 1 | 0 |
| $R_2$ | 0 | 0 | 0.2 | 0.1 |
| $Q^*_{R_1}$ | 1 | 0 | 1 | 0 |
| $Q^*_{R_2}$ | 0.2 | 0.1 | 0.2 | 0.1 |
| $f_{R_1}$ | 1 | 0 | 0 | 0 |
| $f_{R_2}$ | 0.2 | 0.1 | 0 | 0 |
| $f_{\mathrm{bad}}$ | 0.21 | 0.3 | 0.01 | 0.1 |
| $f_{R_1} + R_1 = Q^*_{R_1}$ | 1 | 0 | 1 | 0 |
| $f_{R_2} + R_2 = Q^*_{R_2}$ | 0.2 | 0.1 | 0.2 | 0.1 |
| $f_{R_1} + R_2$ | 1 | 0 | 0.2 | 0.1 |
| $f_{\mathrm{bad}} + R_2$ | 0.21 | 0.3 | 0.21 | 0.1 |

Table 3: Algorithm-specific counterexample for RFOLIVE without completeness assumption (Assumption 2).

**Construction** In Figure 2, taking action left and action right in state $x_0$ transits to $x_A$ and $x_B$ respectively. We denote the null action at $x_A, x_B$ and the null state at level $H = 2$ as NULL and $x_{\mathrm{NULL}}$ respectively. In this example, the length of horizon is $H = 2$. We construct $\mathcal{F} = \mathcal{F}_0 \times \mathcal{F}_1$, where $\mathcal{F}_0 = \{\mathbf{0}, f_{R_1,0}, f_{R_2,0}, f_{\mathrm{bad},0}\}$ and $\mathcal{F}_1 = \{\mathbf{0}, f_{\mathrm{bad},1}\}$. In addition, we construct $\mathcal{R} = \mathcal{R}_0 \times \mathcal{R}_1$, where $\mathcal{R}_0 = \{\mathbf{0}\}$ and $\mathcal{R}_1 = \{R_{1,1}, R_{2,1}\}$. Recall that the second subscript of $f \in \mathcal{F}_0, \mathcal{F}_1$ and $R \in \mathcal{R}_1$ is the index for the timestep. The details are shown in Table 3. Notice that here we use the layered MDP for simplicity. To convert it to a non-layered MDP, we only need to set corresponding values in the transition function, reward function, and $f \in \mathcal{F}$ to be 0.

**Verifying realizability and low Bellman Eluder dimension** One can immediately see that realizability (Assumption 1) is satisfied. For example, we have $Q^*_{R_1,1} = f_{R_1,1} + R_{1,1} = \mathbf{0} + R_{1,1}$, which implies that $Q^*_{R_1,1} \in \mathcal{F}_1 + \mathcal{R}_1$. Similarly, we can verify that $Q^*_{R_2,1} \in \mathcal{F}_1 + \mathcal{R}_1, Q^*_{R_1,0} \in \mathcal{F}_0 + \mathcal{R}_0, Q^*_{R_2,0} \in \mathcal{F}_0 + \mathcal{R}_0$.

In addition, $\mathcal{F} - \mathcal{F}$ has a low Bellman Eluder dimension. It is because the Bellman Eluder dimension can be upper bounded by the Bellman rank (Proposition 3) and the Bellman rank can be upper bounded by the number of states (Jiang et al., 2017). Therefore, the Bellman Eluder dimension is just a small bounded finite number. Later we will show that with even infinite amount data RFOLIVE fails, which implies that we cannot get a polynomial sample complexity bound in this case.

**Violation of completeness** We can easily see that for $f_{\text{bad},1} \in \mathcal{F}_1$ and $R_{2,1} \in \mathcal{R}_1$, its Bellman backup $\mathcal{T}_0^{\mathbf{0}}(f_{\text{bad},1} + R_{2,1}) \notin \mathcal{F}_0$. This means that completeness (Assumption 2) does not hold.

**RFOLIVE fails in the counterexample** We first consider running (Q-type) RFOLIVE on this counterexample during the online phase. In the following, we will assume the more favorable case where the agent can collect infinitely many number of samples in line 5 and line 11 (i.e., no statistical/estimation error for the average Bellman error in the empirical version).

The agent will pick the most optimistic function for exploration. In the first iteration, such an optimistic function at level 0 will be equal to $f_{R_1,0} - \mathbf{0}$. Therefore, starting from $x_0$, the agent will choose the action left. With at least probability 0.5, it will pick level 0 to eliminate and collect data (i.e., collecting data at $(x_0, \text{left})$ in line 11). The reason is that for line 9, the large average Bellman error always exists at $(x_0, \text{left})$ while the Bellman error could be large at $(x_{\text{left}}, \text{NULL})$. By adversarial tie-breaking, there is at least 0.5 probability that $(x_0, \text{left})$ is chosen.

Now consider the case that the agent pick $(x_0, \text{left})$ to collect data in line 11. We can see that only function $\mathbf{0}, f_{\text{bad}} - f_{R_2}, f_{R_2} - f_{\text{bad}} \in \mathcal{F} - \mathcal{F}$ will survive while all other functions violate the collected constraint. Here we notice that for any $f \in \mathcal{F} - \mathcal{F}$ we have $V_f(x_A) = 0$ or $V_f(x_A) = \pm 0.01$. So the survived function $f \in \mathcal{F} - \mathcal{F}$ belongs to one of the following cases: (i) $f(x_0, \text{left}) = 0$ and $V_f(x_A) = 0$, (ii) $f(x_0, \text{left}) = 0.01$ and $V_f(x_A) = 0.01$, or (iii) $f(x_0, \text{left}) = -0.01$ and $V_f(x_A) = -0.01$. At the second iteration, RFOLIVE will choose $f_{\text{bad}} - f_{R_2}$ (i.e., case (ii)) and action right. Due to adversarial tie-breaking, we have that with probability 0.5, the agent collects data at $(x_B, \text{NULL})$ and eliminates $f_{\text{bad}} - f_{R_2}$. In this case, for the third iteration, the agent chooses function $\mathbf{0}$ and then terminates in line 6 since the average Bellman error is 0.

For the offline phase, let us consider the reward function $R_2$ and the elimination on $\mathcal{F}_{\text{off}}(R_2) = \mathcal{F} + R_2$. Recall that in the online phase, we only collect data on $(x_0, \text{left}, x_A)$ and $(x_B, \text{NULL}, x_{\text{NULL}})$. It is easy to see that we will eliminate $f_{R_1} + R_2$ from this constraint. However, we cannot eliminate either $f_{R_2} + R_2$ or $f_{\text{bad}} + R_2$ because they all have zero average Bellman error under these two constraints. Then by optimistic selection criteria, the agent will pick $f_{\text{bad}} + R_2$. This induces a sub-optimal policy (right) with accuracy $\varepsilon = 0.1$.

Therefore, with probability at least 0.25, (Q-type) RFOLIVE fails to output a 0.1 optimal policy for some reward $R \in \mathcal{R}$ in this counterexample. □

## G Discussions on other variants of OLIVE

In this section, we briefly discuss that some other variant of OLIVE in the reward-free setting could easily fail under Assumption 1, Assumption 2, and low Bellman Eluder dimension (where we know that RFOLIVE works).

One adaptation of OLIVE to the reward free case is to perform exploration on the joint function class space and we call it JOINTOLIVE. More specifically, we maintain a version space $\mathcal{F}^t + \mathcal{R}^t \subseteq \mathcal{F} + \mathcal{R}$ during the online phase. In each online iteration, we pick the most optimistic function $f_{\text{on}}^t = f^t + R^t = \arg\max_{f+R \in \mathcal{F}^t + \mathcal{R}^t} V_{f+R}(x_0)$ and explore according to $\pi^t = \pi_{f_{\text{on}}^t}$. Here for $f_{\text{on}}^t$, we decompose it as the sum of $f^t \in \mathcal{F}^t$ and $R^t \in \mathcal{R}^t$. Then we roll out policy $\pi^t$ and estimate the average Bellman error. For the termination condition, like OLIVE, we check whether $f_{\text{on}}^t$ has a small average Bellman error under reward $R^t$. If the algorithm is not terminated, we pick a level for elimination and collect the constraint. At the end of each online iteration, we shrink the version space of $\mathcal{F}^t$ and $\mathcal{R}^t$ according to the average Bellman error. For the offline phase, we use collected

constraints to perform elimination like RFOLIVE and then output the greedy policy of the optimistic survived function. We will show that this variant could get stuck in the following counterexample even in the case that we are allowed to collect infinite amount of samples to build estimates (i.e., no statistical/estimation error).

**Construction**   We consider the MDP in Figure 2 and reward function class $\mathcal{R} = \mathcal{R}_0 \times \mathcal{R}_1$, where $\mathcal{R}_0 = \{\mathbf{0}\}$ and $\mathcal{R}_1 = \{R_{1,1}, R_{2,1}\}$. The function class $\mathcal{F} = \mathcal{F}_0 \times \mathcal{F}_1$ is constructed as $\mathcal{F}_0 = \{\mathbf{0}, f_{R_1,0}, f_{R_2,0}, f_{\mathrm{bad},0}\}$ and $\mathcal{F}_1 = \{\mathbf{0}\}$. Compared with the counterexample in Theorem 7, the difference is that $f_{\mathrm{bad}}$ is changed and now we only have a single function $\mathbf{0}$ in $\mathcal{F}_1$. The details as shown in Table 4. As discussed in the proof of Theorem 7, we can convert the layered MDP here to a non-layered one.

|  | $(x_0, \mathrm{left})$ | $(x_0, \mathrm{right})$ | $(x_A, \mathrm{NULL})$ | $(x_B, \mathrm{NULL})$ |
|---|---|---|---|---|
| $R_1$ | 0 | 0 | 1 | 0 |
| $R_2$ | 0 | 0 | 0.2 | 0.1 |
| $Q^*_{R_1}$ | 1 | 0 | 1 | 0 |
| $Q^*_{R_2}$ | 0.2 | 0.1 | 0.2 | 0.1 |
| $f_{R_1}$ | 1 | 0 | 0 | 0 |
| $f_{R_2}$ | 0.2 | 0.1 | 0 | 0 |
| $f_{\mathrm{bad}}$ | 0.2 | 0.3 | 0 | 0 |
| $f_{R_1} + R_1 = Q^*_{R_1}$ | 1 | 0 | 1 | 0 |
| $f_{R_2} + R_2 = Q^*_{R_2}$ | 0.2 | 0.1 | 0.2 | 0.1 |
| $f_{R_1} + R_2$ | 1 | 0 | 0.2 | 0.1 |
| $f_{\mathrm{bad}} + R_2$ | 0.2 | 0.3 | 0.2 | 0.1 |

Table 4: Algorithm-specific counterexample for JOINTOLIVE under all assumptions.

**Verifying realizability, completeness, and low Bellman Eluder dimension**   Realizability and low Bellman Eluder dimension can be verified in the same way as the counterexample for RFOLIVE in Theorem 7. For completeness (Assumption 2), one can easily verify that by noticing we have $\mathcal{F}_1 = \{\mathbf{0}\}$ now.

**JOINTOLIVE fails in the counterexample**   In JOINTOLIVE, the agent will pick the optimistic function in the joint function space to explore during the online phase. At the first iteration, there are two candidates $f_{R_1} + R_1$ and $f_{R_1} + R_2$. By adversarial tie-breaking, the agent will choose $f_{R_1} + R_1$ with probability 0.5 and choose action left. Then the agent will terminate immediately because the average Bellman error is 0 for $f_{R_1} + R_1$ everywhere under reward $R_1$.

For the offline phase, we similarly consider reward $R_2$. It is easy to see that $f_{R_1} + R_2$ will be eliminated while both $f_{R_2} + R_2$ and $f_{\mathrm{bad}} + R_2$ will survive. Then by optimistic selection, the agent will choose the greedy policy of $f_{\mathrm{bad}} + R_2$. This induces a sub-optimal policy (right) with accuracy $\varepsilon = 0.1$.

Therefore, with probability 0.5, JOINTOLIVE fails in this counterexample.

# H   Auxiliary results

In this section, we provide auxiliary results for the paper. We show covering number arguments in Appendix H.1 and some bounds on Bellman Eluder dimensions in Appendix H.2.

## H.1   Covering number

In this part, we present the covering number argument for the linear function class used in the paper.

**Lemma 8** (Size of $\varepsilon$-cover for linear function class). *We have three claims here*

1. *Consider $\mathcal{F}(\{\phi^{\mathrm{lc}}\}) = \mathcal{F}_0(\{\phi^{\mathrm{lc}}\}, H-1) \times \ldots \times \mathcal{F}_{H-1}(\{\phi^{\mathrm{lc}}\}, 0)$, where $\mathcal{F}_h(\{\phi^{\mathrm{lc}}\}, B_h) = \{f_h(x_h, a_h) = \langle \phi_h^{\mathrm{lc}}(x_h, a_h), \theta_h \rangle : \|\theta_h\|_2 \le B_h \sqrt{d_{\mathrm{lc}}}, \langle \phi_h^{\mathrm{lc}}(\cdot), \theta_h \rangle \in [-B_h, B_h]\}$. Then we have $\mathcal{N}_{\mathcal{F}(\{\phi^{\mathrm{lc}}\})}(\varepsilon) \le \left( \frac{2H^2 \sqrt{d_{\mathrm{lc}}}}{\varepsilon} \right)^{d_{\mathrm{lc}}}$.*

2. *Consider $\mathcal{F}(\{\phi^{\mathrm{lr}}\}) = \mathcal{F}_0(\{\phi^{\mathrm{lr}}\}, H-1) \times \ldots \times \mathcal{F}_{H-1}(\{\phi^{\mathrm{lr}}\}, 0)$, where $\mathcal{F}_h(\{\phi^{\mathrm{lr}}\}, B_h) = \{f_h(x_h, a_h) = \langle \phi_h^{\mathrm{lr}}(x_h, a_h), \theta_h \rangle : \|\theta_h\|_2 \le B_h \sqrt{d_{\mathrm{lr}}}, \langle \phi_h^{\mathrm{lr}}(\cdot), \theta_h \rangle \in [-B_h, B_h]\}$. Then we have $\mathcal{N}_{\mathcal{F}(\{\phi^{\mathrm{lr}}\})}(\varepsilon) \le \left( \frac{2H^2 \sqrt{d_{\mathrm{lr}}}}{\varepsilon} \right)^{d_{\mathrm{lr}}}$.*

3. *Consider $\mathcal{F}(\Phi^{\mathrm{lr}}) = \mathcal{F}_0(\Phi^{\mathrm{lr}}, H-1) \times \ldots \times \mathcal{F}_{H-1}(\Phi^{\mathrm{lr}}, 0)$, where $\mathcal{F}_h(\Phi^{\mathrm{lr}}, B_h) = \{f_h(x_h, a_h) = \langle \phi_h(x_h, a_h), \theta_h \rangle : \phi_h \in \Phi_h^{\mathrm{lr}}, \|\theta_h\|_2 \le B_h \sqrt{d_{\mathrm{lr}}}, \langle \phi_h(\cdot), \theta_h \rangle \in [-B_h, B_h]\}$. Then we have $\mathcal{N}_{\mathcal{F}(\Phi^{\mathrm{lr}})}(\varepsilon) \le |\Phi^{\mathrm{lr}}| \left( \frac{2H^2 \sqrt{d_{\mathrm{lr}}}}{\varepsilon} \right)^{d_{\mathrm{lr}}}$.*

*Proof.* This is a standard result. For the first one, we can construct a cover over $\{\theta_h : \|\theta_h\|_2 \le (H-h-1)\sqrt{d_{\mathrm{lc}}}\}$ in the 2-norm at scale $\varepsilon/H$ for each level $h \in [H]$. Then this cover immediately implies a cover over the function $\mathcal{F}(\{\phi^{\mathrm{lc}}\})$. The covering number directly follows the covering number of the 2-norm ball. The second follows the same steps. For the third result, we additionally union over $\phi \in \Phi^{\mathrm{lr}}$. $\qquad\square$

## H.2 Bounds on the Bellman Eluder dimension

In this part, we show that Q-type and V-type Bellman Eluder dimensions for the instantiated linear MDP, low-rank MDP, and linear completeness with known feature settings are indeed small. We will use the following relation between Bellman rank and Bellman Eluder dimension from Jin et al. (2021):

**Proposition 3** (Bellman rank $\subseteq$ Bellman Eluder dimension, Proposition 11 and 21 in Jin et al. (2021))**.** *If an MDP with function class $\mathcal{F}$ has Q-type (or V-type) Bellman rank $d_{\mathrm{br}}$ with normalization parameter $\zeta$, then the respective Bellman Eluder dimension $\dim_{\mathrm{qbe}}^{R}(\mathcal{F}, \mathcal{D}_\mathcal{F}, \varepsilon)$ (or $\dim_{\mathrm{vbe}}^{R}(\mathcal{F}, \mathcal{D}_\mathcal{F}, \varepsilon)$) is bounded by $\tilde{O}\left( 1 + d_{\mathrm{br}}^{R} \log \left( 1 + \frac{\zeta}{\varepsilon} \right) \right)$.*

**Linear/low-rank MDPs** Before stating the result for low-rank MDPs, we recall the following well-known property for the class:

**Lemma 9** (Jin et al. (2020b); Modi et al. (2021))**.** *Consider a low-rank MDP $M$ (Definition 8) with embedding dimension $d_{\mathrm{lr}}$. For any function $f : \mathcal{X} \to [-c, c]$, we have:*
$$\mathbb{E}\left[ f(x_{h+1}) \mid x_h, a_h \right] = \left\langle \phi_h^{\mathrm{lr}}(x_h, a_h), \theta_f^* \right\rangle$$
*where $\theta_f^* \in \mathbb{R}^{d_{\mathrm{lr}}}$ and we have $\|\theta_f^*\|_2 \le c\sqrt{d_{\mathrm{lr}}}$. A similar linear representation is true for $\mathbb{E}_{a \sim \pi_{h+1}}[f(x_{h+1}, a) \mid x_h, a_h]$ where $f : \mathcal{X} \times \mathcal{A} \to [-c, c]$ and a policy $\pi_{h+1} : \mathcal{X} \to \mathcal{A}$.*

*Proof.* For state-value function $f$, we have
$$
\begin{aligned}
\mathbb{E}\left[ f(x_{h+1}) \mid x_h, a_h \right] &= \int f(x_{h+1}) P_h(x_{h+1} \mid x_h, a_h) d(x_{h+1}) \\
&= \int f(x_{h+1}) \left\langle \phi_h^{\mathrm{lr}}(x_h, a_h), \mu_h^{\mathrm{lr}}(x_{h+1}) \right\rangle d(x_{h+1}) \\
&= \left\langle \phi_h^{\mathrm{lr}}(x_h, a_h), \int f(x_{h+1}) \mu_h^{\mathrm{lr}}(x_{h+1}) d(x_{h+1}) \right\rangle \\
&= \left\langle \phi_h^{\mathrm{lr}}(x_h, a_h), \theta_f^* \right\rangle,
\end{aligned}
$$
where $\theta_f^* := \int f(x_{h+1}) \mu_h^{\mathrm{lr}}(x_{h+1}) d(x_{h+1})$ is a function of $f$. Additionally, we obtain $\|\theta_f^*\|_2 \le c\sqrt{d_{\mathrm{lr}}}$ from Definition 8.

For Q-value function $f$, we similarly have
$$\mathbb{E}_{a \sim \pi_{h+1}} \left[ f(x_{h+1}, a) \mid x_h, a_h \right] = \left\langle \phi_h^{\mathrm{lr}}(x_h, a_h), \theta_f^* \right\rangle,$$
where $\theta_f^* := \iint f(x_{h+1}, a_{h+1}) \pi(a_{h+1} \mid x_{h+1}) \mu_h^{\mathrm{lr}}(x_{h+1}) d(x_{h+1}) d(a_{h+1})$ and $\|\theta_f^*\|_2 \le c\sqrt{d_{\mathrm{lr}}}$. $\quad\square$

Now, we can state the following bound on the V-type Bellman Eluder dimension for low-rank MDPs:

**Proposition 4** (Low-rank MDP). *Consider a low-rank MDP $M$ of embedding dimension $d_{\mathrm{lr}}$ with a realizable feature class $\Phi^{\mathrm{lr}}$ (Assumption 3). Define the corresponding linear function class $\mathcal{F}(\Phi^{\mathrm{lr}}) = \mathcal{F}_0(\Phi^{\mathrm{lr}}, H-1) \times \ldots \times \mathcal{F}_{H-1}(\Phi^{\mathrm{lr}}, 0)$ using*

$$\mathcal{F}_h(\Phi^{\mathrm{lr}}, B_h) = \Big\{ f_h(x_h, a_h) = \langle \phi_h(x_h, a_h), \theta_h \rangle : \phi_h \in \Phi_h^{\mathrm{lr}}, \|\theta_h\|_2 \le B_h \sqrt{d_{\mathrm{lr}}}, \langle \phi_h(\cdot), \theta_h \rangle \in [-B_h, B_h] \Big\}.$$

*Then, for the difference class $\mathcal{F}_{\mathrm{on}} = \mathcal{F}(\Phi^{\mathrm{lr}}) - \mathcal{F}(\Phi^{\mathrm{lr}})$ we have*

$$\dim_{\mathrm{vbe}}^{\mathbf{0}}(\mathcal{F}_{\mathrm{on}}, \mathcal{D}_{\mathcal{F}_{\mathrm{on}}}, \varepsilon) \le O\left(1 + d_{\mathrm{lr}} \log\left(1 + \frac{H\sqrt{d_{\mathrm{lr}}}}{\varepsilon}\right)\right).$$

*Proof.* We start by showing that the V-type Bellman rank for function class $\mathcal{F}_{\mathrm{on}}$ in the low-rank case is small. To that end, consider the Bellman error defined for any roll-in policy $\pi$ and function $f \in \mathcal{F}_{\mathrm{on}}$:

$$\mathcal{E}_{\mathrm{V}}^{\mathbf{0}}(f, \pi, h) = \mathbb{E}\left[f_h(x_h, a_h) - f_{h+1}(x_{h+1}, a_{h+1}) \mid a_{0:h-1} \sim \pi, a_{h:h+1} \sim \pi_f\right]$$
$$= \mathbb{E}\left[f_h(x_h, a_h) - \left\langle \phi_h^{\mathrm{lr}}(x_h, a_h), \theta_{f,h}^* \right\rangle \mid a_{0:h-1} \sim \pi, a_h \sim \pi_f\right]$$

where we used Lemma 9 for low-rank MDPs to write $\mathbb{E}\left[f_{h+1}(x_{h+1}, a_{h+1}) \mid x_h, a_h, a_{h+1} \sim \pi_f\right]$ as $\langle \phi_h^{\mathrm{lr}}(x_h, a_h), \theta_{f,h}^* \rangle$. Here, $f_{h+1} \in [-2(H-h-2), 2(H-h-2)]$ and $f_h \in [-2(H-h-1), 2(H-h-1)]$ implying that $f_h - \left\langle \phi_h^{\mathrm{lr}}(x_h, a_h), \theta_{f,h}^* \right\rangle \in [-4(H-h-1), 4(H-h-1)]$. Therefore, using Lemma 9 again, we have:

$$\mathcal{E}_{\mathrm{V}}^{\mathbf{0}}(f, \pi, h) = \mathbb{E}\left[f_h(x_h, a_h) - \left\langle \phi_h^{\mathrm{lr}}(x_h, a_h), \theta_{f,h}^* \right\rangle \mid a_{0:h-1} \sim \pi, a_h \sim \pi_f\right]$$
$$= \mathbb{E}\left[\left\langle \phi_{h-1}^{\mathrm{lr}}(x_{h-1}, a_{h-1}), \tilde{\theta}(f) \right\rangle \mid x_{h-1}, a_{h-1}, a_{0:h-1} \sim \pi\right]$$
$$= \left\langle \nu(\pi), \tilde{\theta}(f) \right\rangle$$

where $\|\tilde{\theta}(f)\|_2 \le 4(H-h-1)\sqrt{d_{\mathrm{lr}}}$ and $(\nu(\pi))(x_{h-1}, a_{h-1}) = \mathbb{E}[\phi_{h-1}^{\mathrm{lr}}(x_{h-1}, a_{h-1}) \mid a_{0:h-1} \sim \pi]$. Hence, the V-type Bellman rank for this function class is bounded by $d_{\mathrm{lr}}$ with normalization parameter $4(H-h-1)\sqrt{d_{\mathrm{lr}}}$. Finally using Proposition 3, we get the desired bound on the Bellman Eluder dimension. $\square$

For linear MDPs, where the feature $\phi^{\mathrm{lr}}$ is the known feature case, we show that its Q-type Bellman Eluder dimension is also small:

**Proposition 5** (Linear MDP). *Consider a low-rank MDP $M$ (Definition 8) with embedding dimension $d_{\mathrm{lr}}$ and $\phi^{\mathrm{lr}}$ is known. Define the corresponding linear class $\mathcal{F}(\{\phi^{\mathrm{lr}}\}) = \mathcal{F}_0(\{\phi^{\mathrm{lr}}\}, H-1) \times \ldots \times \mathcal{F}_{H-1}(\{\phi^{\mathrm{lr}}\}, 0)$ using*

$$\mathcal{F}_h(\{\phi^{\mathrm{lr}}\}, B_h) = \Big\{ f_h(x_h, a_h) = \langle \phi_h^{\mathrm{lr}}(x_h, a_h), \theta_h \rangle : \|\theta_h\|_2 \le B_h \sqrt{d_{\mathrm{lr}}}, \langle \phi_h^{\mathrm{lr}}(\cdot), \theta_h \rangle \in [-B_h, B_h] \Big\}.$$

*Then, for the difference class $\mathcal{F}_{\mathrm{on}} = \mathcal{F}(\{\phi^{\mathrm{lr}}\}) - \mathcal{F}(\{\phi^{\mathrm{lr}}\})$ we have*

$$\dim_{\mathrm{qbe}}^{\mathbf{0}}(\mathcal{F}_{\mathrm{on}}, \mathcal{D}_{\mathcal{F}_{\mathrm{on}}}, \varepsilon) \le O\left(1 + d_{\mathrm{lr}} \log\left(1 + \frac{H\sqrt{d_{\mathrm{lr}}}}{\varepsilon}\right)\right).$$

*and*

$$\dim_{\mathrm{vbe}}^{\mathbf{0}}(\mathcal{F}_{\mathrm{on}}, \mathcal{D}_{\mathcal{F}_{\mathrm{on}}}, \varepsilon) \le O\left(1 + d_{\mathrm{lr}} \log\left(1 + \frac{H\sqrt{d_{\mathrm{lr}}}}{\varepsilon}\right)\right).$$

*Proof.* The V-type Bellman Eluder dimension bound (Proposition 4) implies the same upper bound for $\dim_{\mathrm{vbe}}^{\mathbf{0}}(\mathcal{F}_{\mathrm{on}}, \mathcal{D}_{\mathcal{F}_{\mathrm{on}}}, \varepsilon)$, where $\mathcal{F}_{\mathrm{on}}$ is defined using the singleton feature class $\{\phi^{\mathrm{lr}}\}$. For Q-type Bellman Eluder dimension, we again start with the Q-type Bellman rank. For any $f \in \mathcal{F}_{\mathrm{on}}$, we have:

$$\mathcal{E}_{\mathrm{Q}}^{\mathbf{0}}(f, \pi, h) = \mathbb{E}\left[f_h(x_h, a_h) - f_{h+1}(x_{h+1}, a_{h+1}) \mid a_{0:h} \sim \pi, a_{h+1} \sim \pi_f\right]$$
$$= \mathbb{E}\left[\langle \phi_h^{\mathrm{lr}}(x_h, a_h), \theta_h - \theta_h' \rangle - \langle \phi_h^{\mathrm{lr}}(x_h, a_h), \theta_{f,h}^* \rangle \mid a_{0:h} \sim \pi\right]$$
$$= \left\langle \mathbb{E}\left[\phi_h^{\mathrm{lr}}(x_h, a_h) \mid a_{0:h} \sim \pi\right], \theta_h - \theta_h' - \theta_{f,h}^* \right\rangle.$$

Using the same magnitude calculations for $\theta_{f,h}^*$, we have $\|\theta_h - \theta_h' - \theta_{f,h}^*\|_2 \leq 4(H - h - 1)\sqrt{d_{\mathrm{lr}}}$. Therefore, we again have the Q-type Bellman rank bounded by $d_{\mathrm{lr}}$ with normalization parameter $4(H - h - 1)\sqrt{d_{\mathrm{lr}}}$. Using Proposition 3, we get the same bound on the Q-type Bellman Eluder dimension. □

**Linear completeness setting**  For the linear completeness setting in the known feature case, we show that its Q-type Bellman Eluder dimension is small.

**Proposition 6** (Linear completeness setting)**.** *Consider an MDP $M$ that satisfies linear completeness (Definition 6) with feature $\phi^{\mathrm{lc}}$. Define the corresponding linear class $\mathcal{F}(\{\phi^{\mathrm{lc}}\}) = \mathcal{F}_0(\{\phi^{\mathrm{lc}}\}, H - 1) \times \ldots \times \mathcal{F}_{H-1}(\{\phi^{\mathrm{lc}}\}, 0)$ using*

$$\mathcal{F}_h(\{\phi^{\mathrm{lc}}\}, B_h) = \left\{ f_h(x_h, a_h) = \left\langle \phi_h^{\mathrm{lc}}(x_h, a_h), \theta_h \right\rangle : \|\theta_h\|_2 \leq B_h \sqrt{d_{\mathrm{lc}}}, \left\langle \phi_h^{\mathrm{lc}}(\cdot), \theta_h \right\rangle \in [-B_h, B_h] \right\}.$$

*Then, for the difference class $\mathcal{F}_{\mathrm{on}} = \mathcal{F}(\{\phi^{\mathrm{lc}}\}) - \mathcal{F}(\{\phi^{\mathrm{lc}}\})$ we have:*

$$\dim_{\mathrm{qbe}}^{\mathbf{0}}(\mathcal{F}_{\mathrm{on}}, \mathcal{D}_{\mathcal{F}_{\mathrm{on}}}, \varepsilon) \leq O\left(1 + d_{\mathrm{lc}} \log\left(1 + \frac{H\sqrt{d_{\mathrm{lc}}}}{\varepsilon}\right)\right).$$

*Proof.* Consider the Q-type Bellman rank, for any $f \in \mathcal{F}_{\mathrm{on}}$, we have:

$$\begin{aligned}
&\mathcal{E}_{\mathrm{Q}}^{\mathbf{0}}(f, \pi, h) \\
&= \mathbb{E}\left[f_h(x_h, a_h) - f_{h+1}(x_{h+1}, a_{h+1}) \mid a_{0:h} \sim \pi, a_{h+1} \sim \pi_f\right] \\
&= \mathbb{E}\left[\left\langle \phi_h^{\mathrm{lc}}(x_h, a_h), \theta_h - \theta_h' \right\rangle - \left\langle \phi_{h+1}^{\mathrm{lc}}(x_{h+1}, a_{h+1}), \theta_{h+1} - \theta_{h+1}' \right\rangle \mid a_{0:h} \sim \pi, a_{h+1} \sim \pi_f\right] \\
&= \mathbb{E}\left[\left\langle \phi_h^{\mathrm{lc}}(x_h, a_h), \theta_h - \theta_h' \right\rangle - \left\langle \phi_h^{\mathrm{lc}}(x_h, a_h), \theta_{f,h}^* \right\rangle \mid a_{0:h} \sim \pi\right] \\
&= \left\langle \mathbb{E}\left[\phi_h^{\mathrm{lc}}(x_h, a_h) \mid a_{0:h} \sim \pi\right], \theta_h - \theta_h' - \theta_{f,h}^* \right\rangle
\end{aligned}$$

where the penultimate step follows from Definition 6 with the $\|\theta_h - \theta_h'\|_2, \|\theta_{f,h}^*\|_2, \|\theta_{h+1} - \theta_{h+1}'\|_2 \leq 2(H - h - 1)\sqrt{d_{\mathrm{lc}}}$. Thus, we have $\|\theta_h - \theta_{f,h}^*\|_2 \leq 4(H - h - 1)\sqrt{d_{\mathrm{lc}}}$ implying a Q-type Bellman rank bound of $d_{\mathrm{lc}}$ with normalization parameter $4(H - h - 1)\sqrt{d_{\mathrm{lc}}}$. Using Proposition 3, we get the stated bound on the Q-type Bellman Eluder dimension for linear completeness setting. □