# OpenReview forum: "On the Statistical Efficiency of Reward-Free Exploration in Non-Linear RL"
_NeurIPS.cc/2022/Conference — NeurIPS 2022 Accept_

### Official Review · Reviewer_A559 · 2022-07-07

**Rating:** 7
**Confidence:** 4
**Soundness:** 4 excellent
**Presentation:** 3 good
**Contribution:** 4 excellent

**Summary:**

Reward-free RL is a well-studied RL setting where the agent must explore an unknown environment without access to the rewards, and then given an arbitrary number of reward functions, must propose a near-optimal policy for each of them. This work builds on results on reward-free RL in the linear function approximation setting and shows that reward-free RL is possible for general function classes (with bounded Bellman-Eluder dimension).

----------------

Update: After reading the author's reply and other reviews, I would like to maintain my score of 7.

**Questions:**

It was not clear to me why the algorithm uses the difference function class (i.e. $\mathcal{F}^{on}$). Could the authors explain this?

**Limitations:**

Limitations are discussed. No discussion is given on societal impact. While the work is primarily theoretical and may not have obvious immediate societal impact, it is contributing to the larger field of RL which does have practical impact. Given this, I would encourage the authors to consider what such impacts might be, and how negative impacts might be mitigated.

**Strengths And Weaknesses:**

Strengths:
Both reward-free RL and RL with general function approximation are problems of much interest in the RL community. This work is the first to study the reward-free problem in the general function approximation setting. The results obtained (while not optimal) are likely close to the optimal rates. Several corollaries are given in various specific settings of interest specializing the results to these settings, and a lower bound is also given showing that in some settings (the linear completeness and unknown features setting) reward-free RL is not in general efficient.

The algorithm is a modification of an existing algorithm (OLIVE) yet it is not a priori obvious how to extend OLIVE to the reward-free setting, and I believe the extension proposed in this work is technically interesting.

Overall, I believe this work makes a fundamental contribution to the theory of RL and will be of much interest to the RL theory community. Furthermore, it is well-written and appears to be technically sound.

I did not see any major weaknesses but had several minor questions and corrections:
- It is stated that “prior works require explorability/reachability assumptions, which, roughly speaking, assert that every direction in the feature space can be visited with sufficient probability. These assumptions are often not needed in reward-aware RL but suspected to be necessary for reward-free settings.” It is true that works like [Zanette et al., 2020] do require such assumptions, but other works on reward-free RL with linear function approximation (e.g. [Wang et al., 2020], [Wagenmaker et al., 2022]) do not require such assumptions. The statement above seems to imply the opposite—I believe this should be clarified.
- Some more discussion on Q-type vs V-type would be helpful. It’s not entirely clear why we need to consider two separate settings unless the reader has seen them before.
- In Section 3.1, perhaps reference Definition 6 or otherwise remind the reader what $\phi^{lc}$ corresponds to.

---

> ### Author Response · Authors · 2022-08-02
> **Response to Reviewer A559**
>
> We thank Reviewer A559 for recoganizing our work to have "fundamental contribution to the theory of RL and will be of much interest to the RL theory community". Please find our response below.
>
> -----
>
> 1\. Q: "It is stated that “prior works require explorability/reachability assumptions, which, roughly speaking, assert that every direction in the feature space can be visited with sufficient probability. These assumptions are often not needed in reward-aware RL but suspected to be necessary for reward-free settings.” It is true that works like [Zanette et al., 2020] do require such assumptions, but other works on reward-free RL with linear function approximation (e.g. [Wang et al., 2020], [Wagenmaker et al., 2022]) do not require such assumptions. The statement above seems to imply the opposite—I believe this should be clarified."
>
> A: The linear MDP studied in Wang et al., (2020) and Wagenmaker et al. (2022) is more restricted than and subsumed by the setting in Zanette et al. (2020b) and Modi et al. (2021). Here we mainly wanted to highlight the assumptions which are suspected to be necessary in these more general settings (e.g., linear completeness). This is also summarized in Table 1, where only we include and highlight explorability and reachability in row 2 and row 5 respectively. We will add more clarification on this issue.
>
> -----
>
> 2\. Q: "Some more discussion on Q-type vs V-type would be helpful. It’s not entirely clear why we need to consider two separate settings unless the reader has seen them before."
>
> A: Thanks for the comment! This is rooted in the reward-aware general function approximation setting. Both Q and V type BE dimension can capture interesting examples that are exclusive in the other type. Therefore, naturally there also exists two separate settings in the reward-free case. We will add more discussions in the paper. Please refer to the common response to all reviewers for more details.
>
> -----
>
> 3\. Q: "In Section 3.1, perhaps reference Definition 6 or otherwise remind the reader what $\phi^{lc}$ corresponds to."
>
> A: Thanks for the suggestion! We will address that in the next version.
>
> -----
>
> 4\. Q: "It was not clear to me why the algorithm uses the difference function class (i.e. $\mathcal F^{\mathrm{on}}$). Could the authors explain this?"
>
> A: In the proof, we show the constraints we gather in the online phase is sufficient to eliminate any bad function in the offline phase (line 312). However, simply using $\mathcal F$ in the online phase does not give us such a good guarantee and we need to use this difference function class $\mathcal F^{\mathrm{on}}$. This is also one novel part in our analysis. More details can be found in line 604-606, 609, 628-634 and around. We are willing to discuss more here if the reviewer is interested in the details.

---

### Official Review · Reviewer_Xxv4 · 2022-07-11

**Rating:** 6
**Confidence:** 3
**Soundness:** 3 good
**Presentation:** 3 good
**Contribution:** 3 good

**Summary:**

The paper studies reward-free RL with non-linear function approximation. They propose an extension to the OLIVE algorithm of Jiang et al. (called RFOlive) for which they establish polynomial sample complexity in relevant problem variables (using Bellman-eluder-dimension-based arguments).  As a positive side result, they show that the explorability or reachability assumptions used in prior works are actually not necessary for achieving polynomial sample complexity. As a negative result, they show that polynomial sample complexity in all relevant variables is impossible in the setting where linear completeness holds for unknown features (despite realizability).

**Questions:**

Overall, while there are some limitations, I think that the paper presents interesting and significant results. I am thus more inclined towards acceptance. Here a few comments/questions:

1. As stated above, I am quite puzzled about the novelty of this work (specifically of the algorithm and its analysis). I am not claiming that the paper is not novel, but rather that it was unclear whether it is sufficiently novel after reading it. This would be an important point to clarify in the rebuttal and in an updated version of the paper.

2. V-type RFOlive seems to be more complicated as an algorithm (eg it must build the cover of F explicitly) and has worse sample complexity (eg depending on K). What is its advantage over the Q-type variant?

3. The result for low-rank MDPs with unknown features (Corollary 4) is stated only for V-type RFOlive. Is it possible to have it also for the Q-type variant?

4. Theorem 5 is a little informal and not very clear. Does the fact that a polynomial sample complexity in all stated variables is impossible mean that, for any algorithm, there exists at least one instance where the sample complexity is exponential in at least one of those variables ? In that case, it should be clarified

Other minor comments:
- When first introducing the function class at line 107, it is not clear what it should model (and footnote 3 is confusing), though everything becomes clear later on. I would anticipate a comment on that.
- Line 147: the rightmost x_h should be a_h
- Line 252: "dateset"

**Limitations:**

Limitations have been properly discussed

**Strengths And Weaknesses:**

Strengths:
- The paper studies a relevant problem (RF RL with general function approximation) which was previously unaddressed
- The results are significant and they do not rely on strong assumptions
- The hardness result of Section 4 is significant and of independent interest
- The paper is well-written and easy to follow. All results are clearly stated (though they can be fully understood only by looking at the appendix)

Weaknesses:
- Novelty: it seems to me that the proposed algorithm is essentially a direct application of Olive (or its Q-type variant by Jin et al.) with the zero reward function. The authors mention in the introduction the RFOlive is "nontrivially adapted from its reward-aware counterpart", though it is not clear what are the novel algorithmic components here. Moreover, while Section 3.3 specifies what are some novel aspects in the analysis, it would be good to elaborate a little more on how much of the analysis is actually novel and how much is taken from prior works
- The proposed algorithm is computationally intractable as it explicitly maintains version spaces. Even worse, the V-type variant of RFOlive maintains explicitly a finite cover of the function class (which is of exponential size in the dimension). Therefore, there is no real practical impact
- No numerical result is presented (though, as a consequence of the previous point, it is not possible to do much beyond very toy problems)

---

> ### Author Response · Authors · 2022-08-02
> **Response to Reviewer Xxv4**
>
> We thank Reviewer Xxv4 for mentioning both our positive and negative results are significant and of interest and our paper is well-written and easy to follow. Please find our response below.
>
> -----
>
> 1\. Q: Novelty
>
> A: Please refer to our bullet points 2 and 3 in the general response for the discussion about novelty.
>
> -----
>
> 2\. Q: "The proposed algorithm is computationally intractable as it explicitly maintains version spaces. Even worse, the V-type variant of RFOlive maintains explicitly a finite cover of the function class (which is of exponential size in the dimension). Therefore, there is no real practical impact" and "No numerical result is presented (though, as a consequence of the previous point, it is not possible to do much beyond very toy problems)"
>
> A: Our paper makes it clear in the title itself that the focus is on statistical rather than computational efficiency. It is quite common to study statistical limits of various problem settings without computational efficiency, in all areas of machine learning and computer science more broadly. We respectfully disagree that such works have no practical impact, even if it is often more delayed. In the specific case here, there are no known computationally-efficient techniques even in the easier and more studied reward-aware setting, which work for general non-linear RL problems with low BE dimension/Bellman rank. Please also refer to the first bullet point in our general response.
>
> -----
>
> 3\. Q: "V-type RFOlive seems to be more complicated as an algorithm (eg it must build the cover of F explicitly) and has worse sample complexity (eg depending on K). What is its advantage over the Q-type variant?"
>
> A: We believe this question is rooted in the reward-aware general function approximation setting. As mentioned in the common response to all reviewers, V-type permits feature learning and other non-linear scenarios not easily captured in Q-type. We refer the reviewer to the simple contextual bandit lower bound for Q-type in Agarwal and Zhang (2022) and please also find related discussions in the common response.
>
> -----
>
> 4\. Q: "The result for low-rank MDPs with unknown features (Corollary 4) is stated only for V-type RFOlive. Is it possible to have it also for the Q-type variant?"
>
> A: That is an interesting question! The recent paper (Agarwal and Zhang, 2022) provides a lower bound on the Q-type Bellman rank for contextual bandit problem with a realizable reward class in Appendix B. We can construct an $H=1$ low-rank MDP and set the feature class $\Phi$ to be the reward class in Agarwal and Zhang (2022). Then their lower bound implies that the natural way to construct the function class $\mathcal F$ as all the linear function w.r.t. $\Phi$ in our paper will fail due to the large Q-type Bellman rank. Therefore, we cannot use the Q-type variant here.
>
> -----
>
> 5\. Q: "Theorem 5 is a little informal and not very clear. Does the fact that a polynomial sample complexity in all stated variables is impossible mean that, for any algorithm, there exists at least one instance where the sample complexity is exponential in at least one of those variables ? In that case, it should be clarified"
>
> A: Yes, you are correct here. It means that for any algorithm, there exists one instance (MDP) in our constructed MDP class, where the algorithm need exponential number of samples in at least one of those variables. We will clarify that in the next version.
>
> 6\. Thanks for the minor comments! We will address them.
>
> -----
>
> Reference:
>
> Agarwal, A., Zhang, T.. (2022). Non-Linear Reinforcement Learning in Large Action Spaces: Structural Conditions and Sample-efficiency of Posterior Sampling. Proceedings of Thirty Fifth Conference on Learning Theory, 178:2776-2814, PMLR.

---

> > ### Comment · Reviewer_Xxv4 · 2022-08-08
> > **Response to Response**
> >
> > I thank the authors for the detailed response. It clarified all my doubts. I will keep my initial score.

---

### Official Review · Reviewer_GjCj · 2022-07-11

**Rating:** 5
**Confidence:** 3
**Soundness:** 3 good
**Presentation:** 3 good
**Contribution:** 2 fair

**Summary:**

This work studies different settings of reward free reinforcement learning under general function approximation, including linear MDPs, linear completeness, and low-rank MDPs. The main contributions are the release of exploitability or reachability assumptions in previous works. Moreover, it shows the intrinsic hardness of the RL problems under linear completeness assumptions with comparisons to low-rank ones.

**Questions:**

1. In Line 217, it mentioned that adapting Golf may lead to a sharper result. So why not directly adapt that state-of-art method?
2. Since I'm not so familiar with the literature on reward-free, why do we need two types of algorithms (Q and V)? what is the difference between them? Or, what are the advantages of them respectively. The motivation to have them is not introduced clearly.

**Limitations:**

This work already mentioned the limitations such as the computation cost of the proposed algorithms are still expensive.

**Strengths And Weaknesses:**

Strengths:
1. The organization of this paper is clear and easy to read.
2. This work proposed provable algorithms for different settings without reachability/explorability assumptions that are necessary for previous works, which is a big improvement.

Weakness:
1. The results in Table.1 only show the difference in the settings/assumptions of different works, while lacking the comparisons of the sample complexity or other main results. Adding the comparisons of the main results may be more clear.
2. Although the main results show provable algorithms without reachability/explorability assumptions that are necessary for previous works,  the sample complexity bound is also worse in H as claimed in Line 238. It is confusing that if this is a tradeoff that releasing assumptions will lead to worse sample complexity or due to technical tools?

---

> ### Author Response · Authors · 2022-08-02
> **Response to Reviewer GjCj**
>
> We thank Reviewer GjCj for appreciating the organization of our paper and the big improvement upon prior works. Please find our response below.
>
> -----
>
> 1\. Q: "The results in Table.1 only show the difference in the settings/assumptions of different works, while lacking the comparisons of the sample complexity or other main results. Adding the comparisons of the main results may be more clear."
>
> A: Thanks for the suggestion! Our major contribution is to investigate the minimal assumptions for statistically efficient reward-free RL algorithms. Therefore given the limited space we only expose and highlight the assumptions (settings) in Table 1, while deferring most of the detailed comparisons of sample complexities to the later part of the main text and the appendix (see e.g., line 714, 738, 927). We will add clearer discussions/pointers in the next version. We also refer the review to common response for more details.
>
> -----
>
> 2\. Q: "Although the main results show provable algorithms without reachability/explorability assumptions that are necessary for previous works, the sample complexity bound is also worse in H as claimed in Line 238. It is confusing that if this is a tradeoff that releasing assumptions will lead to worse sample complexity or due to technical tools?"
>
> A: We want to highlight that our bound only \emph{appears} to be worse in $H$ factors. As we discussed in line 714, the result in Zanette et al. (2020b) has a hidden $1/\nu_{\min}$ explorability factor (by only considering $\varepsilon$ that is "asymptotically small"  relative to $\nu_{\min}$). Since such a factor can be arbitrarily large while $H$ is always bounded in a fixed horizon problem, our bound could be much tighter than theirs. Similar dependence on reachability factor $1/\eta_{\min}$ also exists in the sample complexity bounds of block MDPs (Du et al., 2019; Misra et al., 2020) and low-rank MDPs (Modi et al., 2021) as they assume the reachability assumption. In summary, by making additional assumptions, these works at the same time pay additional dependence on the reachability/explorability factor, which can be arbitrarily worse than ours. In terms of the $H$ dependence itself, we believe there is still some room to improve it.
>
> -----
>
> 3\. Q: "In Line 217, it mentioned that adapting Golf may lead to a sharper result. So why not directly adapt that state-of-art method?"
>
> A: This is a great question! We choose OLIVE because it is conceptually a little cleaner than GOLF due to the simpler average Bellman error constraint, and this allows us to better focus on the complexities of the reward-free setting. Adapting our analysis with GOLF as the base template is an interesting avenue for future work!
>
> -----
>
> 4\. Q: "Since I'm not so familiar with the literature on reward-free, why do we need two types of algorithms (Q and V)? what is the difference between them? Or, what are the advantages of them respectively. The motivation to have them is not introduced clearly."
>
> A: The reason that we study both Q and V types is not specific to reward-free learning. It is because different (Q and V types) versions exist in the reward-aware general function approximation RL (e.g., Jiang et al., 2017; Jin et al., 2021; Du et al., 2021). They can capture different interesting examples and we refer the reader to check these references. Please also refer to the common response for more details.
>
> In terms of the difference between Q-type and V-type algorithms, V-type RFOLIVE (or V-type OLIVE) requires one uniform action in exploration and therefore has an additional $K$ factor (the cardinality of action space) in the sample complexity bound.
>
> -----
>
> 5\. We also want to mention that the hardness result is also one of our major contributions. It provides the first sharp exponential separation between linear MDPs and linear completeness setting. Together with the positive results, we obtain a clear roadmap towards statistically efficient reward-free RL.

---

### Official Review · Reviewer_zknp · 2022-07-23

**Rating:** 5
**Confidence:** 3
**Soundness:** 3 good
**Presentation:** 3 good
**Contribution:** 2 fair

**Summary:**

The authors consider reward-free reinforcement learning under general class of function approximation. They propose the RFO-LIVE algorithm, a reward-free version of the OLIVE algorithm by Jiang et al., 2017, for reward-free under minimal structural assumptions. In particular, the considered set of assumptions covers the previously studied settings of linear MDPs (Jin et al., 2020b), linear completeness (Zanette et al., 2020b), and low-rank MDPs with unknown representation (Modi et al., 2021). In particular, they show that previous assumptions made for the latter two settings are unnecessary for reward-free RL. Finally, they provide a lower bound showing that reward-free (and even reward-aware) exploration under linear completeness assumptions when the underlying features are unknown is intractable.

**Questions:**

General comments:
-It would be interesting to compare clearly the obtained sample complexity for the proposed algorithms with the ones of previous work. Additionally, it could also be interesting to provide lower bounds for the different settings (particularly those of Corollary 2 and 4) and compare them to the obtained rates.
-

Specific comments:
-L45: At this point it is not clear what is the set of constraints.
-L127: Do you have simple concrete examples of MPD and associated classes \cR and \cF which verify these two assumptions?
-L160, Algorithm 1: It may be clearer to specify in the algorithm what you keep in memory from the online phase for the offline phase, i.e. (\pi^t,h^t,\cD^t).
-L186,(3): h_t instead of h.
-L200: Do you have an example where this assumption is verified?
-L235: Is there any lower bound for this setting to compare with? And how the obtained rate compare to the one in Zanette et al.(2020b).
-L267:  Can you discuss the differences between Q-type and V-type RFOlive, in partícular in terms of theoretical guarantees.
-L290: Can you provide a table with the different rates given the algorithm and the setting to ease the comparison with the previous baselines?
-L296:Can you explain why it 'significantly improves' over the previous bounds.
-L551:By realizability you mean that assumption 5 is verified?
-L558: At this point, it is difficult to follow the proof without reading one of Theorem 18 by Jin et al. 2021. And I think you should include this argument in your proof.
-L581: A hat is missing.
-L597 Maybe you should precise the events.

**Limitations:**

Yes.

**Strengths And Weaknesses:**

Contributions:
-Algorithm RFOlive an adaptation for reward-free exploration of the Olive algorithm by Jiang et al. (2017). Novelty: low; relevance: medium.
-Exploitability assumptions for linear completeness setting and reachability for low-rank MDPs are not necessary for reward-free exploration. Novelty: low;  relevance: high.
-Reward-free (and reward-aware) exploration with linear completeness assumptions and when the
underlying features are unknown is intractable. Novelty: medium, relevance: medium.

The paper is well-written. The proofs seem correct from what I read. Note that most of the technical argument used in the proofs of Theorem 1 and 2 seem to be adapted from Jin et al. 2021, see specific comments. I think characterizing the sample complexity of reward-free exploration for general function approximation is a valuable contribution, especially knowing which assumption is necessary or not. The improvements in terms of sample complexity over the previous work seem a bit incremental. But my main concern is that the proposed algorithms are computationally intractable. Indeed, usually, the interest of such computationally inefficient algorithm is to characterize the optimal rate for the sample complexity which seems not to be the case here.

---

> ### Author Response · Authors · 2022-08-02
> **Response to Reviewer zknp**
>
> We thank Reviewer zknp for the detailed comments and suggestions. Please find our response below.
>
> -----
>
> 1\. Q: "General comments: -It would be interesting to compare clearly the obtained sample complexity for the proposed algorithms with the ones of previous work. Additionally, it could also be interesting to provide lower bounds for the different settings (particularly those of Corollary 2 and 4) and compare them to the obtained rates. -"
>
> A: As acknowledged by the reviewer, our major contribution is to investigate the minimal structural assumptions for statistically efficient reward-free RL algorithms. Therefore, given the limited space, we defer most of the detailed comparisons of sample complexities to the appendix (see e.g., line 714, 738, 927). We will add clearer discussions/pointers in the next version. We agree that showing lower bounds for the different settings would certainly be interesting future directions but for these cases we can compare our bounds with the lower bounds for tabular and linear MDPs (Jin et al., 2020a and Wagenmaker et al., 2022) which are special cases of studied settings.
>
> -----
>
> 2\. Q: "-L127: Do you have simple concrete examples of MPD and associated classes $\mathcal R$ and $\mathcal F$ which verify these two assumptions?"
>
> A: Yes, as we have shown in Corollary 2, Corollary 4 and Corollary 6, these assumptions hold in low-rank MDPs with unknown feature (Modi et al., 2021), linear completeness setting with known feature (Zanette et al., 2020b), and linear MDPs (Jin et al., 2020b).
>
> -----
>
> 3\. Q: "-L200: Do you have an example where this assumption is verified?"
>
> A: It is a standard component in bounds in statistical learning theory literature where the metric entropy of many function classes can be shown to have a parametric growth (Mendelson et al. 2002, Mohri et al. 2018). We want to clarify that it is not an assumption that we make in the paper. It is purely for the cleanness of the presentation and readability. We provide the full version of the theorem in the appendix (line 540).
>
> -----
>
> 4\. Q: "-L235: Is there any lower bound for this setting to compare with? And how the obtained rate compare to the one in Zanette et al.(2020b)."
>
> A: We are not aware a direct lower bound in this setting to compare with. But the lower bounds in Jin et al. (2020a) and Wagenmaker et al. (2022) are applicable here as these are simply special cases of the linear completeness setting. The detailed comparison of the rate to the one in Zanette et al., 2020b can be found around line 714.
>
> -----
>
> 5\. Q: "-L267:  Can you discuss the differences between Q-type and V-type RFOlive, in particular in terms of theoretical guarantees."
>
> A: The major difference between Q-type and V-type RFOLIVE is due to the difference between Q-type and V-type OLIVE in reward-aware RL. In summary, V-type RFOLIVE (and OLIVE) requires one uniform action in exploration and therefore has an additional $K$ factor (the cardinality of action space) in the sample complexity. However, they can capture different interesting examples that are exclusive in the other. Please refer to the common response to all reviewers for further details.
>
> -----
>
> 6\. Q: "-L290: Can you provide a table with the different rates given the algorithm and the setting to ease the comparison with the previous baselines?"
>
> A: Thanks for the suggestion! We already have some comparisons of sample complexities in the appendix (see e.g., line 714, 738, 927). We will add clearer a table/discussions/pointers in the next version.
>
> -----
>
> 7\. Q: "-L296:Can you explain why it 'significantly improves' over the previous bounds."
>
> A: We refer the reviewer to line 927 for detailed discussions. Our rate is significantly better in most terms (e.g. $d_{\mathrm{lr}}^3$ vs $d_{\mathrm{lr}}^8$, $K$ vs $K^{13}$, and no $1/\eta_{\min}$ dependence) while only slightly worse in $H$ factor. As mentioned in the common response, the reachability factor $\eta_{\min}$ can be arbitrarily small. Improving the sample complexity result in our general case is an interesting avenue for future work.
>
> -----
>
> 8\. Q: "-L551:By realizability you mean that assumption 5 is verified?"
>
> A: Yes, you are correct.
>
> -----
>
> 9\. Thanks for other (minor) comments and suggestions! We will address them in the subsequent version.
>
> -----
>
> 10\. We also refer the reviewer to our general response to the computational tractability and novelty/contributions in the paper.
>
> -----
>
> References:
>
> Mendelson, S., & Vershynin, R. (2002, July). Entropy, combinatorial dimensions and random averages. In International Conference on Computational Learning Theory (pp. 14-28). Springer, Berlin, Heidelberg.
>
> Mohri, M., Rostamizadeh, A., & Talwalkar, A. (2018). Foundations of machine learning. MIT press.

---

> > ### Comment · Reviewer_GjCj · 2022-08-07
> > **Response**
> >
> > Thanks for the answers from the authors and it addresses almost all my concerns. I will keep the score for this moment.

---

### Author Response · Authors · 2022-08-02
**General response**

We thank all the reviewers for constructive comments and suggestions. Here we provide general responses to some common questions raised by different reviewers.

------

**1. Computational tractability** (zknp, GjCj, Xxv4)

Proposing a statistically efficient algorithm for reward-free RL in general function approximation with minimal structural assumptions itself is already challenging, and we focus on this problem and take the first step in this paper. The computational intractability of our algorithm is rooted in its reward-aware counterpart OLIVE and such difficulties also widely exist in reward-aware general function approximation (Jiang et al., 2017; Dann et al., 2018; Jin et al., 2021; Du et al., 2021). It is quite common to study statistical limits of various problem settings without computational efficiency, in all areas of machine learning and computer science more broadly. We leave the computationally friendly and more practical variant (and empirical investigations) as the interesting future work.

------
**2. Novelty of RFOLIVE** (zknp, Xxv4)
- Comparison to other reward-free exploration works: RFOLIVE is the first algorithm to address reward-free exploration under general function approximation. Contrary to prior works which either try to reach all states or all directions in the feature space, we propose a novel value function elimination template and ensure that the collected exploration data can be used to identify and eliminate non-optimal value functions for downstream planning. This significantly sets us apart from the existing reward-free exploration works.

- Comparison to reward-aware OLIVE: While the final algorithm presented in the paper might look like a relatively easy adaptation of OLIVE, we emphasize that while the modification of OLIVE analyzed here works, many other natural modifications do not work in our setting. We will include a discussion of the variants which do not work in the final version. We also note that the way we use OLIVE in the offline setting is novel to our knowledge and is crucial to getting a good sample complexity under our assumptions, as opposed to more standard FQI style approaches.
Because we have to coordinate between the online and offline phases, the analysis bears significant novelty beyond the original analysis of OLIVE (and its reward-aware follow-up works), and this is one of our key contributions; see next bullet point for more details.
------
**3. Technical novelty over reward-aware OLIVE** (zknp, Xxv4)

The key step of the analyses of reward-aware OLIVE (Jiang et al. 2017; Jin et al. 2021) is to show that any bad function whose average Bellman error is large under the given reward function is eliminated (recall that they only have the online phase and the reward is always revealed). This is ensured by the online exploration process. However, the difficulty in our reward-free RL setting is that such a reward function is only revealed in the offline phase, where we no longer actively explore. To overcome this difficulty, we use completely new and novel proof techniques here: For each bad function $g\in \mathcal F_{\mathrm{off}}(R)$ with a large average Bellman error under the true reward $R$, we construct a surrogate function $\tilde f$ in the online phase. Our construction guarantees that $\tilde f$ has the same large average Bellman error as $g$, but the error is instead under the zero reward which we use during exploration (Eq. (9)). Then we show that all these constructed $\tilde f$ belong to the "difference" function class $\mathcal F_{\mathrm{on}}$ and $\tilde f$ will be eliminated in the online phase since we use $\mathcal F_{\mathrm{on}}$ and zero reward there. The collected data tuples (gathered constraints) that eliminate $\tilde f$ will be used in the offline phase and they guarantee eliminating its corresponding bad function $g$. Notice that in the design of $\tilde f$, we need to guarantee that it has a large average Bellman error at the same timestep as $g$ does so that it can correctly witness the average Bellman error of $g$, which we ensure via a Bellman backup construction (line 606).

In summary, both the construction of the surrogate function $\tilde f$ and the translation of average Bellman error from bad function $g\in\mathcal F_{\mathrm{off}}(R)$ to $\tilde f\in\mathcal F_{\mathrm{on}}$ are novel to the best of our knowledge. They reflect crucial difference between reward-aware and reward-free RL. And at the same time, no reward-aware RL works have used such mechanisms before.

---

> ### Author Response · Authors · 2022-08-02
> **General response**
>
> **4. Minimal (weaker) structural assumptions and more detailed comparisons of sample complexity rates** (zknp, GjCj)
>
> Our major contribution is to investigate the minimal structural assumptions for statistically efficient reward-free RL algorithms. Therefore, given the limited space, we defer most of the detailed comparisons of sample
> complexities to the appendix (see e.g., line 714, 738, 927). We will add clearer discussions/pointers in the next version.
>
> Further, the goal of this paper is to expand the scope of tractable reward-free RL to most problem classes where reward-aware RL is possible. There are still open gaps in such general settings in the more studied reward-aware version too, and understanding these issues better is an important direction for future work, and our work lays the foundation for such explorations in the reward-free setting.
>
> -----
>
> **5. Q-type vs V-type** (all reviewers)
>
> The reason that we study both Q and V types is not specific to reward-free exploration. Different versions  (Q and V types) already exist in the reward-aware general function approximation RL (e.g., Jiang et al., 2017, Jin et al., 2021; Du et al., 2021). They capture different scenarios of interest and so far, it seems difficult to unify them even in the reward-aware setting. Therefore, to give a comprehensive treatment of general function approximation we need to consider both together. Fortunately, the algorithms and analyses for the two types are not very different, with only moderate differences (see details below).
>
> In terms of the difference between Q-type and V-type algorithms, V-type permits representation learning and other non-linear scenarios that are not easily captured in Q-type. For instance, any contextual bandit problem is admissible under the V-type assumption (the V-type Bellman rank is 1), while Q-type does not capture all finite action, non-linear contextual bandit problems with a realizable reward. We refer the reviewer to the detailed lower bound on the Q-type Bellman rank in the contextual bandit setting in Appendix B of Agarwal and Zhang (2022). Further, V-type RFOLIVE (or V-type OLIVE) requires one uniform action in exploration and therefore has an additional $K$ factor (the cardinality of action space) in the sample complexity bound.
>
> -----
>
> Reference:
>
> Agarwal, A., Zhang, T.. (2022). Non-Linear Reinforcement Learning in Large Action Spaces: Structural Conditions and Sample-efficiency of Posterior Sampling. Proceedings of Thirty Fifth Conference on Learning Theory, 178:2776-2814, PMLR.

---

### Comment · Area_Chair_Tbh5 · 2022-08-04
**Revised version of the paper**

Dear authors,

thank you taking the time to answer to the reviewers. In the rebuttal, you mentioned multiple times that you will address several or the reviewer's questions in the revised version of the paper. I would like to remind that you have the opportunity to upload a revised version of the paper before August 9th (see NeurIPS 2022 FAQ for Authors for more details https://nips.cc/Conferences/2022/PaperInformation/NeurIPS-FAQ). I would really appreciate that since this would allow the reviewers to investigate the changes.

Thank you.

---

> ### Author Response · Authors · 2022-08-08
> **Rebuttal revision uploaded**
>
> Dear Area Chair Tbh5 and all reviewers,
>
> We would like to thank the AC for the reminder and thank you all again for your time and suggestions!
>
> As suggested by the AC, we have prepared a rebuttal revision (we need to rely on the extra page limit in the camera ready version for the changes) and uploaded it  as the **supplementary material** . We summarize the changes we made in the revision ($\textcolor{orange}{\text{highlighted in orange}}$) below:
>
> ---
> 1\. **Novelty of RFOLIVE**
>
> (i) We add more discussions and a pointer in Section 3.3.
>
> (ii) We add more discussions in Section 5.
>
> (iii) We add the technical novelty of RFOLIVE over reward-aware OLIVE in Appendix C.3.
>
> 2\. **More detailed comparisons of sample complexity rates and lower bounds**
>
> (i) We add a pointer in the caption of Table 1.
>
> (ii) We clarify the discussion in Section 3.1.
>
> (iii) We add a pointer in Section 3.2.
>
> (iv) We add the comparison table and detailed discussions about sample complexity rates and lower bounds in Appendix A.
>
> 3\. **Q-type vs V-type**
>
> (i) We add a pointer in the footnote 4 in page 4.
>
> (ii) We add the detailed discussion in Appendix B. In addition to the changes we mentioned in the rebuttal, we give a concrete example where the Q-type BE dimensions can be shown to be exponentially large compared to the V-type BE dimension.
>
> 4\. We also address some other (minor) comments/concerns raised by reviewer znkp, Xxv4, and A559. Among them, more significant changes in the main text are also marked.
>
> ---
> We will address other changes that take longer time in the final version. Please also notice that the line number, equation number, and section number in our previous response/rebuttal refer to **the version that was reviewed before the rebuttal**. They can be inconsistent in our rebuttal revision version.

---

### Meta-Review · Area_Chair_Tbh5 · 2022-08-26

**Recommendation:** Accept
**Confidence:** Certain

**Metareview:**

Despite a few concerns about the novelty of the paper (mostly from an algorithmic perspective), I think the investigation of the minimal structural assumptions for reward-free RL is interesting for the community. This work provides a first step in the direction of obtaining a better understanding about reward-free RL outside the tabular setting.

**Award:**

No

---

### Decision · Program_Chairs · 2022-09-14

Accept